    

# Acquired resistance to PD-L1 inhibition enhances a type I IFN-regulated secretory program in tumors

Yuhao Shi[1], Amber McKenery[2], Melissa Dolan[1], Michalis Mastri[2], James W Hill[3], Adam Dommer[2], Sebastien Benzekry [4], Mark Long[5], Scott I Abrams [6], Igor Puzanov[7] & John M L Ebos [1,2,7✉]

## Abstract

Therapeutic inhibition of programmed cell death ligand (PD-L1) is linked to alterations in interferon (IFN) signaling. Since IFN-regulated intracellular signaling can control extracellular secretory programs in tumors to modulate immunity, we examined IFN-related secretory changes in tumor cells following resistance to PD-L1 inhibition. Here we report an anti-PD-L1 treatment-induced secretome (PTIS) in tumor models of acquired resistance that is regulated by type I IFNs. These secretory changes can suppress activation of T cells ex vivo while diminishing tumor cell cytotoxicity, revealing that tumor-intrinsic treatment adaptations can exert broad tumor-extrinsic effects. When reimplanted in vivo, resistant tumor growth can slow or stop when PTIS components are disrupted individually, or when type I IFN signaling machinery is blocked. Interestingly, genetic and therapeutic disruption of PD-L1 in vitro can only partially recapitulate the PTIS phenotype highlighting the importance of developing in vivo-based resistance models to more faithfully mimic clinically-relevant treatment failure. Together, this study shows acquired resistance to immune-checkpoint inhibitors 'rewires' tumor secretory programs controlled by type I IFNs that, in turn, can protect from immune cell attack.

Keywords PD-L1; Resistance; Secretome; IFN; Immune-checkpoint
Subject Categories Cancer; Immunology; Signal Transduction

## Introduction

Cancer therapies can provoke unexpected (and often unwanted) cellular reactions that include the secretion of proteins such as growth factors and cytokines—many of which have been exploited as possible biomarkers of treatment effect or toxicity in patients (Ebos, 2015; Kerbel and Ebos, 2010; Madden et al, 2020). Such therapy-induced secretomes (TIS) can also contribute to cancer progression, particularly in settings of acquired resistance where tumor cell populations adapt to treatments over prolonged periods (Madden et al, 2020). For immune-checkpoint inhibitors (ICIs) that target the programmed cell death 1 (PD-1) pathway, early cytokine changes (e.g., IL6, IL8) in patients after treatment can correlate with initial responses (Sanmamed et al, 2017; Tsukamoto et al, 2018a) or adverse events (Naqash et al, 2018), but less is known about whether tumor-specific secretory profile changes can be a cause or a consequence of acquired resistance (Bridge et al, 2018; Madden et al, 2020).

In this regard, tumoral control of secretory programs by interferons (IFNs) may be of interest in the setting of acquired resistance to PD-L1 inhibition for several reasons (Madden et al, 2020; Mastri et al, 2018a). First, IFNs have been linked to multiple ICI treatment resistance mechanisms, mostly via the induction of IFN-stimulated genes (ISGs) activated by type I (α/β) and type II (γ) IFN subtypes. Currently, the precise effect of IFNs on anti-tumor immunity in the presence of PD-L1 inhibitors remains enigmatic because they can, somewhat paradoxically, both protect and weaken immune defenses (often simultaneously) (Benci et al, 2019; Benci et al, 2016). For instance, IFNs can boost antigen presentation (e.g., via beta-2-microglobulin and MHC-I expression) to improve PD-1 inhibitor responses (Sade-Feldman et al, 2017; Vraetz et al, 1999), while also suppressing immune cell attack via the induction of T-cell inhibitory ligands (Benci et al, 2016; Garcia-Diaz et al, 2017) and proteins, such as NOS2 (Jacquelot et al, 2019) and many others (Boukhaled et al, 2021; Chen et al, 2019; Chen et al, 2018a). Second, IFNs also can regulate a range of cellular processes that involve additional cytokine production that, in turn, can have positive and negative effects on tumor progression (Cheon et al, 2014; Parker et al, 2016). Finally, several studies have now identified a crosstalk between tumor intrinsic PD-L1 functions and IFN signaling that control STAT3/Caspase 7 pathways (Gato-Canas et al, 2017) and tumor cell DNA damage response (Cheon et al, 2021) —all of which can regulate protein production with extrinsic functions (Pilger et al, 2021; Yu et al, 2009). Currently it is unknown whether these IFN-controlled secretory programs are enhanced or inhibited in tumor cells in the context of acquired resistance where persistent PD-L1 blockade may impact immune-protective processes.

[1]Department of Experimental Therapeutics, Roswell Park Comprehensive Cancer Center, Buffalo, NY 14263, USA. [2]Department of Cancer Genetics and Genomics, Roswell Park Comprehensive Cancer Center, Buffalo, NY 14263, USA. [3]Jacobs School of Medicine and Biomedical Sciences, SUNY at Buffalo, Buffalo, USA. [4]Computational Pharmacology and Clinical Oncology (COMPO), Inria Sophia Antipolis-Méditerranée, Centre de Recherches en Cancérologie de Marseille, Inserm U1068, CNRS UMR7258, Institut Paoli-Calmettes, Faculté de Pharmacie, Aix-Marseille University, Marseille, France. [5]Department of Biostatistics and Bioinformatics, Roswell Park Comprehensive Cancer Center, Buffalo, NY 14263, USA. [6]Department of Immunology, Roswell Park Comprehensive Cancer Center, Buffalo, NY 14263, USA. [7]Department of Medicine, Roswell Park Comprehensive Cancer Center, Buffalo, NY 14263, USA. ✉E-mail: John.Ebos@RoswellPark.org

To examine this, we generated anti-PD-L1 treatment-resistant (PTR) tumor cells to evaluate changes in secretory profiles regulated by IFN-signaling. PTR cell variants representing acquired resistance were generated using in vivo tumor models that were initially sensitive to anti-PD-L1 (αPD-L1) treatment and thus able to better mimic the complex tumor:immune cell interactions that occur during the progression to treatment failure. Using transcriptomic and proteomic analysis, we identified an αPD-L1 treatment-induced secretome (PTIS) gene signature in PTR cells that was enriched for ISGs and could be validated in multiple clinical and preclinical datasets involving αPD-L1 therapy. Using a knockdown model of the interferon alpha and beta receptor subunit 1 (IFNAR1), we found that the PTIS could be regulated by type I IFN signaling and that IFN-controlled secretory products in PTR cells could shield tumors from CD8 + T cell cytotoxicity. When reimplanted in vivo, targeting individual PTIS proteins such as IL-6 could slow PTR tumor growth but broader PTIS inhibition via type I IFN signaling disruption provided a more potent anti-tumor effect. Interestingly, when therapeutic and genetic methods were used to mirror chronic PD-L1 inhibition in vitro, the PTIS could be partially recapitulated suggesting tumor-intrinsic secretory changes depend, at least in part, on extrinsic host responses to treatment. Together, these results show that PD-L1 inhibition disrupts immune protective secretory programs in treatment-sensitive tumors and identify a unique 'rewiring' of type I IFN signaling that may be exploited as biomarkers and therapeutic targets for patient populations with acquired resistance.

## Results

### Acquired resistance to PD-L1 inhibition increases secretory profiles enriched for type I IFN-regulated genes

To examine acquired resistance to PD-L1 inhibition, the PD-1 pathway inhibitor-*sensitive* murine breast tumor EMT6 cell line (Clift et al, 2019; Lan et al, 2018; Schofield et al, 2021) was implanted orthotopically in BALB/c mice and treated with αPD-L1 (Clone 80) or IgG control antibody (Fig. 1A, schematic shown). Following continuous treatment until endpoint, a αPD-L1 treatment-resistant (PTR) cell variant (EMT6-PTR) was selected from mice with tumors that resumed growth after an initial significant delay (Fig. 1B, circles shown). Transcriptome RNA-sequencing of EMT6-PTR and EMT-P (parental) tumor tissues revealed multiple genes to be up- or down-regulated (Fig. 1C). Gene-set enrichment analysis (GSEA) showed EMT6-PTR tumors to be significantly enriched (FDR ≤ 0.25) for genes associated with extracellular matrix, growth factor, and cytokine signaling pathways, several of which were secreted and IFN-regulated (Fig. 1D; Appendix Table S1). Using the Gene Ontology (GO) database term GO:0005576 consisting of products outside or unattached to the cell (Dolan et al, 2024; Mastri et al, 2018a), secretory genes were found to increase in EMT6-PTR tumor transcripts and associate with inflammatory signaling, wound healing, and immune cell function/migration (Fig. 1E). Since many of these processes also associate with IFN signaling (Parker et al, 2016; Snell et al, 2017), we examined IFN-regulated genes using the Interferome database—a compilation of published in vitro and in vivo experimental datasets identifying transcriptomic and proteomic changes after IFN treatment

(Rusinova et al, 2013). Compared to P controls, EMT6-PTR tumors had several IFN-related genes up- and down-regulated (54% and 63%, respectively), with type I IFN gene upregulation the most common (22% of total) (Fig. 1F). To confirm association of IFN regulated genes in EMT6-PTR tumors, we assessed for relative enrichment of IFN signaling gene-sets found in several publications (Benci et al, 2019; Benci et al, 2016; Liu et al, 2019; Thorsson et al, 2018; Weichselbaum et al, 2008) and in the Hallmark Molecular Signatures Database (MSigDB) (Liberzon et al, 2015) where positive enrichment was observed for all datasets examined (Fig. 1G; Appendix Table S2). Similar positive enrichment was observed when GSEA was performed on an IFNγ-associated gene-set identified in durvalumab-treated non-small cell lung carcinoma (NSCLC) patient tumor biopsies (Higgs et al, 2018) (Fig. 1H). At the individual gene level, qRT-PCR analysis confirmed upregulation of several type I IFN-regulated ISGs in EMT6-PTR cell variants (Fig. 1I). To test whether this was specific to acquired resistance, we examined IFN and ISG expression in tumor cells following PD-L1 inhibition in an innately resistant model. To do this, we first identified a PD-1 pathway inhibitor-*insensitive* murine tumor model which we found after implanting the kidney RENCA cell line orthotopically into BALB/c mice (Mosely et al, 2017) and treating with αPD-L1 or IgG antibody (Fig. 1J). To compare to the in vivo-selected EMT6 P/PTR variants, we similarly selected a RENCA tumor cell variant that did not respond to treatment (Fig. 1K, circles shown). GSEA of IFN-associated gene sets showed a negative enrichment in the innately resistant αPD-L1 treated RENCA cells (RENCA-PDL1) compared to the IgG-treated variants (RENCA-P) (Fig. 1L). Taken together, these results demonstrate that αPD-L1 treatment can induce ISG-related secretory gene changes in tumors that are enhanced in acquired resistance settings.

### Identification of an αPD-L1 treatment-induced secretome (PTIS) signature regulated by type I IFN signaling

We next tested whether these enriched secretory effects after resistance could be linked to specific genes regulated by type I IFN signaling. To do this, we sought to identify an IFN-regulated secretory signature that significantly increased in EMT6-PTR tumors, and could be validated by various transcriptomic and proteomic techniques. Using the workflow shown in Fig. 2A, we generated an αPD-L1 treatment-induced secretome (PTIS) signature consisting of 5 genes that included CCL2, IL1RN, IL6, LCN2, and NOS2 (Fig. 2A, schematic shown). The PTIS was identified in the following steps. First, RNAseq data was used to identify IFN-regulated genes representing secreted products that increased in EMT6-PTR tumors (Fig. 2B, PTIS genes shown in right inset; full gene-set listed in Appendix Table S3; see Methods for description). Second, we used a cytokine array to identify proteins increased/decreased in EMT6-PTR cells adapted to cell culture (Fig. 2C, left panel; Appendix Fig. S1 shows array exposures; Appendix Fig. S2 shows array layout). Third, using the same protein array, we compared variants of EMT6-P and -PTR cells that had IFNAR1 knocked down (IFNAR1$^{KD}$) so we could identify array proteins regulated by type I IFN signaling (Fig. 2C, middle panel; PTIS right panel; Fig. EV1A shows IFNAR1$^{KD}$ cell validation). Fourth, using the Interferome database, we identified 54 IFN-regulated proteins on the array that increased in EMT6-PTR cells (Fig. 2D, left panel;

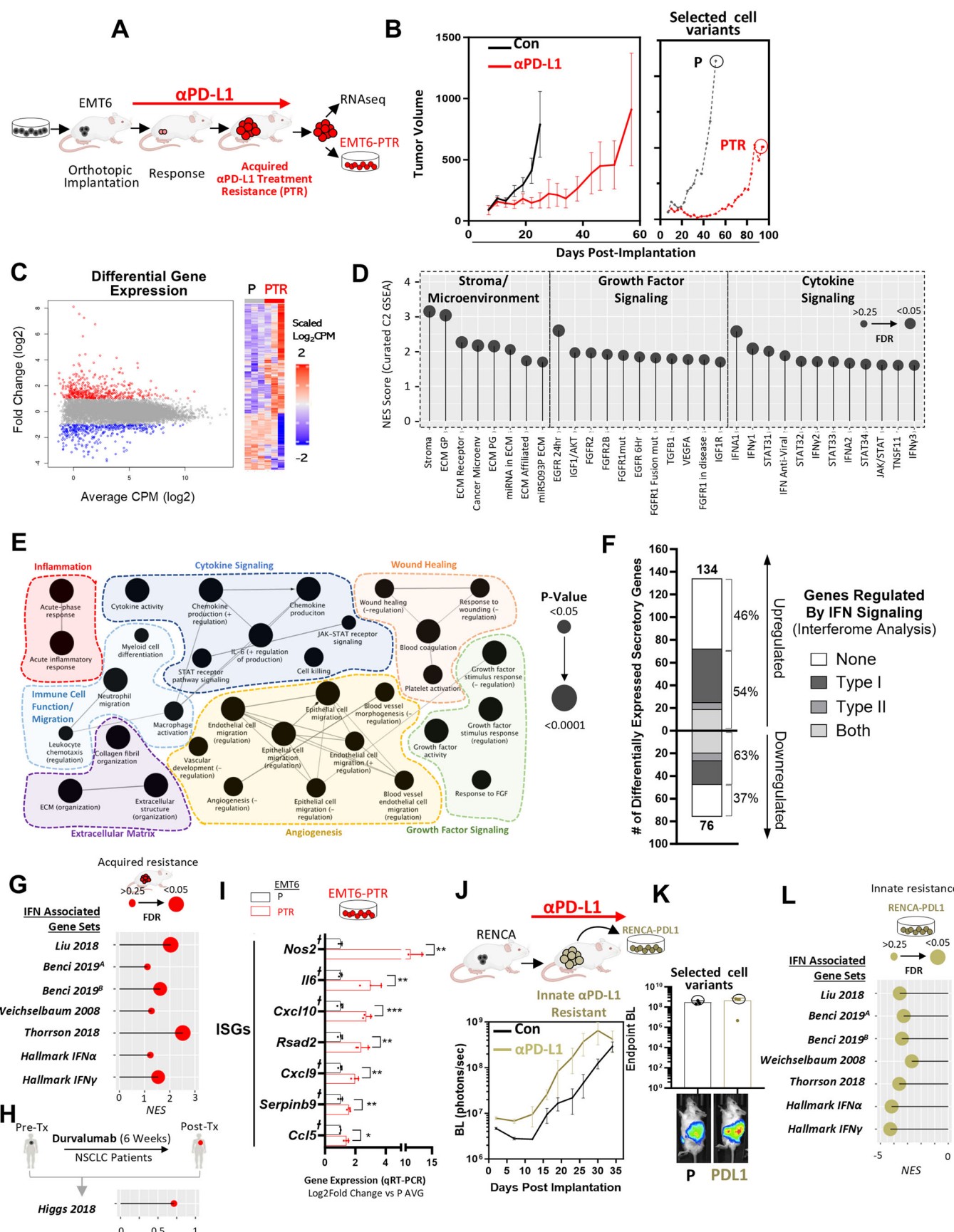

**Figure 1.  Acquired αPD-L1 treatment resistance (PTR) increases secretory profiles enriched for type I IFN regulated genes.**

(A) Schematic showing orthotopic breast EMT6 model of acquired resistance following PD-L1 inhibition. (B) Continuous αPD-L1 treatment in BALB/c mice ($n = 3$–4 biological replicates) bearing orthotopically-implanted mouse mammary EMT6 cells (left) with tumor growth of selected P and PTR variants shown (right; circle indicates selection). PTR and P variants were maintained in vitro with respective αPD-L1 or IgG antibody (see Methods for details). (C–F) RNA sequencing analysis of EMT6-P and EMT6-PTR tumor tissues. $n = 3$. (C) Differentially expressed genes ($Log_2$ [Fold Change] $\leq -2$ or $\geq 2$) in EMT6-PTR as summarized by dot plot (left) and heatmap (right). Red = upregulated; blue = downregulated. (D) Summary of stroma/tumor microenvironment, growth factor signaling, and cytokine signaling gene sets with significant positive enrichment (FDR ≤ 0.25) found in PTR tumors via GSEA of all canonical pathways (C2, Molecular Signatures Database Collection). Size of circles correspond to GSEA calculated FDR significance. See Appendix Table S1 for details. (E) Cytoscape GO analysis of significantly enriched biological processes in upregulated secretory genes, grouped by signaling categories. Size of circles correspond to $p$ value significance of each process calculated via Fisher's exact test and lines represent term-term interactions defined by Kappa score. (F) Bar graph representing Interferome Database secretory genes up- and down-regulated in EMT6-PTR (compared to EMT6-P). (G, H) GSEA of EMT6-PTR tumors (compared to P controls) showing lollipop plot representing (G) NES of published/Hallmark IFN gene sets, and (H), NES of an IFN-specific gene-set identified in αPD-L1-treated (durvalumab) NSCLC patients (described in Ref (Higgs et al, 2018)). Size of circles correspond to FDR significance calculated via GSEA. See Appendix Table S2 for details. (I) Type I IFN-regulated ISGs in EMT6-PTR selected cell variants (qRTPCR). ┼ represent genes associated with secretory proteins. qRT-PCR, statistics performed via two-tailed t-test, $n = 3$–4. (J) Schematic showing orthotopic mouse kidney RENCA tumor model of innate resistance to αPD-L1 inhibition (top) and the BLI quantification in mice treated with αPD-L1 and IgG vehicle control (bottom; Balb/c mice; $n = 3$ biological replicates). (K) BLI of mice at endpoint (top) with representative images of mice shown (bottom). Circles indicate mice used to select P and PTR tumor variants which were selected from dissociated kidneys and maintained in vitro with IgG or αPD-L1 antibody, respectively (see Methods for details). (L) GSEA of RENCA-PDL1 tumor cells (compared to P controls) showing lollipop plot representing NES of published/Hallmark IFN gene sets. Size of circles correspond to FDR significance calculated via GSEA. See Appendix Table S2 for details. Data Information: Parental (P); αPD-L1 Treatment-Resistant (PTR); Control (Con); Gene Set Enrichment Analysis (GSEA); Gene Ontology (GO); interferon stimulated genes (ISGs); Counts per million (CPM); Bioluminescence imaging (BLI); normalized enrichment scores (NES); Non-small cell lung carcinoma (NSCLC); Standard Deviation (SD); Standard Error of the Mean (SEM). αPD-L1 (clone 80) and IgG/PBS were administered at 250 µg/mouse every 3 days until endpoint. Primary tumor burden was assessed by caliper measurement. Tumor growth line graph quantitative data represent mean ± SEM. Bar graphs show mean ± SD. Time to institutional endpoint was assessed by Kaplan–Meier. *$p \leq 0.05$, **$p \leq 0.01$, ***$p \leq 0.001$, ****$p \leq 0.0001$, for exact $p$ values see Fig. 1 Source data. All replicates shown represent technical replicates unless otherwise specified. Source data are available online for this figure.

Fig. EV1B shows IFN-regulated proteins) and then we identified 35/54 that decreased in EMT6-PTR IFNAR1$^{KD}$ variants (Fig. 2D, right panel; Fig. EV1C shows data summary). Fifth, we undertook separate assays to confirm/select individual PTIS candidates that increased in EMT6-PTR and decreased in EMT6-PTR-IFNAR1$^{KD}$. This included western blot tests for LCN2 and NOS2 (Fig. 2E; densitometry shown in Fig. 2F; Appendix Fig. S3 shows uncropped replicate blots), ELISA tests for LCN2, IL6, and CCL2 (Fig. 2G), and qRT-PCR for all (Fig. 2H,I). Increases in all PTIS factors were observed along with decreases following IFNAR1$^{KD}$ in PTR cells (Fig. 2J shows data summary). Together, these results identify a unique type I IFN-regulated secretory signature associated with acquired resistance to αPD-L1 treatment that could be confirmed by at least one transcriptomic and one proteomic method (Fig. EV1D).

## The PTIS is enriched in αPD-L1 treatment-sensitive tumors

To validate whether the PTIS signature was associated with αPD-L1 treatment-sensitive tumors, we tested for PTIS enrichment in publicly available NCBI GEO and dbGaP whole transcriptome datasets from published preclinical and clinical studies involving αPD-L1-treated tumors. We used 5 published preclinical datasets that contained 3 studies reported as αPD-L1 treatment-*sensitive* (Lan et al, 2018; Sceneay et al, 2019; Efremova et al, 2018) (Data ref: Lan et al, 2018; Data ref: Sceneay et al, 2019; Data ref: Efremova et al, 2018) and 2 studies reported as αPD-L1 treatment-*insensitive* (Sceneay et al, 2019) (Data ref: Sceneay et al, 2019) and RENCA-PDL1 tumor cells from this study) (see Methods). We found that PTIS signature expression was increased in all αPD-L1 treatment-*sensitive* models as defined by average counts per million (CPM) levels, with 3 of 3 models demonstrating significant positive GSEA enrichment (FDR ≤ 0.25) and 1 of 3 models showing significance by both CPM expression and GSEA enrichment (Fig. 3A). Conversely,

PTIS signature expression was decreased in αPD-L1 treatment-*insensitive* models (Fig. 3B; Fig. 3C shows data summary). We next examined 2 clinical studies which contained transcriptomic data from tumor biopsies of NSCLC (Gettinger et al, 2017) (Data ref: Gettinger et al, 2017) or Merkel cell carcinoma (MCC) (Paulson et al, 2018) (Data ref: Paulson et al, 2018) taken from patients described as initially αPD-L1 treatment-sensitive (see Methods). In the NSCLC samples, PTIS signature expression increased with significant positive GSEA enrichment in bulk RNAseq data from patients who developed acquired resistance (Fig. 3D). In MCC samples, single-cell RNAseq (scRNAseq) datasets also showed increased PTIS signature expression in tumors following PD-L1 inhibitor treatment (Fig. 3E, left panel). Notably, in these same patients, scRNAseq clustered analysis showed PTIS enrichment in macrophage and T cell compartments suggesting tumor-extrinsic secretory changes in 'host' cell populations can also be altered by treatment (Fig. 3E, right panel). In separate studies, we generated a PTIS using only genes downregulated in EMT6-PTR cells (termed 'PTIS$^{DOWN}$') and found dataset validations were not consistent, suggesting upregulated PTIS genes are more representative of acquired resistance (described in Fig. EV2A–E). Together, these findings demonstrate that the IFN pathway associated PTIS is enriched in multiple preclinical/clinical tumors initially *sensitive* to αPD-L1 treatment including the setting of acquired resistance and can occur independent of cancer type.

## IFNs modulate PTIS, STAT signaling, and surface ISGs in PTR cells

Since IFNα/IFNβ are known to bind to IFNAR (Schoggins, 2019) and drive ISG expression, we next examined whether activation of type I IFN signaling could further modulate PTIS expression. First, we tested endogenous/intrinsic IFNα/IFNβ levels in EMT6-P and -PTR variants. IFNα was not detectable at the transcript level and, while IFNβ transcript levels did increase in PTR cells, these values

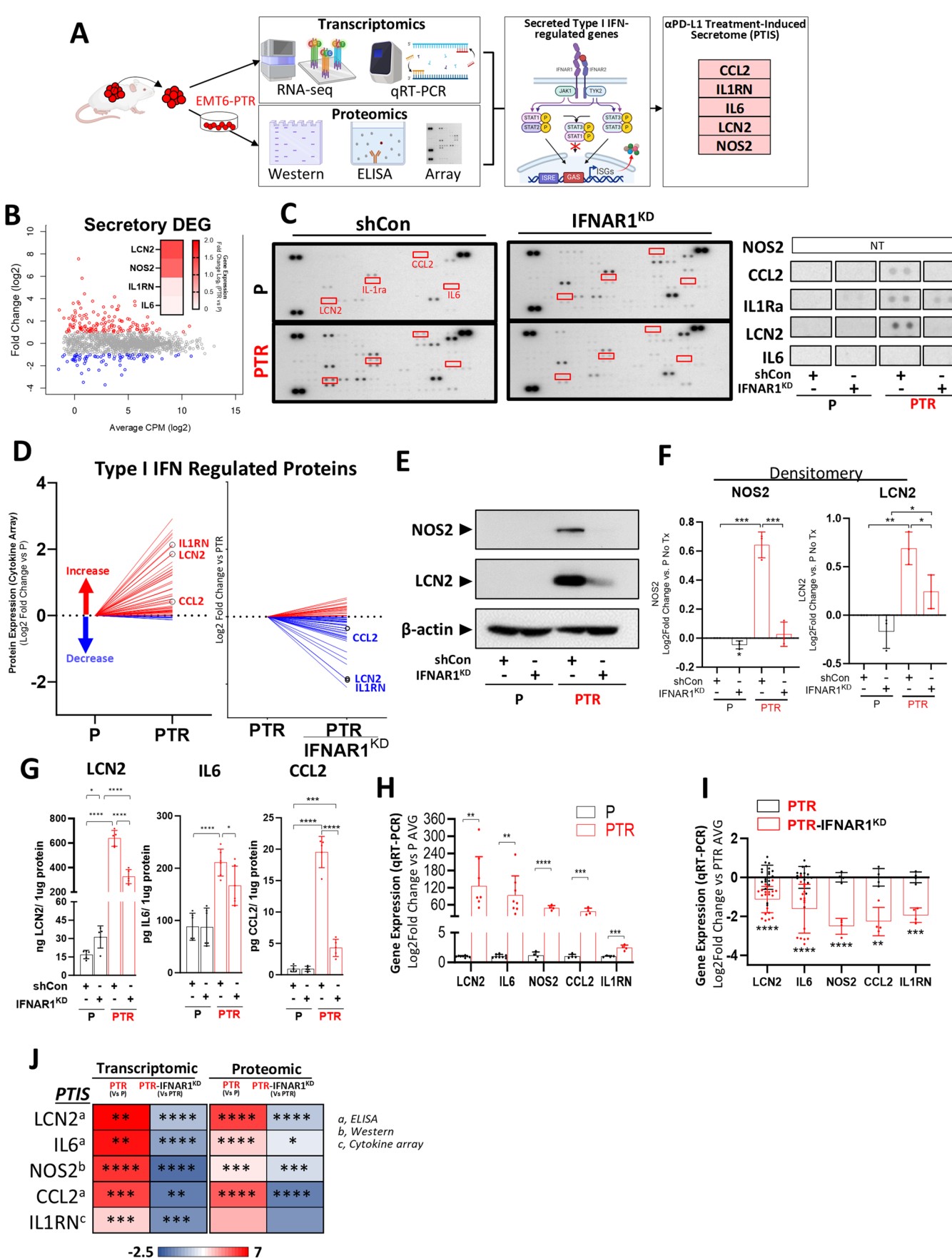

Figure 2.   Identification of an αPD-L1 treatment-induced secretome (PTIS) gene signature.

(A) Generation of a PTIS signature comprised of 5 upregulated genes identified from transcriptome and proteomic analysis using EMT6-PTR cells. (B) Differentially expressed secretory genes (GO:0005576) (Log$_2$ [Fold Change] ≤ −2 or ≥2) in EMT6-PTR cells as summarized by dot plot (left) and heatmap (right). Red = upregulated; blue = downregulated. Embedded heatmap showing gene expression of PTIS factors. $n = 3$. (C) Blot images of cytokine protein array of EMT6-P and -PTR tumor cells before and after IFNAR1 knockdown. PTIS factors highlighted in red and shown as enlarged images (right). $n = 1$. (D) Cytokine protein array analysis of type I IFN regulated proteins in EMT6-PTR tumor cells compared to parental controls (left) and EMT6-PTR compared to EMT6-PTR IFNAR1$^{KD}$ tumor cells. Circled proteins showing expression of PTIS factors. $n = 1$. (E) NOS2 and LCN2 levels in lysates of EMT6-P, EMT6-PTR, EMT6-P IFNAR1$^{KD}$, and EMT6-PTR IFNAR1$^{KD}$ cells. Western blot, representative image $n = 1$. (F) Densitometry quantification of western blots shown in (E) representing LCN2 (left) and NOS2 (right) levels. $n = 3$, statistics performed via two-tailed t-test. (G) LCN2, IL6, and CCL2 levels in lysates of EMT6-P, EMT6-PTR, EMT6-P IFNAR1$^{KD}$, and EMT6-PTR IFNAR1$^{KD}$ cells. ELISA, $n = 5$–6, statistics performed via two-tailed t-test. (H) Summary data of PTIS factor expression in EMT6-P and -PTR cells shown as relative to P controls and represented as bar graphs. qRT-PCR, $n = 4$–8 (1–2 biological replicates), statistics performed via two-tailed t-test. (I) Summary data of PTIS factor expression in EMT6-PTR-IFNAR1$^{KD}$ cells shown as relative to PTR controls and represented as bar graphs. qRT-PCR, $n = 4$–20 (1–4 biological replicates), statistics performed via two-tailed t-test. (J) Heatmap summary of baseline transcriptomic and proteomic expression of PTIS factors showing comparisons of EMT6-P/PTR and IFNAR1$^{KD}$ cells, statistics performed via two-tailed t-test. Data Information: αPD-L1 Treatment-Induced Secretome (PTIS); αPD-L1 Treatment-Resistant (PTR); Counts per million (CPM); IFNAR1 knockdown (IFNAR1$^{KD}$); Control Knockdown (shCon); Not Tested (NT); Standard Deviation (SD). Bar graphs show mean ± SD. *$p ≤ 0.05$, **$p ≤ 0.01$, ***$p ≤ 0.001$, ****$p ≤ 0.0001$, for exact $p$ values see Fig. 2 Source data. All replicates shown represent technical replicates unless otherwise specified. Source data are available online for this figure.

were low and highly variable, and no protein could be detected in lysates (Fig. 4A; protein IFNβ shown in Appendix Fig. S4). Together this suggests that PTIS and ISG expression in PTR cells is not driven by an autocrine IFNAR1 activation. Next, we added exogenous IFNα/IFNβ and found that, after 48 h stimulation, PTIS expression increased in EMT6-PTR cells (Fig. 4B shows relative PTIS compared to parental controls; Fig. EV3A shows full results), which then reversed in EMT6-PTR-IFNAR1$^{KD}$ cells (Fig. 4C shows relative PTIS levels to PTR controls; significance not reached for LCN2 compared to shRNA controls). Interestingly, we found the *magnitude* of IFN-modulated PTIS expression in EMT6-PTR variants to be significantly enhanced for several factors after stimulation compared to untreated controls, including IL6 and NOS2 (Fig. 4D; Fig. EV3B shows statistical summary for Fig. 4B–D). For IL6, enhanced expression was confirmed in conditioned media (CM) after IFNα/β treatment in EMT6-PTR cells (Fig. 4E). Notably, this IFN-enhanced effect extended beyond the PTIS, as other ISGs shown in Fig. 1I also were increased in PTR cells after IFNα/IFNβ stimulation (Fig. EV3C). Control experiments to test whether known anti-proliferative effects of IFN stimulation (Bekisz et al, 2010) might be impacting PTIS expression were ruled out as EMT6-P and PTR cells did not consistently respond differently to IFN exposure (Fig. EV3D). Overall, these results demonstrate that exogenous IFNs may further stimulate key PTIS genes, which are already increased in PTR cells. Interestingly, type II IFNγ could also stimulate PTIS increases, suggesting PTR cells have a broad sensitivity to the IFN ligand family (Appendix Fig S5). To further connect the PTIS and IFN-regulated programs, we next examined intracellular STAT proteins as these are known to be activated by type I IFN signaling (Fig. 4F; blotting replicates in Appendix Fig. S6). We found that relative phosphorylation of STAT3 was significantly increased in EMT6-PTR cells while relative phosphorylation of STAT1 remained unchanged (Fig. 4G) and total STAT1 levels decreased (Fig. EV3E). Enhanced pSTAT3 activation could be partially reversed in EMT6-PTR-IFNAR1$^{KD}$ cells (Fig. 4F) though this did not reach significance. Notably, both pSTAT1 and pSTAT3 were more significantly activated following IFNβ stimulation in EMT6-PTR cells (Fig. 4H; Fig. EV3F shows total protein levels; Fig. EV3G shows confirmation via pSTAT3 ELISA), suggesting an increased sensitivity to stimulation after resistance. Importantly, since IL6 can activate STAT3 signaling, we sought to rule out an autocrine IL6 mechanism to explain pSTAT3 increases

in EMT6-PTR cells. To test this, we treated EMT6-P and -PTR cells with a mouse anti-IL6 (αIL6) antibody and found no changes to the results observed in 4F-H (Appendix Fig. S7; Appendix Fig. S8 shows densitometry analysis). Taken together, these results suggest intracellular pSTAT3 signaling is uniquely enhanced in EMT6-PTR cells and a potential driver of PTIS. Next, since ISGs found in the PTIS represent secreted factors that have diverse and, at times, opposing effects on tumor development and growth (Cheon et al, 2014; Schneider et al, 2014), we sought to examine additional extracellular (non-secreted) ISGs representing various antigen presentation genes and immune-regulating surface proteins. This included immune-stimulatory molecules such as TAP1 (transporter protein supporting antigen processing), H2D1/B2M (MHC class I molecules), and MHC-I (Bander et al, 1997), as well as immune-suppressive molecules such as PD-L1 (Garcia-Diaz et al, 2019). Interestingly, TAP1/H2D1/B2M genes and MHC-I/PD-L1 proteins were all found to be decreased in EMT6-PTR cells (Fig. 4I), and were not as robustly increased when stimulated with IFNβ stimulation compared to P controls (Fig. 4J, only MHC-I/PD-L1 shown; Fig. EV3H shows full results). Overall, these results show acquired resistance to inhibition of PD-L1 rewires the IFN-regulated secretory machinery in tumor cells, an effect that can be enhanced by IFN stimulation and include immune-modulating cellular ISG expression (Fig. 4K summary; Fig. EV3I shows statistics summary).

## Intracellular PD-L1 signaling only partially regulates PTIS expression

Since PD-L1 has intrinsic cell signaling functions that can be regulated by IFN signaling (Garcia-Diaz et al, 2019; Gato-Canas et al, 2017), it is possible that PTIS in tumor cells after acquired resistance may stem directly from PD-L1 inhibition. To test this, we targeted PD-L1 genetically and therapeutically ex vivo and assessed PTIS expression. First, we assessed exposure of the αPD-L1 antibody (Clone 80) or IgG on EMT6 cells in vitro for >4 weeks, generating EMT6-PTR$^{VITRO}$/P$^{VITRO}$ variants (Fig. 5A, top). Proteomic analysis of EMT6-PTR$^{VITRO}$ cells via a cytokine antibody array demonstrated increased expression of PTIS factors (Fig. 5A, bottom; Fig. 5B; Appendix Fig. S9 shows array exposures) and an increase in the majority of type I IFN regulated factors (Fig. 5C). Notably, several PTIS and type I IFN regulated protein increases

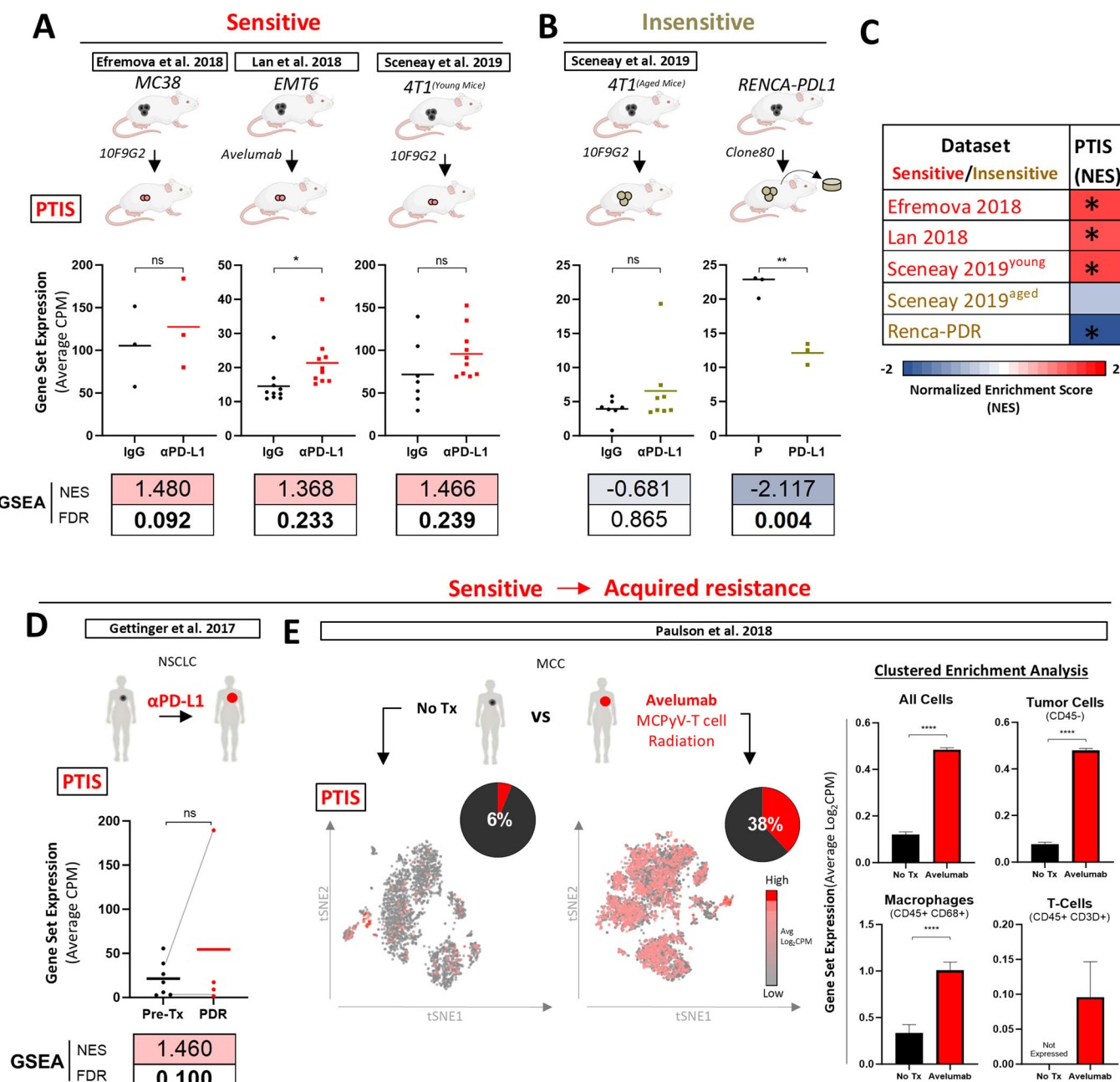

**Figure 3. Clinical and preclinical validation of PTIS in αPD-L1 treatment-sensitive models.**

(A–C) PTIS expression using average CPM expression and GSEA in published bulk RNAseq datasets from preclinical studies involving PD-L1 treatment-*sensitive* and -*insensitive* tumor models. FDR calculated via GSEA. (A) PTIS expression in αPD-L1 treatment-*sensitive* tumor models included GEO: GSE130472, GSE93017, GSE107801. Data is compared to vehicle/IgG-treated controls, *n* = 3–10 biological replicates. (B) PTIS expression in αPD-L1 treatment-*insensitive* tumor models included GEO: GSE130472 and RENCA-PDL1 model from this study, *n* = 3–8 biological replicates. (C) Heatmap summary of GSEA analysis of preclinical studies in (A) and (B). * indicate FDR < 0.25. (D, E) PTIS expression using average CPM expression and GSEA in published bulk and single cell RNAseq datasets taken from tumor biopsies of αPD-L1 treatment-*sensitive* patients. (D) NSCLC patients (dbGaP # phs001464.v1.p1): bulk RNAseq from Pre- and Post-treatment (Tx) tumor sample comparisons (Gray lines indicate matched Pre- and Post-Tx samples), *n* = 4–7 biological replicates. (E) MCC patients (GEO: GSE118056): single-cell RNAseq from untreated (No-Tx) or treated (avelumab, MCPyV-T cell, radiation) tumor samples (*n* = 1) with tSNE plots (left) representing average log2 CPM expression of PTIS in whole dataset, and bar graphs (right) representing clustered enrichment analysis populations identified by markers for tumors (CD45−), macrophages (CD68+), and T cells (CD3D+). Tumor sample that received No-Tx was compared to treated. Statistics performed via two-tailed t-test. Data Information: αPD-L1 Treatment-Induced Secretome (PTIS); αPD-L1 Treatment-Resistant (PTR); Counts per million (CPM); Gene set enrichment analysis (GSEA); False Discovery Rate (FDR); Gene Expression Omnibus (GEO); GEO Series records (GSE); database of Genotypes and Phenotypes (dbGaP); t-distributed stochastic neighbor embedding (tSNE); Treatment (Tx); non-small cell lung carcinoma (NSCLC); Merkel cell carcinoma (MCC). Bar graphs show mean ± SEM. Scatter dot plot central line shows mean. Bolded numbers for GSEA represent FDR < 0.25 (see Methods). *p ≤ 0.05, **p ≤ 0.01, ***p ≤ 0.001, ****p ≤ 0.0001, except for GSEA as indicated above, for exact p values see Fig. 3 Source data. Source data are available online for this figure.

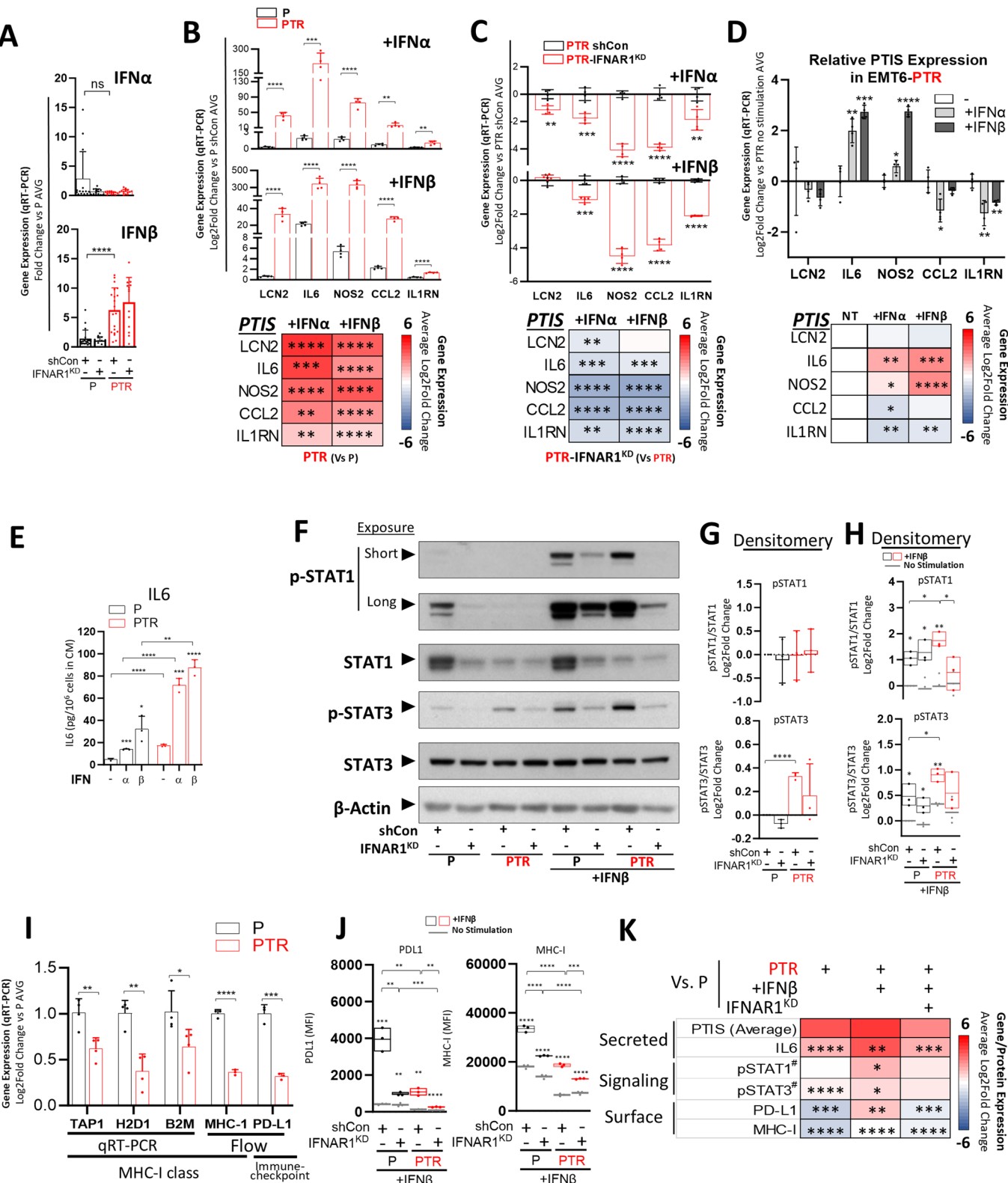

were similar in the in vivo-derived EMT6-PTR and EMT6-PTR^VITRO cells (Fig. EV4A and EV4B, respectively). Next, we performed whole transcriptomic analysis in EMT6-PTR^VITRO cells and found significant positive enrichment of PTIS via GSEA both at

baseline and after IFNβ stimulation (Fig. 5D; Fig. 5E shows data summary). However, unlike the in vivo-derived EMT6-PTR cells, assessment of PTIS factors on an individual level did not demonstrate enhancement in EMT6-PTR^VITRO cells both at baseline

◄ **Figure 4. Type I IFNs modulate PTIS, STAT signaling, and surface ISGs in PTR cells.**

(A) IFNα and IFNβ expression in EMT6-P and -PTR before and after knockdown of IFNAR1[KD]. qRT-PCR, $n = 14$–20 (3–4 biological replicates), statistics performed via two-tailed t-test. (B) PTIS expression in EMT6-P and -PTR cells shown as relative to non-stimulated P controls after IFNα/β stimulation and represented as bar graph (top) and heatmap (bottom). qRT-PCR, $n = 4$, statistics performed via two-tailed t-test. (C) PTIS factor expression in EMT6-PTR-IFNAR1[KD] cells shown as relative to PTR controls after IFNα/β stimulation and represented as bar graph (top) and heatmap (bottom). qRT-PCR, $n = 4$, statistics performed via two-tailed t-test. (D) PTIS factor expression in EMT6-PTR cells after IFNα/β stimulation shown as relative to non-stimulated PTR and represented as bar graph (top) and heatmap (bottom). qRT-PCR, $n = 4$, statistics performed via two-tailed t-test. (E) IL6 expression in EMT6-P and -PTR cell conditioned media. ELISA, $n = 3$, statistics performed via two-tailed t-test. (F) Phosphorylated and total levels of STAT1/3 in lysates of EMT6-P and -PTR before and after knockdown of IFNAR1[KD] following IFNβ stimulation. Western Blot. (G, H) Densitometry quantification of western blots shown in (F) representing relative phosphorylated STAT1 compared to total STAT1 (top) and phosphorylated STAT3 levels compared to total STAT3 levels (bottom) at (G) baseline and (H) after IFNβ stimulation. $n = 3$, statistics performed via two-tailed t-test. (I) Antigen presentation machinery expression in EMT6-P and -PTR cells. qRT-PCR and flow cytometry as indicated $n = 3$–4, statistics performed via two-tailed t-test. (J) PD-L1 and MHC-I expression in EMT6-P and -PTR before and after knockdown of IFNAR1[KD], and after IFNβ stimulation. ELISA lysate and flow cytometry, respectively, $n = 3$, statistics performed via two-tailed t-test. (K) Heatmap summary of protein and gene expression analysis. Statistics comparing PTR versus P for each condition, statistics performed via two-tailed t-test. Data Information: Parental (P); αPD-L1 Treatment-Resistant (PTR); Conditioned Media (CM); Not Treated (NT); Mean Fluorescent Intensity (MFI); IFN stimulated genes (ISGs); IFNAR1 knockdown (IFNAR1[KD]); shRNA vector control (shCon; shown here as a '–'); Cells were treated with 10 ng/ml of IFNs and collected after 15 min (western blot shown in (F)), and 5 days (IL6, PD-L1, MHC-I). Bar graphs show mean ± SD. Box plot indicate range between minimum and maximum, central line depicts the mean. *$p \leq 0.05$, **$p \leq 0.01$, ***$p \leq 0.001$, ****$p \leq 0.0001$ indicate significance compared to untreated controls unless otherwise shown (lines), for exact p values see Fig. 4 Source data. #STAT activation measured by comparing phosphorylated STAT protein compared to total STAT protein levels. All replicates shown represent technical replicates unless otherwise specified. Source data are available online for this figure.

and after IFNβ stimulation (Fig. 5F shows relative values, Fig. EV4C shows full data). Interestingly, several ISGs increased in EMT6-PTR such as CXCL9, CXCL10, and SerpinB9 from Fig. 1I were also increased in EMT6-PTR[VITRO] cells, indicating that antibody exposure alone can, to an extent, modulate IFN signaling (Fig. EV4C). For further confirmation, a separate cell line (CT26, a mouse colorectal cell) and separate αPD-L1 antibody (MIH5) were used to generate CT26-PTR[VITRO]/P[VITRO] cells where only select PTIS factors (CCL2 and IL1RN) were found to be enhanced compared to P controls and further enhanced after IFNβ stimulation (Fig. 5G; Fig. EV4D shows full data). Next, we assessed PTIS expression in EMT6 cells with PD-L1 knocked down (EMT6-PDL1[KD]) where only IL6 was found to have increased expression in EMT6-PDL1[KD] cells at baseline and not after IFNβ stimulation (Fig. 5H; Fig. EV4E shows cell generation; Fig. EV4F shows full data). Again, several ISGs increased in EMT6-PTR such as CXCL9, CXCL10, and SerpinB9 from Fig. 1I were also increased in EMT6-PDL1[KD] cells, suggesting that intracellular PD-L1 signaling can modulate ISG production (Fig. EV4F). Together, these results reveal that intrinsic PD-L1 signaling can account for some of the IFN-regulated PTIS observed in in vivo-derived resistance models, but these secretory changes were partial and not consistent (Fig. 5I,J show PTIS summary: Fig. EV4G show statistics summary for all in vitro studies) indicating that tumor:immune cell interactions play a key role in PTIS adaptations in treatment-sensitive tumors.

## Immune-suppression and -stimulation by PTR cells is IFN-regulated

Since secreted PTIS cytokines such as IL6, LCN2, CCL2 and non-secreted ISGs (i.e., MHC-I and PD-L1) can have opposing effects on immune activation and affect how tumor cells respond to ICI treatments (Adler et al, 2023; Tu et al, 2020), we examined whether PTR cells might influence (or be influenced by) immune cell populations that are part of the anti-tumor response. To test this, we first performed CIBERSORT/ImmuCC tissue deconvolution analysis to identify immune cell populations in PTR tumors using mouse-specific gene scores in RNAseq data (described in (Chen et al, 2017; Chen et al, 2018b)). EMT6-PTR tumors had higher total

immune scores (Fig. 6A), with activated cytotoxic CD8 + T lymphocyte (CTL) and M2 macrophage scores significantly decreased (Fig. 6B), further suggesting conflicting tumor immune responses may occur after acquired PD-L1 resistance. To examine this directly, we next tested EMT6-P/PTR cells for cell cytotoxicity in co-culture studies with dissociated mouse splenocytes containing αCD3/αCD28 activated CD8 + T cells measured by flow cytometry (Fig. 6C, representative images shown). We found tumor markers for apoptosis (annexin V) and cell death (7-AAD) to be significantly decreased in EMT6-PTR cells compared to P controls (Fig. 6D), indicating that PTR cells have an underlying immune-protective effect. Notably, this protection significantly decreased in PTR cells when IFNAR1 expression was knocked down (Fig. 6E, relative comparisons shown). To test whether secretory factors released from PTR cells could influence CTL functions, we next measured CD8 + T-cell proliferation (Quah and Parish, 2010), CD69 (an early T-cell activation marker (Lindsey et al, 2007)), and the T-cell effector proteins Granzyme B (GZMB) and IFNγ which are known to mediate tumor cell-killing (Bhat et al, 2017; Medema et al, 2001) (Figs. 6F and EV5A show schematics). CM from EMT6-PTR cells significantly decreased T-cell proliferation (Fig. 6G, shown as % divided), CD69 (Fig. 6H), IFNγ (Fig. EV5B), and GZMB (Fig. EV5C). Interestingly, these decreases could be significantly weakened (reversed) when CM from EMT6-PTR-IFNAR1[KD] cells were used in the same experimental setup, suggesting that the secretory products in PTR cells have an overall immune-suppressive effect that is IFN-regulated (Figs. 6G,H and EV5B–D). Next, we conducted identical experiments but, rather than using CM, we combined tumor and splenocyte cells together in co-culture, with the goal to simultaneously examine both contact-*independent* (secretory) and contact-*dependent* cellular interactions (Figs. 6I and EV5E show schematics). Our results show that, instead of the decreased immune cell activity seen with CM, co-culture produced significant increases in T-cell proliferation (Fig. 6J) and CD69 (Fig. 6K), or no changes in IFNγ (Fig. EV5F) and GZMB (Fig. EV5G). This indicates that contact-dependent tumor:immune cell effects can offset (and even reverse) contact-independent immune-suppression. Identical co-culture studies performed with EMT6-PTR-IFNAR1[KD] cells significantly

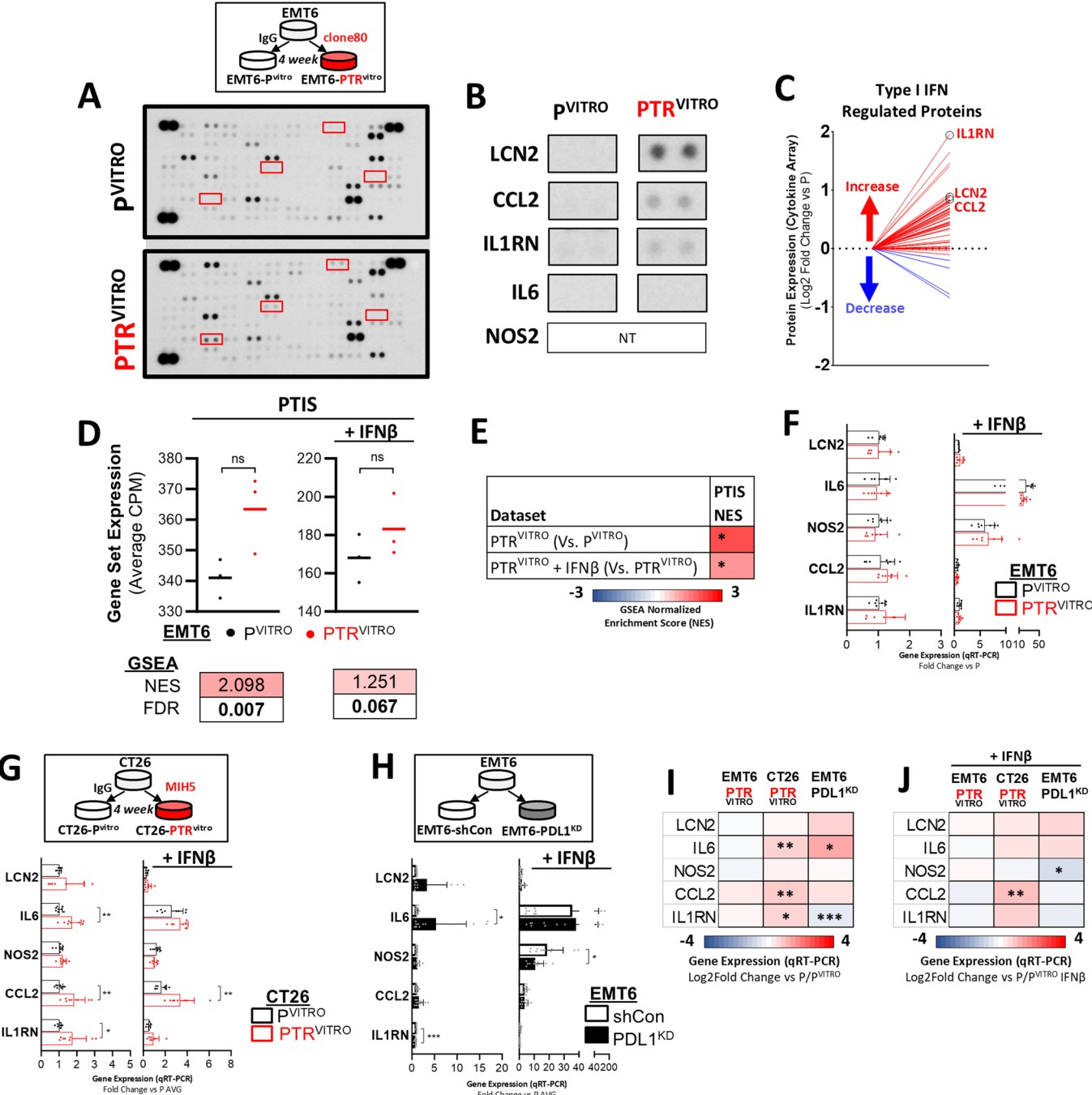

boosted some (but not all) immune-cell activation markers even further with T-cell proliferation and CD69 expression increasing (Fig. 6J,K), while GZMB or IFNγ remained unchanged (Fig. EV5F–H). A data summary of contact-independent (secretory) and contact-dependent co-culture studies is detailed in Fig. 6L. Together, these results show acquired resistance to PD-L1 inhibition 'rewires' tumoral IFN-signaling to produce immune-*suppressive* secretory changes and apoptosis-limiting stimuli; while simultaneously producing an opposing immune-*stimulatory* effect via contact-dependent processes (Fig. 6M shows schematic).

## Inhibition of PTIS regulators selectively inhibits PTR tumor growth

To examine whether tumor growth may be altered by these opposing immune-modulating effects in vivo, we evaluated the impact of PTIS blockade *indirectly* by disrupting IFN-regulation or *directly* by blocking one secretory factor, such as IL6. To do this, we first implanted EMT6-P and EMT6-PTR cells orthotopically in BALB/c mice and found tumors grew at similar rates, suggesting immune-suppressive/promoting functions in PTR cells may offset to yield minimal changes in tumor growth kinetics. However, when

◀ **Figure 5. Intrinsic PD-L1 signaling partially regulates PTIS.**

(A) Schematic showing generation of EMT6-PTR$^{VITRO}$ cell variants following αPD-L1 treatment in vitro for >4 weeks (top box). Blot images of Cytokine protein array of EMT6-P$^{VITRO}$ and -PTR$^{VITRO}$ tumor cells with PTIS factors highlighted in red (bottom). $n = 1$. (B) Enlarged images of array shown in (A). $n = 1$. (C) Line graph summary of cytokine protein array showing type I IFN regulated proteins in EMT6-PTR$^{VITRO}$ tumor cells compared to EMT6–P$^{VITRO}$ shRNA controls. Circled proteins showing expression of PTIS factors. $n = 1$. (D) RNA sequencing analysis of EMT6 PTR$^{VITRO}$ cells before and after IFNβ stimulation shown as average CPM expression (top) and GSEA (bottom). $n = 3$, statistics performed via two-tailed t-test, bolded numbers indicate FDR < 0.25. (E) Heatmap summary of GSEA analysis of EMT6 PTR$^{VITRO}$ cells in (5D). $n = 3$, FDR calculated via GSEA and indicated by (*). (F) PTIS factor expression in EMT6-P$^{VITRO}$ and -PTR$^{VITRO}$ cells shown as relative to P$^{VITRO}$ controls before and after IFNβ stimulation represented as bar graphs. qRT-PCR, $n = 7$ (2 biological replicates). (G) Schematic showing generation of CT26-PTR$^{VITRO}$ cell variants following αPD-L1 treatment in vitro for >4 weeks (top). PTIS factor expression in CT26-P$^{VITRO}$ and -PTR$^{VITRO}$ cells shown as relative to P$^{VITRO}$ controls before and after IFNβ stimulation represented as bar graphs (bottom). qRT-PCR, $n = 8$ (2 biological replicates), statistics performed via two-tailed t-test. (H) Schematic showing generation of PDL1$^{KD}$ cell variants following αPD-L1 treatment in vitro for >4 weeks (top). PTIS factor expression in EMT6-shCon and -PDL1$^{KD}$ cells shown as relative to shCon before and after IFNβ stimulation represented as bar graphs (bottom). qRT-PCR, $n = 11$ (3 biological replicates), statistics performed via two-tailed t-test. (I, J) Heatmap summary of PTIS expression at baseline (I) and after IFNβ stimulation (J) in (F–H). Statistics performed via two-tailed t-test. Data Information: Parental (P); αPD-L1 Treatment-Resistant (PTR); IFNAR1 knockdown (IFNAR1$^{KD}$); shRNA Vector Control (shCon); in vitro-derived Parental (P$^{VITRO}$); in vitro-derived PD-L1 Treatment Resistant (PTR$^{VITRO}$); Gene Set Enrichment Analysis (GSEA); αPD-L1 Treatment-Induced Secretome (PTIS); normalized enrichment scores (NES); false discovery rate (FDR); Conditioned media (CM); PD-L1 knockdown (PDL1-KD). Cells were treated with 10 ng/ml of IFNs and collected after 5 days for IL6 protein expression quantifications. Bar graphs show mean ± SD. Scatter dot plot central line shows mean. *$p \leq 0.05$, **$p \leq 0.01$, ***$p \leq 0.001$, ****$p \leq 0.0001$ except for GSEA as indicated above, for exact p values see Fig. 5 Source data. All replicates shown represent technical replicates unless otherwise specified. Source data are available online for this figure.

type I IFN signaling was disrupted in the same PTR cells, we observed markedly opposing effects on tumor growth (Fig. 7A, area under the curve (AUC) analysis shown in inset). IFNAR1$^{KD}$ in EMT6-P tumors significantly *increased* tumor growth, whereas IFNAR1$^{KD}$ in EMT6-PTR tumors significantly *decreased* tumor growth, compared to respective controls (Fig. 7B, AUC analysis shown in inset). While growth promotion by type I IFN signaling blockade in treatment-naïve tumors has been reported previously (Katlinski et al, 2017; Lu et al, 2019a), our results suggest PTR tumors have a unique vulnerability to IFN-signaling, perhaps the result of reducing the immune protective effect of the PTIS. To examine this further, mouse αIL6 antibody was administered to BALB/c mice bearing orthotopically-grown EMT6-P and EMT6-PTR tumors. PTR tumor variants treated with αIL6 showed significant tumor inhibition compared to P controls, but this inhibition was more muted than the more generalized PTIS blockade observed when IFN-signaling was disrupted (Fig. 7C, AUC analysis shown in inset; Fig. 7D, comparisons shown to respective controls). Interestingly, PTIS-related vulnerabilities in PTR tumor growth were found to extend to metastasis with post-mortem analysis at experiment endpoint showing enhanced inhibition of tumor skin/abdominal wall invasion in PTR tumor variants after IFNAR1$^{KD}$ and IL6 inhibition compared to P controls (Fig. 7E). Taken together, these results show that targeting PTIS has minimal or even growth promoting effects in untreated tumors yet, after acquired resistance to PD-L1 inhibition, tumors have a unique IFN-regulated vulnerability linked to PTIS expression that, when targeted directly or indirectly, has enhanced anti-tumor potency (Fig. 7F; Fig. 7G shows summary graphic).

## Discussion

A subset of cancer patients who are initially responsive to PD-L1 inhibitors will develop acquired resistance (Schoenfeld and Hellmann, 2020; Shah et al, 2018). Mechanisms to explain why immunologically 'hot' tumors turn 'cold' remain complex as the microenvironment can adapt to treatment by relying on alternative checkpoints, inducing permanent T cell exhaustion, and recruiting/expanding an array of immunosuppressive cells—amongst many

other changes attributed to host cell populations (reviewed in (Schoenfeld and Hellmann, 2020; Sharma et al, 2017)). But there is increasing evidence that the tumor also adapts to PD-L1 blockade (Chocarro de Erauso et al, 2020; Kalbasi and Ribas, 2020). Here, we examined the consequences of prolonged PD-L1 inhibition in vivo on tumor cells and identified a unique secretory signature that was associated with acquired resistance, enriched for numerous ISGs, and tightly regulated by type I IFN signaling. These tumor intrinsic adaptations were found to protect tumor cells from immune mediated cytotoxicity *directly*, via decreased sensitivity to lympho-cytic attack, and *indirectly*, via a suppression of T cell activation. Importantly, resistant tumors were found to be uniquely vulnerable to IFN signaling disruption which, when targeted in vivo, could partly reverse immune-protective effects and enhance tumor growth inhibition. Together, these findings suggest that a consequence of chronic PD-L1 blockade includes a tumor secretory signature that may serve as a biomarker and molecular driver of acquired resistance in patients.

Currently, the study of secretory changes in response to cancer treatments represent a phenomenon with broad implications for assessing (and improving) drug efficacy in patients (Mastri et al, 2018a; Shaked, 2016). Measuring systemic protein changes may provide clues into optimal drug dosing and give insight into emerging biological changes such as resistance (McKean et al, 2020; Wang et al, 2021). For ICIs, levels of secretory proteins may correlate with different outcomes depending on disease site and duration of treatment. For instance, secreted circulating factors such as IL6, IL-1Ra, and CCL2 have been found to initially increase in patients after PD-1 pathway inhibition and correlate with tumor stabilization/shrinkage, as measured by objective response rates (ORR) (Feng et al, 2020; Li et al, 2023; Lu et al, 2019b; Murakami et al, 2016; Naqash et al, 2018; Okiyama and Tanaka, 2017; Schoenfeld et al, 2019; Yamazaki et al, 2017). But few studies have measured cytokine changes in patients after prolonged (long-term) treatment durations and assessed progression-free and overall survival (PFS/OS) outcomes (Bridge et al, 2018; Nixon et al, 2019). Even when these are tested, results are often mixed. As an example, pre-treatment serum IL6 can correlate with improved initial response to nivolumab in a phase II trial for advanced melanoma (measured by ORR) (Yamazaki et al, 2017), but IL6 can also

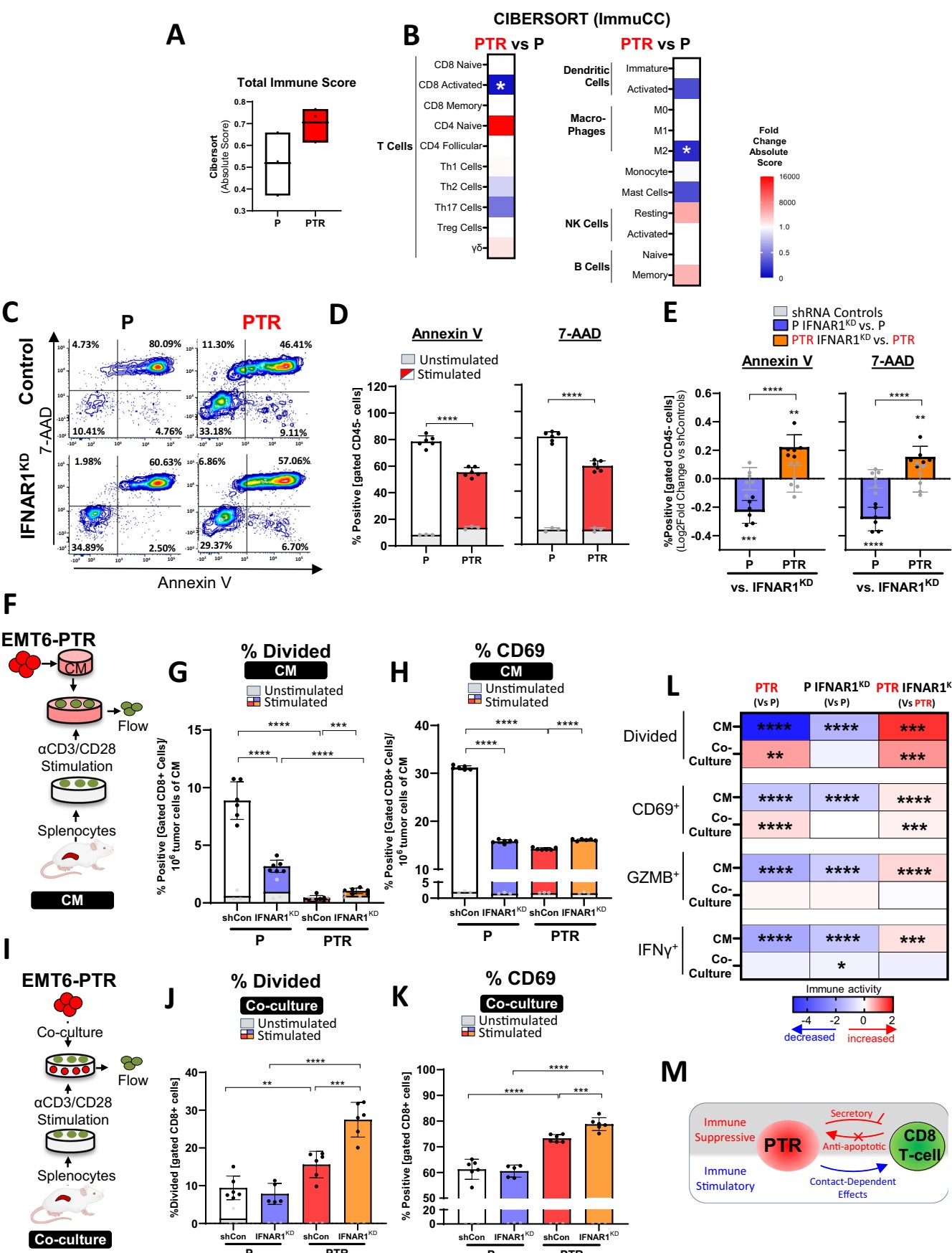

**Figure 6.  PTR immune-protective secretory changes are IFN signaling-dependent.**

(A) Cibersort tissue deconvolution analysis of EMT6-P and -PTR RNAseq data using ImmuCC mouse signature with box-plot representing absolute total immune score. $n = 3$. (B) Heatmap representing log2 fold change of absolute scores of various immune signatures of results from (A), statistics performed via two-tailed t-test. (C–E) Apoptosis (Annexin V) and cell death (7-AAD) staining of CD45-gated tumor cells after co-incubation with splenocytes showing (C) representative contour plots of stimulated splenocyte groups, (D) EMT6-P and -PTR variants, and (E) EMT6-P and -PTR-IFNAR1$^{KD}$ variants compared to controls. Flow cytometry, $n = 3$–6, statistics performed via two-tailed t-test. (F) Schematic of BALB/c-derived splenocyte proliferation and activation following incubation with EMT6-P and -PTR CM for experiments in (G, H). (G) CD8+ splenocyte division (CSFE dilution) after co-incubation with CM derived from EMT6-P and -PTR control and respective IFNAR1$^{KD}$ variants. Flow cytometry, $n = 3$–6, statistics performed via two-tailed t-test. (H) CD8+ splenocyte activation marker expression (CD69) after co-incubation with CM derived from EMT6-P and -PTR control and respective IFNAR1$^{KD}$ variants. Flow cytometry, $n = 3$–6, statistics performed via two-tailed t-test. (I) Schematic of Balb/c-derived splenocyte proliferation and activation following co-culture with EMT6-P and -PTR cells for experiments in (J, K). (J) CD8+ splenocyte division (CSFE dilution) after co-culture with EMT6-P and -PTR control and respective IFNAR1$^{KD}$ variants. Flow cytometry, $n = 3$–6, statistics performed via two-tailed t-test. (K) CD8+ splenocyte activation marker expression (CD69) after co-culture with EMT6-P and -PTR control and respective IFNAR1$^{KD}$ variants. Flow cytometry, $n = 3$–6, statistics performed via two-tailed t-test. (L) Heatmap summary of results from CM and co-culture experiments, statistics performed via two-tailed t-test. (M) Schematic summary of immune suppressive and stimulatory effects of PTR cells. Data Information: Parental (P); αPD-L1 Treatment-Resistant (PTR); IFNAR1 knockdown (IFNAR1$^{KD}$); shRNA vector control (shCon); Conditioned Media (CM); Bar graphs show mean ± SD. Box plot indicates range between minimum and maximum, central line depicts the mean. *$p \le 0.05$, **$p \le 0.01$, ***$p \le 0.001$, ****$p \le 0.0001$ compared to vector controls unless noted otherwise, for exact $p$ values see Fig. 6 Source data. All replicates shown represent technical replicates unless otherwise specified. Source data are available online for this figure.

predict for worse long-term outcome to PD-1/L1 inhibitors in NSCLC (measured by PFS) (Keegan et al, 2020) and to atezolizumab/bevacizumab in hepatocellular carcinoma (measured by PFS and OS) (Myojin et al, 2022). Nevertheless, our studies suggest that PTIS signatures linked to acquired resistance and IFN signaling may warrant further investigation as a biomarker. Already, transcriptomic analysis comparing clinical tumor tissues has shown that ISGs can increase in both tumor and non-tumor cell populations before and after ICI treatment in patients (Chen et al, 2018a; Chen et al, 2016; Efremova et al, 2018; Higgs et al, 2018). Our study showed that PTIS was enriched in 7 clinical and preclinical datasets involving αPD-L1 treatment-sensitive cancers, suggesting that testing PTIS in post-treatment biopsies may have utility as a predictor of tumor sensitivity. Yet challenges remain in using secretory changes as clinical biomarkers in general. First, there is the difficulty in obtaining post-treatment biopsy tissue where treatment dose/duration are closely monitored, as clinical samples may be obtained long after treatment has ceased. Our study utilized published clinical data from αPD-L1 treated NSCLC (Gettinger et al, 2017; Data ref: Gettinger et al, 2017) and MCC (Paulson et al, 2018; Data ref: Paulson et al, 2018) tumors, but such data is rare. Perhaps the ideal setting to test secretory biomarkers would be in tissues/blood obtained after neoadjuvant PD-L1 inhibition, of which several trials are currently underway (recently reviewed (Topalian et al, 2023)) including in cutaneous melanoma patients treated with atezolizumab for 6 weeks (NCT04020809). In such testing, it is important to consider how genomic and proteomic data can be optimally paired to maximize potential predictive utility as gene expression of secretory proteins does not always translate to similar changes in protein expression, and vice versa (Mastri et al, 2018a; Mastri et al, 2018b). For this reason, measuring circulating plasma protein measurements before and after αPD-L1 treatment in patients using multiplex immunoassay and proximity extension assays as a standardized assessment across studies may be considered to provide more consistent results (Arends et al, 2021; Eltahir et al, 2021; Loriot et al, 2021).

By examining tumor cells after long-term drug treatment, the results of our study identify PTIS as a *consequence* of αPD-L1 resistance. Yet, one question raised by our findings is whether PTIS can be a *cause* of resistance. Interestingly, the role of several PTIS

proteins may largely be context-dependent, as several have been linked to tumor-promoting and tumor-inhibiting immune responses. An example includes CCL2 and IL6, which we found to be increased in PTR cells and, thus far, have been associated with immune suppressive effects including promoting T regulatory cells (Mondini et al, 2019) and myeloid-derived suppressor cell populations (Weber et al, 2020), respectively. Furthermore, combination regimens of IL6 and CCL2 inhibition with ICIs have shown to delay development of resistance in preclinical studies (Tsukamoto et al, 2018a; Tu et al, 2020). But in other tumor models, these cytokines have been shown to have anti-tumor effects including proliferation, recruitment and trafficking of cytotoxic T cells (Fisher et al, 2014; Fisher et al, 2011; Harlin et al, 2009; Hopewell et al, 2013). In this regard, it is also of particular interest that the PTIS was found to be enriched in non-tumor populations such as macrophages and T cells of Merkel cell carcinoma patients treated with avelumab (Fig. 3E)—suggesting PD-L1 inhibition can likely induce 'off-target' host effects that can have negative effects on treatment efficacy (Mastri et al, 2018a). Using CIBERSORT tissue deconvolution of PTR tissue, we also show decreased activated T cell scores that suggest the presence of potential immune suppressive mechanisms in the tumor microenvironment. Notably, PTIS factors such as IL6 and IL1-RA have been shown to be associated with immune-related adverse events (irAEs) known to be induced by various ICI treatments (Ramos-Casals et al, 2020). PD-1 targeting agents have also been found to induce cytokine release syndrome, which is an adverse effect characterized by fever, myalgias, malaise, and high levels of cytokines including IL6 and IFNs that can be a barrier for continued treatment in patients (Ceschi et al, 2020; Rotz et al, 2017). Assessment of the PTIS as a potential cause of resistance deriving from the tumor, host, or both would require further investigation.

This raises the question of whether disruption of the PTIS in PTR cells should involve targeting the secretory-controlling IFN-signaling machinery or targeting ISG secretory products directly, thereby potentially avoiding disruption of IFNs anti-tumor functions. In our study, both approaches were evaluated, and IFNAR1 knockdown in PTR cells was found to effectively mitigate PTIS factor increases. In splenocytes studies, we noted PTR secretory factors were capable of suppressing T cell activation

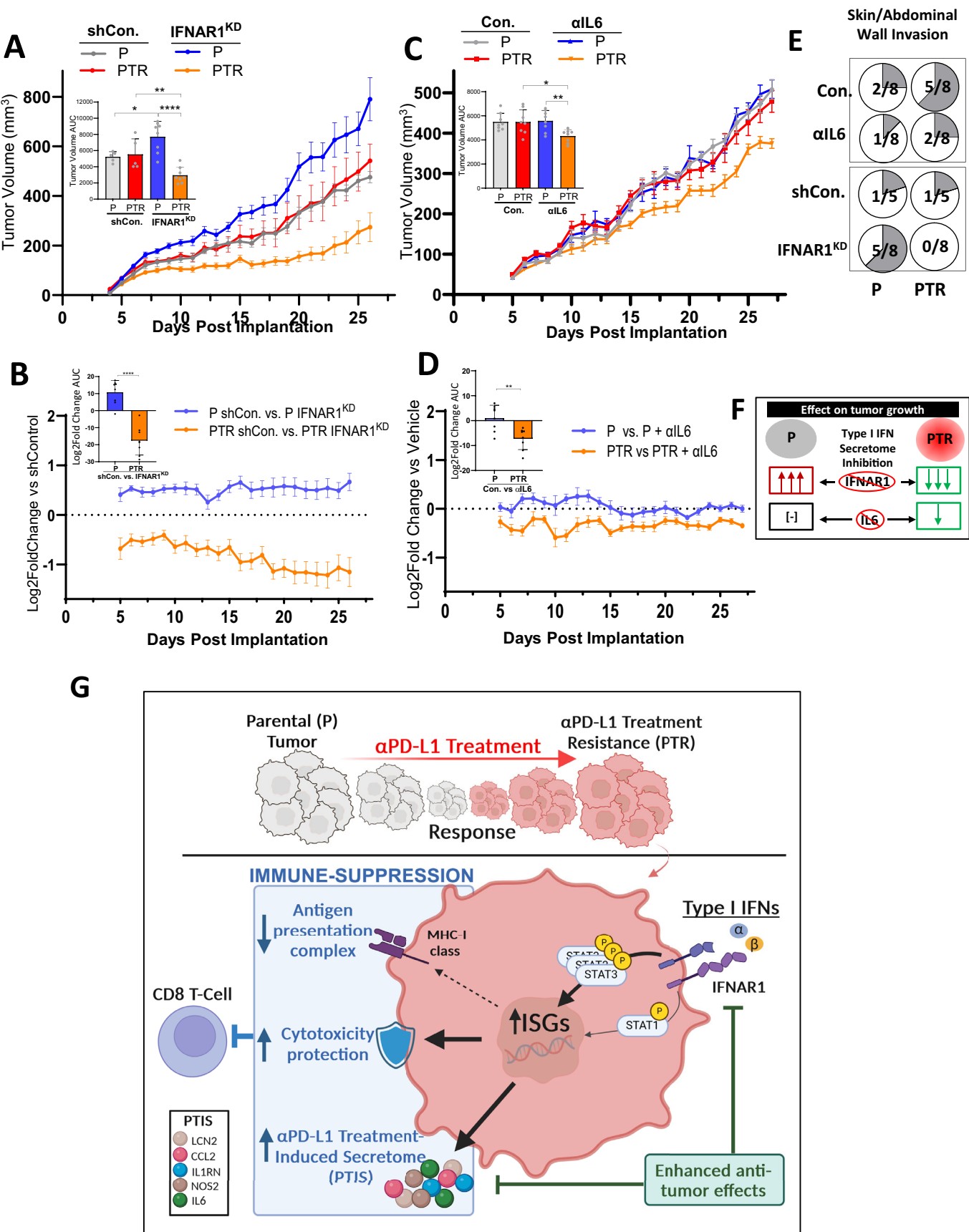

◄  **Figure 7. PTR tumor growth suppression by PTIS inhibition.**

(A) Orthotopic tumor growth of EMT6-P and EMT6-PTR, and respective IFNAR1$^{KD}$ cell variants ($n = 5$–8 biological replicates; Balb/c) with summary of AUC analysis (left-inset), statistics performed via two-tailed t-test. (B) Log$_2$ Fold Change analysis of data shown in (A) comparing IFNAR1$^{KD}$ to respective controls. AUC analysis of comparisons shown (left-inset), statistics performed via two-tailed t-test. (C) Orthotopic tumor growth of EMT6-P and EMT6-PTR treated with αIL6 antibody ($n = 5$–8 biological replicates; Balb/c) with summary of AUC analysis (left-inset), statistics performed via two-tailed t-test. (D) Log$_2$ Fold Change analysis of data shown in (C) comparing αIL6 antibody treatment to respective controls. AUC analysis of comparisons shown (left-inset), statistics performed via two-tailed t-test. (E) Metastasis and invasion of mice bearing EMT6-P and -PTR tumors (shown in A, B) with invasion in the peritoneum wall after PBS, αIL6, Con., or IFNAR1$^{KD}$ ($n = 5$–8 biological replicates). (F) Schematic summarizing the effect of type I IFN secretome inhibition in P and PTR tumors. (G) Proposed model of IFN-signaling 'rewired' tumor cells following acquired resistance to PD-L1 inhibition. Data Information: Parental (P); αPD-L1 Treatment-Resistant (PTR); IFNAR1 knockdown (IFNAR1$^{KD}$); shRNA Vector Control (shCon); Area under the curve (AUC) in vitro-derived Parental (P$^{VITRO}$); in vitro-derived PD-L1 Treatment Resistant (PTR$^{VITRO}$); Bar graphs show mean ± SD. *$p ≤ 0.05$, **$p ≤ 0.01$, ***$p ≤ 0.001$, ****$p ≤ 0.0001$, for exact $p$ values see Fig. 7 Source data. αIL6 was administered at 100 µg/mouse/3 days continuously, PBS was used as a control (Con.). Primary tumor burden was assessed by caliper measurement. Tumor growth line graph represent mean ± SEM. All replicates shown represent technical replicates unless otherwise specified. Source data are available online for this figure.

while direct co-culture studies revealed PTR cells increased T cell activation but had decreased sensitivity to T cell cytotoxicity. This may be explained by changes in cell surface proteins in PTR cells that can alter activation when bound directly with T cells and have reverse signaling that affect tumor cell survival. For instance, PD-L1 levels are significantly decreased in PTR cells which can result in increased T cell activation (Freeman et al, 2000). In addition, PD-L1 is also known to have reverse intrinsic signaling that can affect apoptosis and cell survival (Clark et al, 2016; Gato-Canas et al, 2017). Our studies show that antibody blockade and genetic disruption of PD-L1 can have tumor intrinsic effects including impact on PTIS expression (discussed below). It is important to note that these ex vivo splenocyte studies are limited in recapitulating PTR-immune cell interactions for several reasons. First, splenocyte studies were conducted using CD3/CD28 agonistic antibodies to bypass initial signals for T cell activation such as antigen presentation. However, despite this, our findings suggest that PTR secretomes were capable of suppressing T cell activation independent of antigen presentation. Second, our studies were focused on assessing secretory changes in PTR cells and how these changes affect the resistant phenotype, additional studies would be needed to investigate changes in cell surface and signaling changes after resistance. Lastly, splenocyte studies were focused on assessing interactions between PTR cells and activated T cells, additional tumor-immune cell interactions known to affect tumor progression such as myeloid and NK cell populations would require further investigation. In this regard, we assessed tumor growth in vivo and found that PTR-IFNAR1$^{KD}$ tumors grew slower than PTR tumors, suggesting that constant PD-L1 blockade confers a unique vulnerability to IFN blockade in tumor cells. However, our results also show that IFNAR1$^{KD}$ tumors grew much faster than parental controls, emphasizing the sometimes contradictory role of IFNAR signaling in cancer controlling immunosurveillance (Boukhaled et al, 2021) which might be exacerbated by αPD-L1 treatment and, as a consequence, may introduce potential challenges of intracellular IFN signaling inhibition strategies.

For targeting an individual PTIS factor, we evaluated IL6—which is a specific component of the PTIS found to be consistently increased in PTR cells and then further enhanced after IFNβ stimulation. IL6 is known to activate a multitude of tumor promoting effects that include (i) enhancing expression of pro-angiogenic factors in tumor cells (e.g., VEGF, IL1β, IL8 (Lederle et al, 2011)), (ii) suppressing antigen presentation from dendritic cells (Kitamura et al, 2005), (iii) promoting pro-tumorigenic macrophage phenotypes (Fu et al, 2017), and (iv) suppressing anti-tumor functions of CD4+ T cells (Tsukamoto et al, 2015) (amongst many others (Tsukamoto et al, 2018b)). Trials are currently underway testing IL6 and PD-1 pathway inhibitor combinations for improved anti-tumor efficacy and irAEs (NCT03999749, NCT04258150) (Doms et al, 2020; Stroud et al, 2019). Indeed, our results showing that IL6 inhibition can lead to enhanced growth suppression in the acquired resistance setting adds to the growing literature supporting αIL6/PD-L1 combination/sequencing strategies. However, it should be noted that these effects were largely modest, suggesting targeting multiple PTIS factors simultaneously may yield more robust outcomes. In favor of this, preclinical models involving inhibition of CCL2 (Flores-Toro et al, 2020), NOS2 (Jacquelot et al, 2019), and LCN2 (Leng et al, 2009) pathways have shown anti-tumor effects and may be explored to improve IL6 inhibitory strategies after PD-L1 treatment failure.

A unique feature of our studies is the use of in vivo-derived models to evaluate acquired resistance, which served to mimic the evolution of treatment-sensitive tumor progression to eventual insensitivity. For ICIs, acquired resistance is important but surprisingly difficult to recapitulate in preclinical systems. Murine tumor models have translational value by providing an accurate assessment of tumor-immune interactions, yet the majority do not demonstrate an initial response to single agent ICI treatment (previously reviewed in (Olson et al, 2018; Sanmamed et al, 2016)). This is perhaps best demonstrated by Mosely et al, where six of the most commonly used syngeneic tumor models showed that only two (CT26 and RENCA) were sensitive to anti-CTLA-4 therapy and only one (CT26) was sensitive to anti-PD-L1 (Mosely et al, 2017); with later studies suggesting that CT26 responses to PD-L1 inhibition may depend on antibody clonality (Kumar et al, 2020) and duration/timing of treatment schedules (Schaer et al, 2018). Explanation for this limited efficacy is still unclear and may be model-dependent due to variations on T cell infiltration (Yu et al, 2018), infiltration of immunosuppressive cell types (Mosely et al, 2017), tumor mutation burden (Mosely et al, 2017), and both tumor and non-tumor host expression of immune-checkpoint molecules (Lau et al, 2017). Our studies identify EMT6 as a treatment-sensitive model for acquired resistance that can be used to confirm molecular changes with clinical datasets to establish translational relevance. As ICIs are approved in new disease indications, the use of such models will be increasingly important

for studies examining mechanisms, biomarkers, and therapeutic solutions for ICI efficacy.

Assessing the role of IFN signaling on tumor growth and overall efficacy of ICIs remains complex. This is because IFNs can have opposing, and often contradictory, effects depending on several factors that include the type of cell, the duration of IFN exposure, and the stage of tumor progression (Boukhaled et al, 2021; Reading and Quezada, 2016). These stimulating/inhibiting effects of IFNs can, in turn, influence ICI treatment response. For *inhibitory* effects, IFNs (mostly IFNγ) have historically been characterized to be integral for anti-tumor immunity by driving antigen presentation (Gettinger et al, 2017) and chemokine secretion (Chheda et al, 2016) that are typically part of the immune-editing process in normal physiological conditions. IFNs can improve ICI responses as loss-of-function mutations in IFN signaling components (i.e., JAK1/2) have been identified in melanoma patients after pembrolizumab treatment relapse (Zaretsky et al, 2016), and knockout of IFN pathway mediators such as Jak1, Stat1, Ifngr1 in tumors can weaken ICI treatment efficacy (Manguso et al, 2017; Torrejon et al, 2020). For *stimulating* effects, ICIs can induce an enhanced expression of tumor cell ISGs transcribing additional T-cell co-inhibitory ligands (e.g., TNFRSF14, LGALS9) where blockade of type I and II IFN signaling could reverse this effect and improve ICI responses (Benci et al, 2016). Other tumor ISGs regulated by type I IFNs such as NOS2 (Jacquelot et al, 2019) and CD38 (Chen et al, 2018a) can also have immune-suppressive effects and have been implicated to promote resistance to PD-1 pathway blockade. In this context, one question is whether negative consequences of IFNs can be specifically targeted, without negating the positive IFN signaling effects. In this regard, Benci and colleagues proposed that ISGs promoting ICI resistance may be more associated with tumor cell expression (Benci et al, 2019) and after chronic exposure to IFNs (Benci et al, 2016), compared to the largely positive effects of IFNs on immune cells and after acute IFN exposure. Indeed, in murine tumor models, success of ICI treatment depends on a 'display-on/fast-off' kinetic of IFNβ signaling where fast reduction of ISG expression is associated with responding tumors (Zemek et al, 2022). Therapeutically, sequential treatment of an ICI followed by a JAK inhibitor was found to sensitize IFN-driven resistant tumors to ICI treatment (Benci et al, 2016). Such treatment strategies tested in early phase trials have shown promising efficacy (Mathew et al, 2024; Zak et al, 2024). Notably, we performed GSEA analysis and found an enrichment of JAK/STAT pathways in EMT6-PTR tumors, suggesting that targeting JAK therapeutically may undermine the PTIS when combined with PD-L1 inhibitors (Appendix Fig. S10 and Appendix Table S4). These studies are currently underway.

Lastly, results from our studies suggest tumor-intrinsic signaling can lead to extrinsic secretory changes after PD-L1 inhibition, potentially stemming from changes caused by PD-L1 directly. Though often overlooked, PD-L1 has several tumor intrinsic functions that have recently been identified to regulate mTOR/AKT (Chang et al, 2015; Clark et al, 2016), MAPK (Wu et al, 2018), STAT3/Caspase 7 (Gato-Canas et al, 2017), integrin β4 (Wang et al, 2018), MerTK (Du et al, 2021), and BIM/BIK (Feng et al, 2019) signaling, among others. Recently, Gato-Canas and colleagues reported that conserved motifs of PD-L1 cytoplasmic domains (RMLDVEKC and DTSSK) block STAT3/Caspase 7 cleavage and, in turn, can control type I IFN-induced cytotoxicity (Gato-Canas

et al, 2017). This result introduced PD-L1 as a 'molecular shield' that can protect tumor cells from T cell cytotoxicity caused by various treatments (Azuma et al, 2008), including IFNs (Gato-Canas et al, 2017; Gupta et al, 2016). Our results align with these previous studies suggesting that PD-L1 has both activating and inhibitory domains that cross-talk with IFN signaling where tumor cells following genetic and in vitro therapeutic PD-L1 disruption can recapitulate some; but not all, of the IFN-regulated PTIS observed after PD-L1 resistance (Gato-Canas et al, 2017). In addition, these findings indicate that the molecular 'rewiring' of tumor cells after acquired resistance to PD-L1 requires the complex molecular and cellular changes from host processes that indirectly alter tumor cell PD-L1 and IFN signaling. For instance, PD-L1 inhibition is known to modulate functions of immune cells such as T-, NK-, and myeloid cell populations, amongst several others to drastically remodel the tumor microenvironment (Gubin et al, 2018; Gubin et al, 2014; Wei et al, 2017) as well as modulate local and systemic cytokine and growth factor changes that include IFNs (Mastri et al, 2018a). Given the complexity of these factors, our findings highlight the challenges of studying acquired resistance mechanisms to PD-L1 inhibition and the critical need for appropriate in vivo systems to bridge findings from in vitro studies involving PD-L1 functional signaling, particularly as it pertains to translational significance of therapeutic relapse as it occurs in patients.

Taken together, our results show that tumor cells following acquired resistance to PD-L1 blockade can express an ISG enriched secretory profile associated with diminished sensitivity to immune cell cytotoxicity. Therapeutic approaches involving inhibition of PTIS components or IFN regulators may have enhanced benefit after resistance to PD-L1 inhibition.

# Methods

## Cell lines

Cells used in this study include: Mouse mammary carcinoma EMT6 (from A. Gudkov, Roswell Park Comprehensive Cancer Center, RPCCC), mouse colorectal carcinoma CT26 (A. Gudkov), and mouse kidney RENCA (from R. Pili, RPCCC). Cells were maintained in RPMI (Corning cellgro #10-040-CV) supplemented with 5% v/v FBS (Corning cellgro; 35-010-CV). All cells were maintained at 37 °C with 5% $CO_2$ in a humidified incubator. All untreated parental cell lines were authenticated for species origin (DDC Medical, USA) and tested for mycoplasma contamination.

## Drug and recombinant protein concentrations

IgG1 (NIP228, AstraZeneca), IgG2a (I-1177, Leinco Technologies Inc), αPD-L1 (Clone 80, AstraZeneca), αPD-L1 (MIH5, from M. Azuma, Tokyo Medical and Dental University (Tsushima et al, 2003)), and αIL6 (BE0046/MP5-20F3, BioXCell) were prepared as follows: For in vivo experiments: αPD-L1 (Clone 80) and αIL6 (MP5-20F3) were diluted in PBS and administered by intraperitoneal injection at (250 μg/mouse/3 days) or (100 μg/mouse/3 days), respectively. Tumor-related differences between any vehicle or IgG groups were not observed. In vitro, IgG (NIP228

**Reagents and tools table**

| REAGENT or RESOURCE | SOURCE | IDENTIFIER |
|---|---|---|
| **Antibodies** | | |
| Goat anti-rabbit IgG (H + L); HRP conjugated | Promega | Cat#W4011; RRID: AB_430833 |
| Goat anti-mouse IgG(H + L); HRP conjugated | Promega | Cat#W4021; RRID: AB_430834 |
| IgG1 | AstraZeneca | NIP228 |
| Rat IgG2a Isotype Control | Leinco Technologies Inc | Cat#I-1177<br>RRID: AB_2737530 |
| Anti-mouse PD-L1 antibody (MIH5) | M. Azuma, Tokyo Medical and Dental University | N/A |
| Anti-mouse IL6 antibody | BioXCell | Cat#BE0046/MP5-20F3<br>RRID: AB_1107709 |
| Phospho-STAT1 (Tyr701) (58D6) Rabbit Monoclonal | Cell Signaling | Cat#9167S<br>RRID: AB_561284 |
| STAT1 (D1K9Y) Rabbit Monoclonal | Cell Signaling | Cat#14994<br>RRID: AB_2737027 |
| Phospho-STAT3 (Tyr705) (D3A7) XP Rabbit Monoclonal | Cell Signaling | Cat#9145T<br>RRID: AB_2491009 |
| STAT3 (124H6) Mouse Monoclonal | Cell Signaling | Cat#9139T<br>RRID: AB_331757 |
| Goat Anti-mouse Lipocalin-2/NGAL Polyclonal | R&D Systems | Cat#AF1857-SP<br>RRID: AB_355022 |
| iNos (NOS2) Rabbit Monoclonal | ABclonal | Cat#A3774; RRID: AB_3094627 |
| Anti-Beta-Actin Monoclonal | Sigma Aldrich | Cat#A5441; RRID: AB_476744 |
| Purified anti-mouse CD3 antibody (Clone 17A2) | Biolegend | Cat#100202; RRID: AB_312659 |
| Purified anti-mouse CD28 antibody (Clone 37.51) | Biolegend | Cat#102102; RRID: AB_312867 |
| Anti-mouse CD8b.2 antibody; PE-Cy7 conjugated (Clone 53-5.8) | Biolegend | Cat#103128; RRID: AB_493715 |
| Anti-mouse CD69 antibody; APC conjugated (Clone H1.2F3) | Biolegend | Cat#140416; RRID: AB_2564385 |
| Anti-human/mouse Granzyme B antibody; Pacific Blue conjugated (Clone GB11) | Biolegend | Cat#515408; RRID: AB_2562196 |
| PE Anti-mouse IFNγ antibody (Clone XMG1.2) | Biolegend | Cat#: 505808<br>RRID: AB_315402 |
| Anti-mouse CD69 APC antibody (Clone H1.2F3) | Biolegend | Cat#: 104514<br>RRID: AB_492844 |
| Anti-mouse PD-L1 antibody (Clone 80) | Astra Zeneca (MTA) | N/A |
| APC Annexin V | Biolegend | Cat#: 640920; RRID: AB_2561515 |
| Anti-mouse MHC-I (H-2kb/H-2Db/H-2Dd); APC conjugated (Clone 28-8-6) | Biolegend | Cat#: 114613; RRID: AB_2750193 |
| **Chemicals, peptides, and recombinant proteins** | | |
| Recombinant mouse Interferon-alpha-2 | Sino Biological | Cat#: 50525-MNAY; AS#: NP_034633.2 |
| Recombinant mouse interferon-beta | Sino Biological | Cat#: 50708-MCCH; AS#: P01575 |
| Recombinant mouse interferon-gamma | Peprotech | Cat#: 315-05; CAS: |
| RPMI | Corning | Cat#: 10-040-CV |
| DMEM | Corning | Cat#: 10-013-CV |
| Heat Inactivated-FBS | Gibco | Cat#: 10437028 |
| Heat Inactivated FBS | Corning | Cat#: 35-010-CV |
| Trypsin | Corning | Cat#: 25-053-CI |
| Polybrene: Hexadimethrine bromide | Sigma Aldrich | Cat#: H9268-5G |
| Puromycin Dihydrochloride | Gibco | Cat#: A11138-03 |
| Beta-mercaptoethanol | Gibco | Cat#: 21985023 |

| REAGENT or RESOURCE | SOURCE | IDENTIFIER |
|---|---|---|
| Non-essential amino acids | Gibco | Cat#: 11140050 |
| Sodium Pyruvate | Gibco | Cat#: 11360070 |
| Penicillin/Streptomycin | Corning | Cat#:30-001-CI |
| RBC lysis buffer | Biolegend | Cat#: 420301 |
| CFSE | Biolegend | Cat#: 423801 |
| Zombie Aqua Fixable Viability Kit | Biolegend | Cat#: 423102 |
| Activation cocktail + Brefeldin A1 | Biolegend | Cat#: 423303 |
| Accutase | Biolegend | Cat#: 423201 |
| Matrigel Basement Membrane Matrix | Corning | Cat#: 354234 |
| Tumor Dissociation Kit | Miltenyi Biotec | Cat#: 130-095-929 |
| 7-AAD Viability Staining Solution | Biolegend | Cat#: 420404 |
| **Critical commercial assays** | | |
| Qiagen RNA Mini Isolation Kit | Qiagen | Cat#: 74104 |
| TruSeq Stranded mRNA kit | Illumina Inc. | |
| KAPA Biosystems qPCR kit | | |
| NextSeq500 | Illumina Inc. | |
| AMPure XP Beads | Beckman Coulter | |
| QIAshredder | Qiagen | Cat#: 79654 |
| DNaseI | Qiagen | Cat#: 79254 |
| Mouse XL Cytokine Array Kit | R&D Systems | ARY028 |
| E.Z.N.A. Plasmid Mini Kit I | Omega Bio-Tek, Inc. | Cat#: D6943 |
| DC Protein Assay | Bio-Rad | Cat#: 500-0112 |
| CellTiter 96 Aqueous Non-Radioactive cell proliferation kit (MTS) | Promega | Cat#: G1112 |
| Mouse IL6 ELISA | Biolegend | Cat#: 431304 |
| Mouse Phospho-STAT3 ELISA | Biolegend | Cat#: 7300C |
| Mouse PD-L1 DuoSet ELISA | R&D Systems | Cat#: DY1019-05 |
| Mouse Lipocalin-2/NGAL (LCN2) DuoSet ELISA | R&D Systems | Cat#: DY1857-05 |
| Mouse IFNβ DuoSet ELISA | R&D Systems | Cat#: DY8234-05 |
| **Deposited data** | | |
| Gene expression changes in mouse breast tumor tissue (EMT6) after anti-PD-L1 resistance | This paper | GEO: GSE186032 |
| Gene expression changes in mouse breast cancer cells (EMT6) after anti-PD-L1 treatment | This paper | GEO: GSE186034 |
| Gene expression changes in mouse kidney cancer cells (RENCA) after anti-PD-L1 resistance | This paper | GEO: GSE186037 |
| Gene expression data from 4T1 mouse tumors treated with anti-PD-L1 antibody | Sceneay et al, 2019 | GEO: GSE130472 |
| Gene expression data from mouse EMT6 mouse tumors treated with anti-PD-L1 antibody | Lan et al, 2018 | GEO: GSE107801 |
| Gene expression data from MC38 mouse tumors treated with anti-PD-L1 antibody | Efremova et al, 2018 | GEO: GSE93017 |
| Gene expression data from human lung cancer patients treated with anti-PD-L1 antibody | Gettinger et al, 2017 | dbGaP: phs001464.v1.p1 with permission under protocol #26165 |
| Gene expression data from human Merkel cell carcinoma patients treated with anti-PD-L1 antibody | Paulson et al, 2018 | GEO: GSE117988, GSE118056 |

| REAGENT or RESOURCE | SOURCE | IDENTIFIER |
|---|---|---|
| **Experimental models: cell lines** | | |
| EMT6 | Laboratory of A. Gudkov | |
| RENCA | Laboratory of R. Pili | |
| RENCA-PDL1 | This paper | |
| EMT6-P | This paper | |
| EMT6-PTR | This paper | |
| EMT6[PDL1-KD] | This paper | |
| EMT6[shCon] | This paper | |
| EMT6[IFNAR1-KD] | This paper | |
| CT26 | Laboratory of A. Gudkov | |
| **Experimental models: Organisms/strains** | | |
| Balb/cAnNCrl mouse | Charles River Labs | |
| **Oligonucleotides** | | |
| Primers for qRT-PCR, see Table S4 | This paper; Integrated DNA Technology (IDT) | N/A |
| **Recombinant DNA** | | |
| PD-L1 shRNA: CCGAAATGATACACAATTCGA | This paper | |
| IFNAR1 shRNA: GCCAGAGACTACTTACTGTTT | This paper | |
| **Primers Used for qRT-PCR Assays** | | |
| Ifnb1: (forward)GAGCAGAGATCTTCAGGAAC (length:20) (reverse)AGATTCACTACCAGTCCCAG (length:20) (Mature Amplicon Length: 150) | This paper | |
| Ifna1: (forward)ACCTTCCTCAGACTCATAACC (length:21) (reverse)GGGCATCCACCTTCTCC (length:17) (Mature Amplicon Length: 132) | This paper | |
| Nos2: (forward)GGAGATCAATGTGGCTGTG (length:19) (reverse)CGGTACTCATTCTGCATGTG (length:20) (Mature Amplicon Length: 108) | This paper | |
| Il6: (forward)AGCCAGAGTCCTTCAGAG (length:18) (reverse)GGTCCTTAGCCACTCCTTC (length:19) (Mature Amplicon Length: 148) | This paper | |
| Cxcl10: (forward)CAGCACCATGAACCCAAG (length:18) (reverse)TGGCCCTCATTCTCACTG (length:18) (Mature Amplicon Length: 140) | This paper | |
| Rsad2: (forward)CCCTCTGTGAGCATAGTGAG (length:20) (reverse)GCCAATCAGAGCATTAACCTG (length:21) (Mature Amplicon Length: 129) | This paper | |
| Cxcl9: (forward)CCGCTGTTCTTTTCCTCTTG (length:20) (reverse)GGTCTTTGAGGGATTTGTAGTG (length:22) (Mature Amplicon Length: 135) | This paper | |
| Serpinb9: (forward)GGCATCAACCATTTAACAAAGAG (length:23) (reverse)GGCGAGGTTATATGTGTCTTC (length:21) (Mature Amplicon Length: 110) | This paper | |

| REAGENT or RESOURCE | SOURCE | IDENTIFIER |
|---|---|---|
| Ccl5: (forward)CCCTCACCATCATCCTCAC (length:19) (reverse)GTAGAAATACTCCTTGACGTGG (length:22) (Mature Amplicon Length: 137) | This paper | |
| **Software and algorithms** | | |
| FCS Express 7 De Novo Software | | N/A |
| FACSDiva Software | BD Biosciences | N/A |
| Graphpad v8.4.0 | GraphPad Software Inc., San Diego, CA | |
| 4200 TapeStation D1000 Screentape | Agilent Technologies, Inc. | |
| FASTQC v0.11.5 | | http://www.bioinformatics.babraham.ac.uk/projects/fastqc/ |
| STAR v2.6.0a | | |
| RSeQC v2.6.5 (Rusinova et al, 2013) | | |
| RSEM v1.3.1 (Sade-Feldman et al, 2017) | | |
| R v.3.6.0 | The Comprehensive R Archive Network | N/A |
| RStudio v.1.1.463 | Integrated Development for R/Posit | http://www.rstudio.com/ |
| Living Image Software (IVIS Imaging Systems) | IVIS Spectrum (Perkin Elmer): S10OD16450 | |
| Limma v.3.56.2 | | https://bioconductor.org/packages/release/bioc/html/limma.html |
| EdgeR v.3.17 | | |
| Gene Set Enrichment Analysis (GSEA) | Broad Institute | https://www.gsea-msigdb.org/gsea |
| Gene Ontology Enrichment Analysis (GO) | Gene Ontology Consortium | https://geneontology.org |
| ClueGo | | |
| Cytoscape v3.10.2 | | https://cytoscape.org/ |
| Interferome v2.01 | | http://www.interferome.org/interferome/home.jspx |
| Cibersort/ImmuCC | | |
| Image Lab™ v6.0.1 build 34 | Bio-Rad | |
| Image Lab™ Touch Software v.2.4.0.03 | Bio-Rad | |
| CFX Maestro | Bio-Rad | https://www.bio-rad.com/en-us/product/cfx-maestro-software-for-cfx-real-time-pcr-instruments?ID=OKZP7E15 |
| xMark Microplate Spectrophotometer MPM6.exe v6.3 | Bio-Rad | |
| NanoDrop 2000/2000c v1.6.198 | Thermo Fisher Scientific | |
| **Other** | | |
| Lysis Buffer 17 | R&D Systems | Cat#895943 |
| Protease Cocktail | Fisher Scientific | Cat#P178430 |
| Halt Protease Inhibitor | Thermo Fisher Scientific | Cat#78429 |
| Pierce ECL Western Blotting Substrate | Thermo Scientific | Cat#32106 |
| iTaq SYBR Green Supermix | Bio-rad | Cat# 1725121 |
| FCS Express 7 De Novo Software | | N/A |
| FACSDiva Software | BD Biosciences | N/A |
| iScript cDNA synthesis kit | Bio-Rad | Cat#1708891 |
| LipoD293 Transfection Reagent | SignaGen Laboratories | Cat#SL100668 |

or I-1177) and αPD-L1 (Clone 80 or MIH5) in PBS was directly added to media for maintenance at a concentration of 0.5 µg/ml; αIL6 was used at a concentration of 10 µg/ml; recombinant IFN-alpha-2 (50525-MNAY, Sino Biological), IFN-beta (50708-MCCH, Sino Biological), IFNγ (315-05, Peprotech) were used at 10 ng/ml.

## shRNA knockdown studies

For production of IFNAR1 and PD-L1 knockdown lentivirus, pLKO.1-puro shRNA plasmid DNA was isolated from bacteria glycerol stocks (IFNAR1: TRCN0000301483, PD-L1: TRCN0000068001; Sigma Aldrich) using E.Z.N.A.® Plasmid Mini Kit I (Omega Bio-tek, Inc.). To produce lentiviral media, 293T cells were transiently co-transfected with DNA from the lentiviral pLKO.1-puro shRNA plasmid and psPAX2 and pMD2.G packaging plasmids using LipoD293™ Transfection Reagent (SignaGen Laboratories). Conditioned media containing virions was harvested after 24 and 48 h, filtered through a 0.45-µm membrane, and used to infect EMT6-P or PTR cells. Cells were infected with the shRNA and vector controls by spin inoculation at $600 \times g$ for 45 min at room temperature in the presence of 5 µg/ml polybrene. Viruses were removed after an additional 6 h incubation at 37 °C/5% $CO_2$ and cell culture media was replaced. Puromycin selection was then conducted for 2 weeks at 2 µg/ml until stably infected cells were generated. Knockdown was confirmed via flow cytometry analysis.

## Mouse tumor models

### Study approval

Animal tumor model studies and animal housing conditions were performed in strict accordance with the recommendations in the Guide for Care and Use of Laboratory Animals of the National Institutes of Health and according to guidelines of the IACUC at RPCCC (Protocol: 1227M).

### Orthotopic tumor implantations

EMT6 ($5 \times 10^5$ cells in 100 µl RPMI), RENCA ($4 \times 10^4$ cells in 5 µl 1:1 RPMI:Matrigel) were implanted orthotopically into the right inguinal mammary fad pad or left kidney subcapsular space respectively in 6–8-week-old female NCI Balb/cAnNCr (Balb/c) mice (Charles River, USA). Isoflurane (anesthesia) and buprenorphine (analgesic) were used during all surgical implantations. Mammary fat pad tumors were measured using Vernier calipers and volumes were calculated using the formula (width$^2$ × length) × 0.5. Kidney luciferase expressing tumors were assessed for bioluminescence activity bi-weekly. All animals were assessed 2–3 times daily by veterinary staff or personnel approved by IACUC for pre-defined endpoints. Institutional end-points included primary tumor-based morbidities (>2000 mm$^3$ volume) and metastasis-related morbidities (labored breathing, 20% weight loss, cachexia, limb paralysis) have been previously described in detail (Ebos et al, 2014; Tracz et al, 2014). Mice were randomized before implantation in a blinded manner. Mice in study were not excluded for analysis. Mouse sample size were established as 3–4 for selection of drug-resistant variants and 5–8 for assessment of knockdown and drug treatments. Mice were housed in cages (maximum 5 per cage) individually ventilated with HEPA filtered air and provided with bedding, an ad libitum supply of standard chow diet and water, and regular light-dark cycles as per AAALAC International accreditation standards.

## Resistance cell derivation and maintenance

For in vivo-derived resistant cell variants, mice were orthotopically implanted with EMT6 or RENCA and treated with αPD-L1 (Clone80) until institutional endpoint. For both EMT6 and RENCA, parental (P) cell lines were obtained from IgG-treated mice and used as controls. All variants were selected from primary tumors which were minced, enzymatically digested (Miltenyi Biotech; 130-095-929), and then placed in RPMI media (supplemented with 5% v/v FBS, 100 IU/ml penicillin and 1000 µg/ml streptomycin) with IgG (NIP288) for P variants or with αPD-L1 (clone 80) (0.5 µg/ml) for PTR variants. Antibiotics were then removed 1 week after in vivo cell selection. For derivation of PTR$^{VITRO}$ cell variants, EMT6 and CT26 cells were treated with αPD-L1 antibodies (Clone 80 at 0.5 µg/ml or MIH5 at 0.5 µg/ml, respectively) for >4 weeks.

## Cell proliferation assay

Proliferation was examined using CellTiter 96 Aqueous Non-Radioactive cell proliferation (MTS) assay (Promega; G1112). For 5 day growth studies, 200 cells/well were plated in 48-well plates. The next day, cells were treated with recombinant IFNs or αIL6. Treatments were replaced every 2 days or removed daily for MTS measure of viability. RPMI + 5% FBS was mixed with MTS per manufacturer instructions and added to cells at timepoints. After 2 h incubation, optical density was measured at a wavelength of 490 nm (Bio-Rad xMark).

## RNA isolation

Cells were plated at 80,000 cells/well in a 6-well plate with corresponding treatments as indicated. 48 h later, total RNA was isolated using QIAshredder (QIAGEN; 79654) and RNase mini kit (QIAGEN; 74104). Genomic DNA was then digested using DNaseI (QIAGEN; 79254) per manufacturer instructions. RNA concentration was determined using nanodrop 2000c (Thermo Scientific) before RNAseq and PCR analysis.

## qRT-PCR

For reverse transcription using iScript cDNA synthesis kit (Bio-Rad; 170-8891), 1 µg RNA was used according to the manufacturer's instructions. qRT-PCR was performed using iTaq SYBR Green Supermix (Bio-Rad; 1725121). Thermocycling parameters were: 10 min at 95 °C, 15 s at 95 °C, 40 cycles at 95 °C for 15 s at 95 °C and 1 min at 60 °C, 1 min at 95 °C, followed by a melting curve: 55 to 95 °C with increments of 0.5 °C for 5 s. Relative gene expression was calculated using the formula $2^{-[CT(House Keeping Gene)-Ct(Gene of Interest)]}$, with CT representing the fixed threshold cycle value for fluorescent signal. Gapdh and Actb were used for housekeeping genes. Oligonucleotides were purchased from Integrated DNA Technologies (IDT).

## Proteome profiler (cytokine) array

Cells were lysed with lysis buffer 17 (R&D Systems; 895943) supplemented with protease cocktail (Fisher Scientific, PI78430). Total protein levels were quantified with DC protein assay (Bio-Rad; 500-0112). 200 µg of total mouse protein samples were

analyzed, respectively, with a Mouse XL Cytokine Array Kit (R&D Systems; ARY028) per manufacturer instructions. Membranes were exposed to X-ray films, which were imaged (digitized) with ChemiDoc System (Bio-Rad) and analyzed with Image Lab Software (Bio-Rad). Multiple X-ray film exposures were obtained and shortest exposure was selected for each protein target when signals could be detected. Background was then subtracted from densitometry volumes and all data was normalized to positive controls for each array.

## ELISA analysis

Cells were lysed with lysis buffer I (20 mM Tris (pH 7.5), 127 mM NaCl, 10% Glycerol, 1% v/v NP40 (Igepal), 100 mM NaF, 1 mM $Na_3VO_4$) and protein concentrations were quantified with DC protein assay. For conditioned media collection, cells were counted for normalization. IL6, phospho-STAT3, PD-L1, and LCN2 were measured using mouse IL6 ELISA (431304, Biolegend), mouse phospho-STAT3 ELISA (7300C), mouse PD-L1 Duoset ELISA (DY1019-05), and mouse Lipocalin-2/NGAL(LCN2) DuoSet ELISA (DY1857-05).

## Western blot analysis

Cells were lysed with lysis buffer II (50 mM Tris (pH 8), 2% w/v SDS, 5 mM EDTA, 3 mM EGTA, 25 mM NaF, 1 mM $Na_3VO_4$) supplemented with Halt™ protease inhibitor (Thermo Fischer Scientific 78429). Lysates were sonicated for 2 s and total protein concentration was quantified with DC protein assay. Proteins samples were prepared with 1/5 volume of 5x SDS-PAGE sample buffer (250 mM Tris pH 6.8, 10% w/v SDS, 25% v/v glycerol, 500 mM DTT, and bromophenol blue). Proteins (40 μg per lane) were resolved by SDS-PAGE, electrotransferred to Immobilon-P membrane, and incubated with a primary antibody diluted as recommended by the manufacturer. Membranes were then probed with a horseradish peroxidase-conjugated secondary antibody (Promega W4011 and W4021 or R&D Systems HAF017) and protein signals were developed using the Pierce ECL Western blotting substrate (Thermo Scientific; 32106). X-ray films were imaged (digitized) with ChemiDoc System and analyzed with Image Lab Software. Primary antibodies were purchased from Cell signaling (phospho-STAT1 Tyr701 9167S, STAT1 14994, phospho-STAT3 Tyr705 9145T, STAT3 9139T), R&D Systems (Lipocalin-2/NGAL, AF1857-SP), ABclonal (iNos, A3774), and Sigma Aldrich (β-actin, A5441).

## Splenocyte division and activation assays

Spleens were harvested from Balb/c mice, mechanically dissociated by passing through a 70-μm filter, and collected in complete RPMI media (supplemented with 10% heat-inactivated FBS, 1% non-essential amino acids, 1% sodium pyruvate, 1% penicillin/streptomycin, and 0.1% β-mercaptoethanol). Splenocytes were then treated with RBC lysis buffer (Biolegend, 420301) and incubated overnight. The next day, splenocytes were stained with CFSE (Biolegend, 423801), stimulated with mouse αCD3 (Biolegend, 100202) and αCD28 (Biolegend, 102102) according to manufacturer recommendations. To generate conditioned media, $7 \times 10^6$ tumor cells were plated with 10 ml of RPMI (supplemented with 5% FBS) in a 10-cm dish. After 72 h, conditioned media was collected and passed through a 0.2-μm filter. Splenocytes

were then plated in a 1:1 ratio of complete RPMI:conditioned media at 400,000 cells per well in a 96-well plate. After 72 h of incubation splenocytes were treated for 5 h with activation cocktail with brefeldin A1 (Biolegend, 423303) before staining for CD45 (Biolegend, 103128), CD8b.2 (Biolegend, 140416), Granzyme B (Biolegend, 515408), IFNγ (Biolegend, 505808), and CD69 (Biolegend, 104514) according to manufacture instructions. Splenocytes were analyzed LSRII flow cytometer and data were acquired with FACSDiva software. Data were analyzed with FSC Express 7 (De Novo Software). For tumor cell co-culture experiments, $1 \times 10^4$ tumor cells were plated in a 24-well plate per well and allowed to adhere overnight. The next day, $1 \times 10^6$ stained and stimulated splenocytes were added per well. After 72 h of incubation, splenocytes were processed as described above and adherent tumor cells were concurrently collected for Annexin V (Biolegend, 640920) and 7-AAD staining (Biolegend, 420404) via flow cytometry according to manufacturer instructions.

## Flow cytometry analysis of cell surface proteins

Cells were plated at 80,000 cells/well in a 6-well plate. After two days, cells were collected by accutase (Biolegend, 423201) and analyzed by flow cytometry for H-2Kb/H-2Db/H-2Dd (Biolegend, 114613) expression according to manufacturer instructions.

## Whole transcriptome expression analysis

RNA sequencing for tumor tissue-derived EMT6-PTR, and tumor cell line-derived EMT6-PTR$^{VITRO}$ and RENCA-PDL1 cells, were performed utilizing the Genomic shared resource at RPCCC as previously described (Dolan et al, 2019). Sequencing library were prepared with TruSeq Stranded mRNA kit (Illumina Inc), from 1 μg total RNA, according to manufacturer's instructions. PolyA selection, RNA purification, fragmentation and priming for cDNA synthesis was performed. Using random primers, fragmented RNA was then reverse transcribed into first-strand cDNA. RNA template was then removed, a replacement strand was synthesized and dUTP was incorporated in place of dTTP to generate ds cDNA. ds cDNA was separated from second-strand reaction mix using AMPure XP beads (Beckman Coulter) resulting in blunt-ended cDNA. One 'A' nucleotide was added to the 3' ends of the blunt fragments. Multiple indexing adapters, containing one 'T' nucleotide on the 3' end of the adapter, were ligated to the ends of the ds cDNA, preparing them for hybridization onto a flow cell. Adapter ligated libraries were amplified by PCR, purified using Ampure XP beads, and validated for appropriate size on a 4200 TapeStation D1000 Screentape (Agilent Technologies, Inc.). The DNA libraries were quantitated using KAPA Biosystems qPCR kit, and were pooled together in an equimolar fashion, following experimental design criteria. DNA library pool was denatured and diluted to 2.4 pM with 1% PhiX control library added. The resulting pool was then loaded into the appropriate NextSeq Reagent cartridge, as determined by the number of sequencing cycles desired, and sequenced on a NextSeq500 following the manufacturer's recommended protocol (Illumina Inc.). Sequencing quality control was assessed using FASTQC v0.11.5 (http://www.bioinformatics.babraham.ac.uk/projects/fastqc/). Reads were aligned to the mouse genome GRCM38 M16 (genocode) using STAR v2.6.0a (Dobin et al, 2013) and post-alignment quality control was assessed using RSeQC v2.6.5 (Wang et al, 2012). Aligned reads were quantified using RSEM v1.3.1 (Li and Dewey, 2011). Counts from RSEM were then filtered and then upper quartile normalized using R

package edgeR. Data has been deposited to the NCBI Gene Expression Omnibus (GEO, www.ncbi.nlm.nih.gov/geo/) database for EMT6-PTR (GSE186032), EMT6-PTR^(VITRO) (GSE186034), and RENCA-PDL1 (GSE186037) cells.

## Gene Ontology analysis/Cytoscape

Differentially expressed genes with products located in extracellular regions were identified using gene ontology databases (GO:0005576) as previously described (Mastri et al, 2018a; Mastri et al, 2018b). GO Biological processes terms were then assessed using ClueGo via Cytoscape v3.7.2 and significantly enriched terms and corresponding Kappa scores were plotted based on *p*-values.

## Gene set enrichment analysis

Gene set enrichment analysis (GSEA) was conducted to assess comparisons for molecular pathways and gene set correlations. A rank list was first generated using log2(fold change) gene expression data obtained from limma analysis. GSEA-Preranked was then conducted using a gene-set permutation type with 1000 random permutations to obtain normalized enrichment scores (NES) and false discovery rate (FDR) q-values.

## Interferome analysis

Genes of interest were assessed in the Interferome Database (http://www.interferome.org/interferome/home.jspx) (Rusinova et al, 2013) to examine for evidence of regulation by IFN signaling. Parameters interferome type, subtype, treatment concentration, treatment time, in vivo/in vitro, species, system, organ, cell, cell line, normal/abnormal were set to "any", fold-change thresholds were set to 1.5.

## Identification of a αPD-L1 treatment-induced secretory (PTIS) gene signature

To identify gene signatures associated with secretory products unique to PTR tumor cells, we created a preliminary list of PTIS genes (PTIS 'preliminary') by extracting secretory genes (Appendix Table S3). Using this preliminary list, an enriched PTIS gene signature was created from comparisons of EMT6-P and EMT6-PTR using various transcriptomic and proteomic assays (Appendix Table S3; Figs. 2A–J and EV1B–D). These included individual assessment of differentially expressed secretory genes from bulk RNAseq transcriptomics data (with fold-change >2 or <−2) and proteomic comparisons of cell lysate using a proteome profile (cytokine) array. The PTIS was further enriched with targets added based on individual comparisons of EMT6-P and EMT6-PTR variants using qRT-PCR, western blot and ELISA assays. The final enriched PTIS signature was based on transcriptomic and proteomic confirmation via at least two methods. For transcriptomic comparisons of P and PTR variants (EMT6), a similar methodology was used to generate a PTIS gene signature comprised only of genes downregulated (PTIS^(DOWN); Appendix Table S3).

## CIBERSORT/ImmuCC analysis

Cibersort tissue deconvolution was performed using the ImmuCC signature to obtain absolute score for various cell types (Chen et al,

2017; Chen et al, 2018b; Newman et al, 2015). From 25 immune cell type signatures, no values were detected for eosinophil cells, CD4 Memory T cells, Neutrophil cells, and Plasma Cells; and thus, were excluded for quantification and analysis.

## Confirmation of PTIS signature enrichment in published datasets

Previously published clinical and preclinical datasets derived from studies after treatment or after acquired resistance to PD-L1 inhibition were obtained from the Gene Expression Omnibus (GEO, www.ncbi.nlm.nih.gov/geo/) or database of Genotypes and Phenotypes (dbGaP, https://www.ncbi.nlm.nih.gov/gap/).

Sceneay et al, 2019 (Data ref: Sceneay et al, 2019): In this study, whole tissue RNA-seq (Illumina NextSeq500 with paired-end 75 bp reads) were performed on 4T1 orthotopically implanted mammary tumors in 8–12 week (young; responsive) or >12 months (old, nonresponsive) Balb/c mice treated with αPD-L1 (clone 10F.9G2) or isotype (Clone LTF-2). Tumors were collected when caliper volumes were no larger than ~150 mm^3 after 3 doses of antibody treatment. Processed RNAseq data were obtained from GEO then filtered and upper quartile normalized using R package edgeR.

Lan et al, 2018 (Data ref: Lan et al, 2018): In this study, whole tissue RNAseq was (Illumina Hiseq 2500) conducted on EMT6 orthotopically implanted mammary tumors in Balb/c mice treated with αPD-L1 or isotype control. Tumors were collected 6 days after 3 doses of treatment given daily starting 20 days after implantation. Processed RNAseq data were obtained from GEO then filtered and upper quartile normalized using R package edgeR.

Efremova et al, 2018 (Data ref: Efremova et al, 2018): In this study, whole tissue RNAseq (Ion Torrent Proton) was conducted on MC38 subcutaneously implanted tumors in C57Bl/6 mice treated with αPD-L1 (Clone10F.9G2) or isotype (Clone LTF-2) every 3–4 days until day 14 after implantation. Processed RNAseq data were obtained from GEO then filtered and upper quartile normalized using R package edgeR.

Gettinger et al, 2017 (Data ref: Gettinger et al, 2017): In this study, pre-treatment and post-treatment/acquired resistant biopsies were obtained from patients receiving various immune-checkpoint inhibitor treatments (PD-L1, PD-1, CTLA-4 targeted therapies) for RNA-seq analysis (Illumina HiSeq2500) from formalin fix paraffin embedded samples. Clinical information for samples were downloaded from the dbGaP database (phs001464.v1.p1) under protocol #26165. RNAseq data was then downloaded from the Sequence Read Archive (SRA) database (accession number PRJNA412284). Data were then processed and normalized using the RNA-seq pipeline as described above, with the exception that reads were aligned to the human genome GRCH38 v27(gencode). For validation analysis, all pretreatment samples were compared to samples after acquired resistance to PD-L1 inhibition. Note: the name of the PD-L1 inhibitor was not provided in this publication.

Paulson et al, 2018 (Data ref: Paulson et al, 2018): In this study, single cell RNAseq analysis on tumor tissues were conducted on an untreated patient biopsy (Illumina HiSeq 2500) and a patient biopsy at acquired resistance (unmatched) after αPD-L1 (avelumab), MCPyV-specific T cells, and radiation (Illumina NovaSeq 6000). Processed data were obtained from the GEO database. R packages SingleCellExperiment (Amezquita et al, 2020), scater (McCarthy et al, 2017), limma (Ritchie et al, 2015), and Rtsne were used for analysis. Processed

RNAseq data were obtained from GEO, counts were quartile normalized and converted to counts per million (CPM). Clustered enrichment analysis was then conducted using markers PTPRC/CD45 (tumor cells), CD3D (T cells), and CD68 (Macrophages) similar to previously described (Paulson et al, 2018).

PTIS signature expression levels in these datasets were compared by average counts per million (CPM) levels or GSEA enrichment (defined by FDR ≤ 0.25).

## Statistical analysis

Analysis was conducted using the GraphPad Prism software package v8.4.0 (GraphPad Software Inc., San Diego, CA) and R v3.6.0 through RStudio v1.1.463 (Integrated Development for R; RStudio, Inc., Boston, MA, http://www.rstudio.com/). For in vivo studies, results are represented as mean ± standard deviation (SD) or standard error of mean (SEM), as indicated. Kaplan–Meier methods were utilized for analysis of percent to institutional endpoint curves. Fold change differences between treatment control groups were assessed via two-way ANOVA. For all results, comparisons between two groups were made with Student's two-tailed unpaired t-test, whereas one-way ANOVA was used for comparison of more than two groups. Tumor volume and bioluminescence measurements were compared for specified time points. A minimum FDR value of 0.25 was used for GSEA analysis (as indicated as described by the user guide) and significance level of 0.05 was used for all other analyses.

## Graphics

Figures 1A,G,J, 2A, 3A,B, 6F,I, 7G, EV2B,C, EV5A, and EV5E include images from Biorender.com.

# Data availability

Expression profile data generated and analyzed in this study were deposited to the NCBI Gene Expression Omnibus (GEO, www.ncbi.nlm.nih.gov/geo/) database under accession number GSE186032, GSE186034, and GSE186037.

The source data of this paper are collected in the following database record: biostudies:S-SCDT-10_1038-S44319-024-00333-0.

# Peer review information

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

## Acknowledgements

This work used shared resources supported by the Roswell Park Comprehensive Cancer Center (RPCCC) Support Grant from the National Cancer Institute (NCI) (P30CA016056). This work was supported by grants to JMLE from the American Cancer Society (ACS) via a Research Scholar Grant (RSG-18-064-01-TBG), Mission Boost Grant (MBGI-23-1038434-01-MBG), and Roswell Park Alliance Foundation (RPAF). This work was also supported by an NCI grant to YS (F30 CA243281). Opinions, interpretations, conclusions and recommendations are those of the author and are not necessarily endorsed by the RPAF, NCI, or ACS. We would like to thank K. Eng for consultations and guidance on bioinformatic analysis of single-cell RNA sequencing data; and A. Tracz for animal work assistance. EMT6 and RENCA were kind gifts from A. Gudkov and R. Pili, respectively. We thank M. Azuma for providing the MIH5 hybridoma for antibody production (see Methods) and M. Oberst at AstraZeneca for providing the Clone 80 antibody.

## Author contributions

**Yuhao Shi**: Conceptualization; Data curation; Formal analysis; Funding acquisition; Validation; Investigation; Visualization; Methodology; Writing—original draft; Writing—review and editing. **Amber McKenery**: Data curation; Formal analysis; Validation; Investigation; Visualization; Methodology; Writing—review and editing. **Melissa Dolan**: Formal analysis; Investigation; Methodology; Writing—review and editing. **Michalis Mastri**: Formal analysis; Investigation; Methodology; Writing—review and editing. **James W Hill**: Investigation; Methodology. **Adam Dommer**: Investigation. **Sebastien Benzekry**: Investigation; Methodology. **Mark Long**: Investigation; Methodology. **Scott I Abrams**: Investigation; Writing—review and editing. **Igor Puzanov**: Investigation; Writing—review and editing. **John M L Ebos**: Conceptualization; Data curation; Formal analysis; Supervision; Funding acquisition; Investigation; Visualization; Methodology; Writing—original draft; Project administration; Writing—review and editing.

Source data underlying figure panels in this paper may have individual authorship assigned. Where available, figure panel/source data authorship is listed in the following database record: biostudies:S-SCDT-10_1038-S44319-024-00333-0.

## Disclosure and competing interests statement

The authors declare no competing interests.

# Expanded View Figures

**Figure EV1.  Identification of an αPD-L1 treatment-induced secretome (PTIS) gene signature.**

(**A**) IFNAR1 expression in EMT6-P and -PTR before and after knockdown of IFNAR1 and respective vector controls. Flow cytometry, $n = 3$, statistics performed via two-tailed t-test. Bar graphs show mean ± SD. (**B**) Bar plot summary of cytokine protein array shown as EMT6-PTR compared to parental controls from Fig. 2C. Bars representing mean value of two separate experiments (shown as dots), $n = 2$. (**C**) Heatmap summary of type I IFN regulated proteins in cytokine protein array. (**D**) Table summary of PTIS transcriptomic and proteomic confirmations shown in Fig. 2B–H. Data Information: αPD-L1 Treatment-Induced Secretome (PTIS); αPD-L1 Treatment-Resistant (PTR);Counts per million (CPM); IFNAR1 knockdown (IFNAR1[KD]); Control Knockdown (shCon); Not Tested (NT). *$p \leq 0.05$, **$p \leq 0.01$, ***$p \leq 0.001$, ****$p \leq 0.0001$, for exact $p$ values see Fig. EV1 Source data. All replicates shown represent technical replicates unless otherwise specified. Source data are available online for this figure.

▶

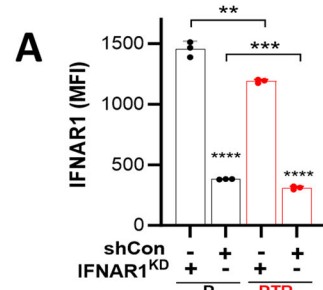

**A**

**B**

**PTR vs P**

Increased

Decreased

Below detection limits:
C-Reactive Protein/CRP
Endoglin/CD105
GM-CSF
IL-2
IL-3
IL-6
MMP-3
MMP-9
Osteoprotegerin/TNFRSF11B
Resistin

Protein Expression
(Cytokine Array)
Log2 Fold Change

**C**

Protein Expression
(Cytokine Array)
Log2 Fold Change

**D**

| PTIS | Confirmation of Increase | |
|---|---|---|
| | Transcriptomic | Proteomic |
| CCL2 | ● | ● |
| IL1RN | ● | ● |
| IL6 | ● | ● |
| LCN2 | ● | ● |
| NOS2 | ● | ● |

| Methodology | |
|---|---|
| ● | RNAseq / RT PCR |
| ● | ELISA |
| ● | Protein Array |
| ● | Western blot |

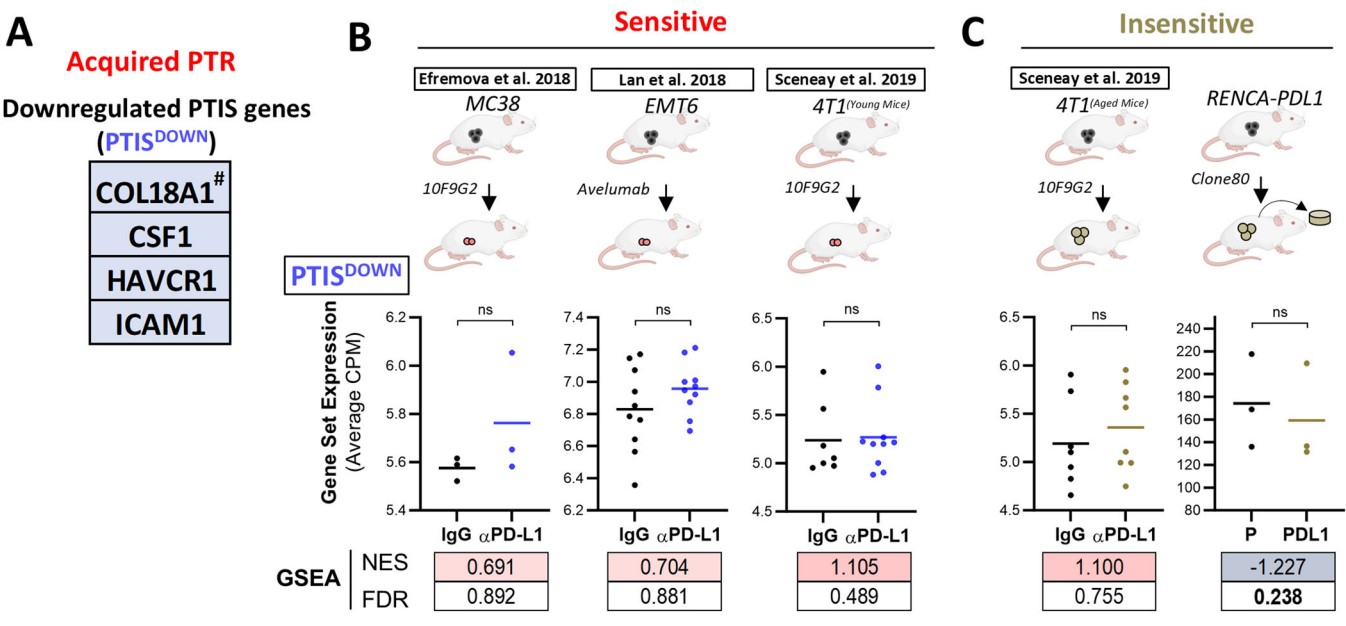

**A**

**Acquired PTR**

**Downregulated PTIS genes**
**(PTIS^DOWN)**

| COL18A1^# |
|-----------|
| CSF1 |
| HAVCR1 |
| ICAM1 |

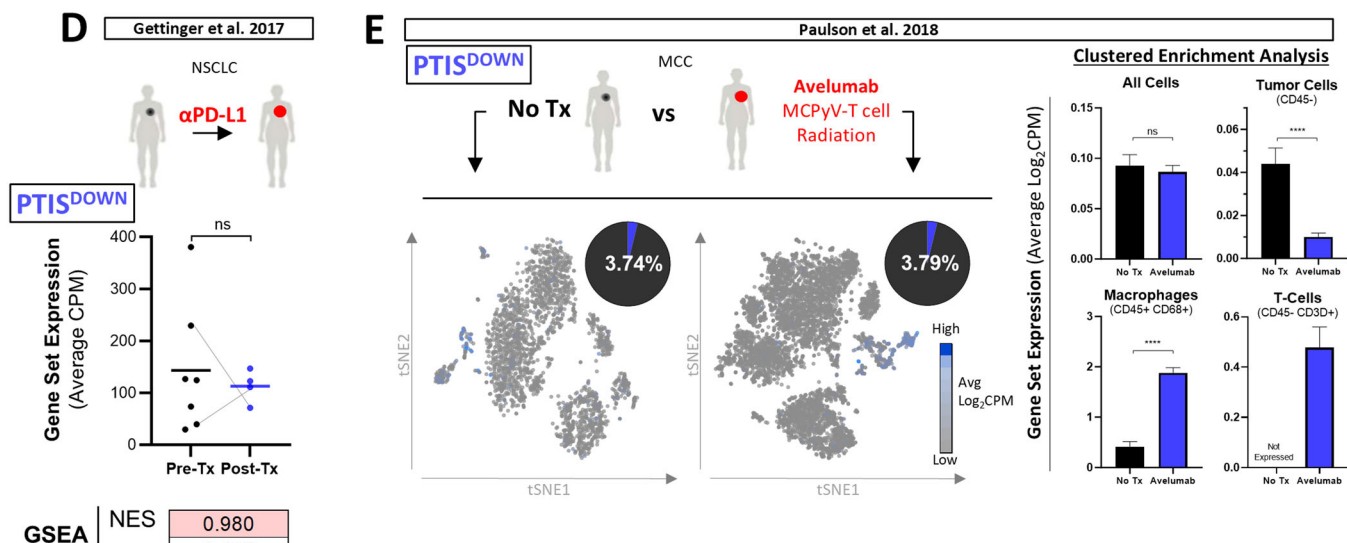

**Figure EV2. A downregulated PTIS signature (PTIS<sup>DOWN</sup>) is variably expressed in clinical and preclinical models sensitive to αPD-L1 treatment.**

(A) Transcriptome and proteomic analysis were used to generate a signature comprised only of genes significantly *down*regulated (PTIS^DOWN) in EMT6-PTR cells. The 4-gene PTIS^DOWN was comprised primarily of IFN-regulated genes (3 out of 4 total). (B, C) PTIS^DOWN was measured in published preclinical datasets described in Fig. 3B. In αPD-L1 treatment-*sensitive* and -*insensitive* tumor models, enrichment for the PTIS^DOWN was variable, with no significant signature expression trends found (as measured by CPM). Only 1 of 5 models (*insensitive;* RENCA-PDL1) showed significant negative GSEA enrichment of the PTIS^DOWN signature. *P* values representative of statistics performed via two-tailed t-test. (B) Analysis of PTIS^DOWN expression in αPD-L1 treatment-*sensitive* tumor models used data from (Lan et al, 2018; Sceneay et al, 2019; Efremova et al, 2018) (Data ref: Lan et al, 2018; Data ref: Sceneay et al, 2019; Data ref: Efremova et al, 2018). Data is compared to vehicle/IgG-treated controls. Central line depicts the median, *n* = 3–10 biological replicates. (C) Analysis of PTIS^DOWN expression in αPD-L1 treatment-*insensitive* tumor models used data from (Sceneay et al, 2019) (Data ref: Sceneay et al, 2019) and RENCA-PDL1 model from this study. Central line depicts the median, *n* = 3–8 biological replicates. (D, E) Using published bulk and single cell RNAseq clinical datasets taken from tumor biopsies of αPD-L1 treatment-*sensitive* patients, PTIS^DOWN expression was assessed by average CPM expression and GSEA comparisons. *P* values representative of statistics performed via two-tailed t-test. (D) Analysis of PTIS^DOWN signature showed decreases in NSCLC after treatment significance not reached). Bulk RNAseq from (Gettinger et al, 2017) (Data ref: Gettinger et al, 2017) to compare Pre- and Post-treatment (Tx) tumor sample comparisons (Gray lines indicate matched Pre- and Post-tx samples). Central line depicts the median, *n* = 4–7 biological replicates. (E) Clustered analysis of MCC avelumab-treated samples showed variable, but significant PTIS^DOWN changes in enriched tumor cell (decreased expression) and macrophage clusters (increased expression). Single-cell RNAseq data obtained from (Paulson et al, 2018) (Data ref: Paulson et al, 2018) represent untreated (No-Tx) or treated (avelumab, MCPyV-T cell, radiation) tumor samples (*n* = 1) with tSNE plots (left) representing average log₂CPM expression of PTIS in whole dataset, and bar graphs (right) representing clustered enrichment analysis populations identified by markers for tumors (CD45−), macrophages (CD68+), and T cells (CD3D+). Tumor sample that received No-Tx was compared to treated. Data Information: αPD-L1 Treatment-Induced Secretome involving only downregulated genes (PTIS^DOWN); αPD-L1 Treatment-Resistant (PTR); Counts per million (CPM); Gene set enrichment analysis (GSEA); False Discovery Rate (FDR); Gene Expression Omnibus (GEO); GEO Series records (GSE); database of Genotypes and Phenotypes (dbGaP); t-distributed stochastic neighbor embedding (tSNE); Treatment (Tx); non-small cell lung carcinoma (NSCLC); Merkel cell carcinoma (MCC). Bar graphs show mean ± SEM. Scatter dot plot central line show mean. Bolded numbers for GSEA represent FDR < 0.25 (see Methods). *$p \leq 0.05$, **$p \leq 0.01$, ***$p \leq 0.001$, ****$p \leq 0.0001$, except for GSEA as indicated above, for exact *p* values see Fig. EV2 Source data. Source data are available online for this figure.

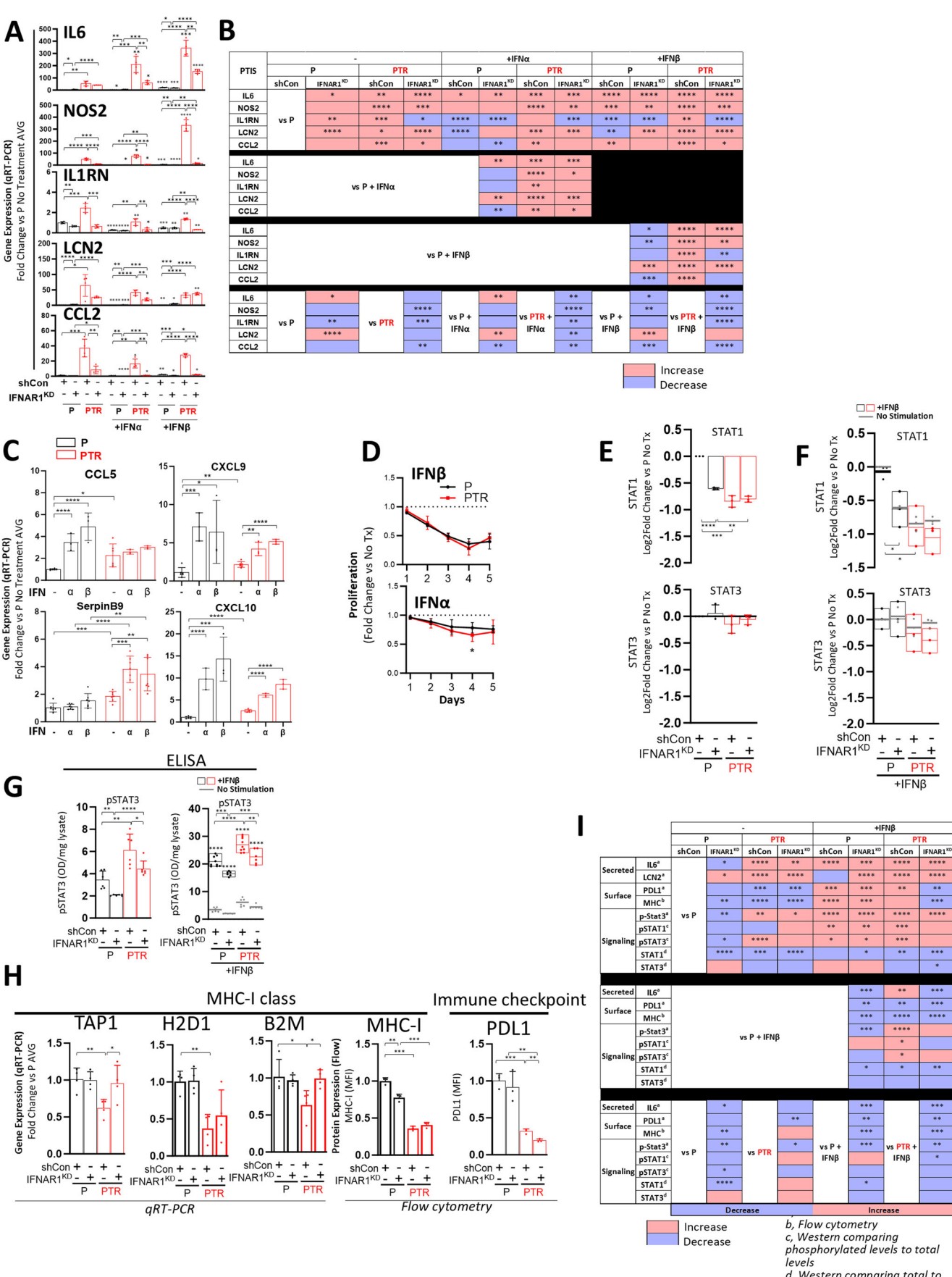

**Figure EV3.   IFNs modulate PTIS, STAT signaling, and surface ISGs in PTR cells.**

(A) PTIS factor expression in EMT6-P and -PTR cells before and after knockdown of IFNAR1[KD], and after IFN α/β stimulation shown as bar plots. Full datasets for Fig. 4B, C, D. qRT-PCR, statistics performed via two-tailed t-test, $n = 4$. (B) Table summarizing two-tailed t-test statistical comparisons for qRT-PCR data shown in Fig. 4B, C, D. (C) ISG expression in EMT6-P and -PTR cells after IFN α/β stimulation shown as bar plots. qRT-PCR, statistics performed via two-tailed t-test, $n = 3–7$ (1–2 biological replicates). (D) EMT6-P and PTR were treated with IFNα (left), and IFNβ (right) for 5 days and proliferation measured daily by MTS, statistics performed via two-tailed t-test, $n = 10$. (E, F) Densitometry quantification of western blots shown in (Fig. 4F) representing relative (top) total STAT1 and (bottom) total STAT3 levels compared to total β-Actin levels (E) at baseline and (F) after IFNβ stimulation. Statistics performed via two-tailed t-test. Boxes indicate range between minimum and maximum, central line depicts the mean, $n = 3$. (G) pSTAT3 expression in lysates of EMT6-P and -PTR before and after knockdown of IFNAR1[KD], and after IFNβ stimulation. ELISA, statistics performed via two-tailed t-test. Boxes indicate range between minimum and maximum, central line depicts the mean, $n = 6–9$. (H) Antigen presentation machinery expression in EMT6-P and -PTR cells before and after knockdown of IFNAR1[KD] shown as bar plots. Full dataset for Fig. 4I, J. qRT-PCR and flow cytometry as indicated, statistics performed via two-tailed t-test, $n = 3–4$. (I) Table summarizing two-tailed t-test statistical comparisons for data shown in Figs. 4 and EV3. Superscripts: a—ELISA, b—Flow cytometry, c—western comparing phosphorylated levels to total levels, d—western comparing total to β-Actin levels. Data Information: Parental (P); αPD-L1 Treatment-Resistant (PTR); Conditioned Media (CM); Mean Fluorescent Intensity (MFI); IFN stimulated genes (ISGs); IFNAR1 knockdown (IFNAR1[KD]); shRNA vector control (shCon); Cells were treated with 10 ng/ml of IFNs and collected after 15 min for STAT1/3 westerns and ELISA; for proliferation experiments cells were treated with 10 ng/ml of IFNs starting on day 0 and treatment and fresh media was replaced on day 3. Bar graphs and line graphs show mean ± SD. Box plots indicate range between minimum and maximum, central line depicts the mean. *$p \le 0.05$, **$p \le 0.01$, ***$p \le 0.001$, ****$p \le 0.0001$ indicate significance compared untreated controls unless otherwise shown (lines), for exact *p* values see Fig. EV3 Source data. All replicates shown represent technical replicates unless otherwise specified. Source data are available online for this figure.

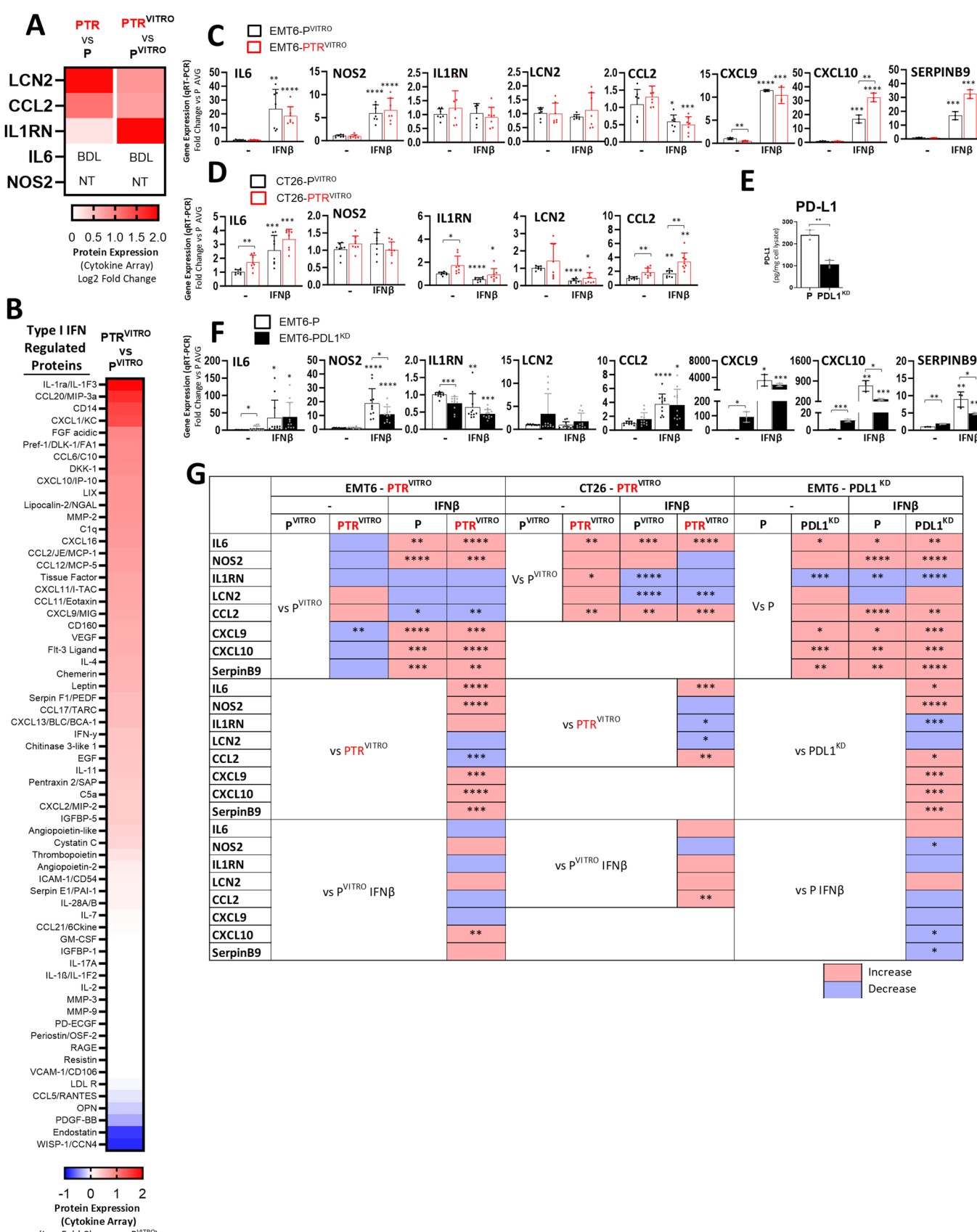

◀　**Figure EV4. PTIS expression in PTR^VITRO and PD-L1^KD models.**

(A) Heatmap summary of PTIS factors in cytokine protein array for EMT6-PTR cells relative to -P controls (Fig. 2A) and EMT6-PTR^VITRO cells relative to -P^VITRO controls (Fig. 5A). (B) Heatmap summary of type I IFN regulated protein changes in cytokine protein array for EMT6-PTR cells relative to -P controls (Fig. 5A) and EMT6-PTR^VITRO cells relative to -P^VITRO controls (Fig. 2A). (C) PTIS and various ISG expression in EMT6-P^VITRO and -PTR^VITRO cells treated with IFNβ shown as relative to P controls and represented as bar graphs. Full dataset for Fig. 5F. qRT-PCR, statistics performed via two-tailed t-test, $n = 3$–7 (1–2 biological replicates). (D) PTIS expression in CT26-P^VITRO and -PTR^VITRO cells treated with IFNβ shown as relative to P controls and represented as bar graphs. Full dataset for Fig. 5G. qRT-PCR, statistics performed via two-tailed t-test, $n = 8$ (2 biological replicates). (E) PD-L1 expression in cell lysates of EMT6-P and EMT6–PD-L1^KD cells. ELISA, statistics performed via two-tailed t-test, $n = 3$. (F) PTIS and various ISG expression in EMT6-shCon and -PDL1^KD cells treated with IFNβ shown as relative to P controls and represented as bar graphs. Full dataset for Fig. 5H. qRT-PCR, statistics performed via two-tailed t-test, $n = 3$–11 (1–3 biological replicates). (G) Tables summarizing two-tailed t-test statistical comparisons for qRT-PCR data shown in Fig. 5E, F, G, EV4C, EV4D, and EV4F. Data Information: Parental (P); αPD-L1 Treatment-Resistant (PTR); PD-L1 knockdown (PDL1^KD) Bar graphs show mean ± SD. *$p \leq 0.05$, **$p \leq 0.01$, ***$p \leq 0.001$, ****$p \leq 0.0001$ compared to vector controls unless noted otherwise, for exact $p$ values see Fig. EV4 Source data. All replicates shown represent technical replicates unless otherwise specified. Source data are available online for this figure.

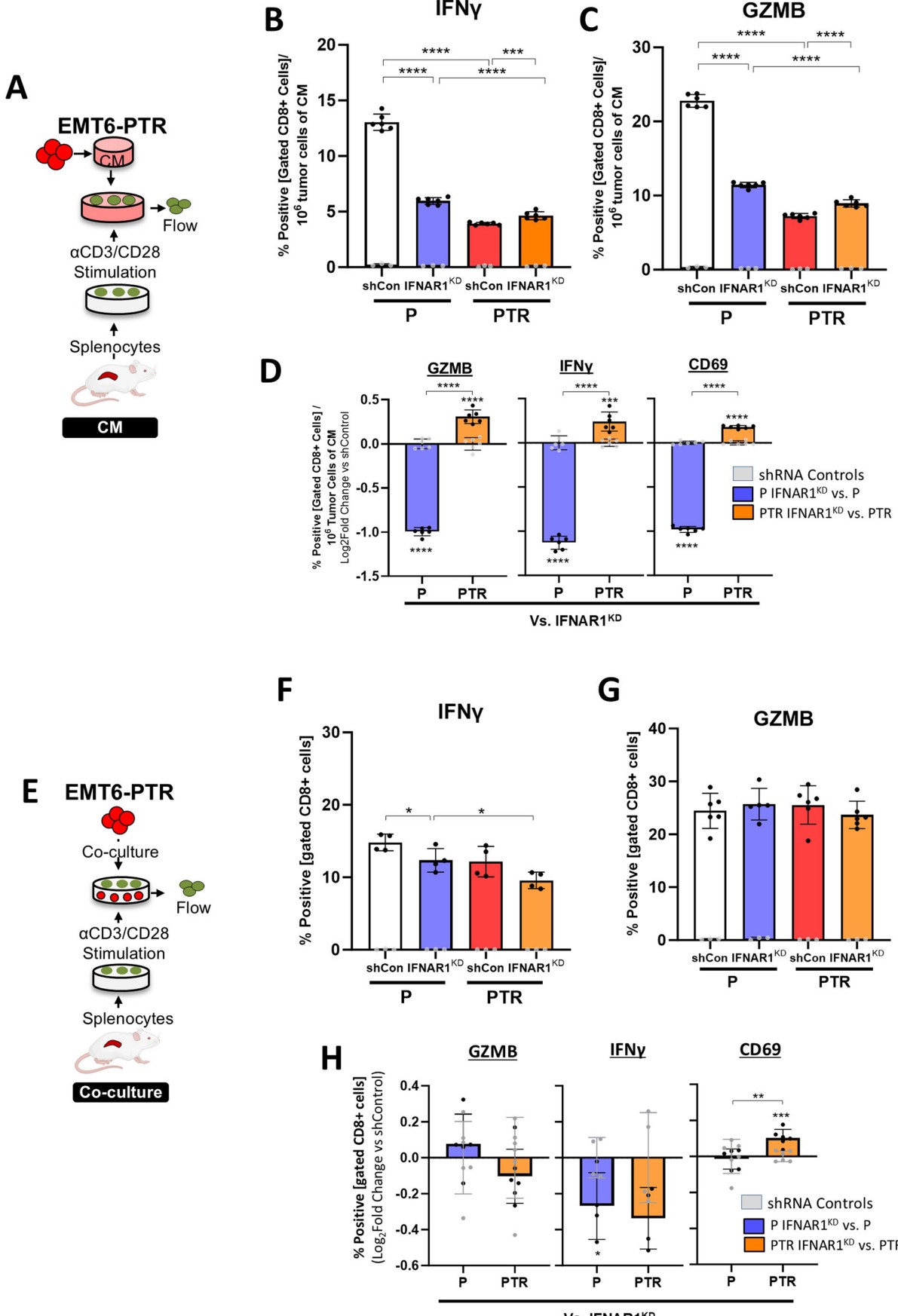

◀ **Figure EV5. Splenocyte activation following incubation with PTR cells and conditioned media.**

(A) Schematic of Balb/c-derived splenocyte proliferation and activation following incubation EMT6-P and -PTR CM for experiments in (B–E). (B, C) CD8+ splenocyte activation marker (B) IFNγ and (C) Granzyme B after co-incubation with CM derived from EMT6-P and -PTR control and respective IFNAR1$^{KD}$ variants. Flow cytometry, statistics performed via two-tailed t-test, $n = 3$–6. (D) Log$_2$ fold change analysis of CD8+ splenocyte activation markers (Granzyme B, IFNγ, CD69) after co-incubation with EMT6-P and -PTR-IFNAR1$^{KD}$ variant CM compared to respective controls. Flow cytometry, statistics performed via two-tailed t-test, $n = 3$–6. (E) Schematic of Balb/c-derived splenocyte proliferation and activation following incubation EMT6-P and -PTR CM for experiments in (F–H). (F, G) CD8+ splenocyte activation marker (F) IFNγ (G) Granzyme B after co-culture with EMT6-P and -PTR control and respective IFNAR1$^{KD}$ variants. Flow cytometry, statistics performed via two-tailed t-test, $n = 3$–6. (H) Log$_2$ fold change analysis of CD8+ splenocyte activation markers (Granzyme B, IFNγ, CD69) after co-culture with EMT6-P and -PTR-IFNAR1$^{KD}$ variant compared to respective controls, statistics performed via two-tailed t-test, $n = 4$–6. Data Information: Parental (P); αPD-L1 Treatment-Resistant (PTR); IFNAR1 knockdown (IFNAR1$^{KD}$); Conditioned Media (CM); Granzyme B (GZMB). Bar graphs show mean ± SD. *$p \leq 0.05$, **$p \leq 0.01$, ***$p \leq 0.001$, ****$p \leq 0.0001$ compared to vector controls unless noted otherwise, for exact $p$ values see Fig. EV5 Source data. All replicates shown represent technical replicates unless otherwise specified. Source data are available online for this figure.

