## [Peer Review File · EMBO Reports]

Acquired resistance to PD-L1 inhibition enhances a type I IFN-regulated secretory program in tumors

Yuhao Shi, Amber McKenry, Melissa Dolan, Michalis Mastro, James Hill, Adam Dommer, Sebastien Benzekry, Mark Long, Scott Abrams, Igor Puzanov, and John Ebos

Corresponding author(s): John Ebos (john.ebos@roswellpark.org)

Review Timeline:

Submission Date:	26th Apr 23
Editorial Decision:	1st Jun 23
Appeal Received:	24th Aug 24
Editorial Decision:	26th Sep 24
Revision Received:	21st Oct 24
Editorial Decision:	28th Oct 24
Revision Received:	10th Nov 24
Accepted:	14th Nov 24

Editor: Achim Breiling

Transaction Report:

Dear Dr. Ebos,

Thank you for the submission of your research manuscript to EMBO reports. I have now received the full set of referee reports that is copied below.

I am sorry to say that the decision on your manuscript is not a positive one. As you will see, referees #2 and #3 state that the conclusions are not fully supported by the data and that no convincing evidence is presented that secretion of endogenous IFN is induced by treatment with anti PD-L1, indicating that the study should not be published in EMBO Reports. Referee #1 is more positive, but has also several concerns, and seems to question the generality of the findings. Moreover, all referees mention technical shortcomings.

Given these comments, also indicating that the study does not provide the advance and broader impact we are looking for, and considering the amount of work required to address them, and finally the fact that EMBO reports can only invite revision of papers that receive strong support from the referees upon initial assessment, I cannot offer to publish your manuscript.

I am sorry to have to disappoint you this time. I nevertheless hope that the referee comments will be helpful in your continued work in this area, and I thank you once more for your interest in our journal.

Yours sincerely

Referee #1:

Shi et al investigate potential mechanisms underlying acquired resistance to anti-PD-L1 therapy. The overarching thesis is that prolonged anti-PD-L1 therapy can rewire type I IFN signaling that then impacts efficacy of immune checkpoint inhibition. The authors focus on the therapy-induced secretome from tumors that results from anti-PD-L1. They first demonstrate that EMT6 tumors can acquire resistance to anti-PD-L1 in vivo and subsequently perform RNAseq from parental (control-treated) or 'resistant' (PDR, growing in the face of anti-PD-L1) tumors. This analysis point them to look at the secretome of the PDR tumors.

They focus on genes regulated by IFNs. They identify 12 factors/genes elevated by anti-PD-L1 (PTIS secretome) and then use this gene signature to probe other data sets. They find that tumors that initially show sensitivity to anti-PD-L1 show an elevation in the PTIS secretome whereas tumors that are intrinsically insensitive to anti-PD-L1 do not. They provide some validation of this using data sets from human patients treated with anti-PD-L1. They investigate the function of IFNAR1 expressed on tumor cells in the phenotype and provide data supporting that at some of the effects observed are due to IFNAR1 expression. In this regard the authors find that loss of IFNAR1 in PDR cells slows tumor growth in vivo, an effect that is stark contrast to the effect of loss of IFNAR1 in parental cells, where it increases tumor growth in vivo. They show similar albeit not as striking effects with the inhibition of IL6. Overall the study provides an overview of the complexity of tumor cell and immune cell response to anti-PD-L1 therapy. Additionally, the study provides a rationale for targeting type I IFN signaling in tumors that show acquired resistance to anti-PD-L1. Although it is not clear how wide spread this phenomena is, yet the authors do provide data in multiple syngenic mouse models.

The introduction and discussion frame the work appropriately. The data are displayed clearly. The study is focused on complex biology and the authors do a good job at providing context.

Comments:

1. Do PDR cells fail to respond to anti-PD-L1 (or anti-PD-1) in subsequent in vivo studies? Meaning once the cells are collected from the original tumor/animal and then passed in culture do these cells maintain a resistant phenotype in subsequent in vivo studies? Is resistance a heritable change?
2. What is the make up of the immune landscape of parental EMT6 tumors vs PDR tumors?
3. Is there evidence of elevated Stat1/3 or JAK signaling in tumor cells in PDR tumors treated with anti-PD-L1 compared to parental tumors or anti-PD-L1 responsive tumors?
4. Results presented in Fig 4I-K should be clarified and/or discussed further. How do PDR cells increase CD8 T cell proliferation and activation (CD69, GZMB, IFNg) via direct contact compared to parental clones?

5. Is the effect in Fig 5A, with loss of IFNAR in PDR cells slowing tumor growth dependent on T cells?

Referee #2:

The authors observe regulation of the expression of secretory proteins (α PD-L1 treatment-induced secretome or PTIS) in response to anti PD-L1 in cancer cells sensitive or resistant to anti PD-L1. The authors claim that PTIS expression is dependent on IFN-I signaling and that inhibition of PTIS by disrupting IFN-I signaling inhibits the growth of anti PD-L1 resistant tumors. The manuscript presents data inconsistent with this claim without sufficient explanation and therefore the conclusions are not well founded. No convincing evidence is presented that secretion of endogenous IFN is induced by treatment with anti PD-L1.

1. In Fig 3e, IL-6 protein level was not affected by IFNAR KD in the absence of treatment with exogenous IFN β , suggesting that endogenous IFN-I signaling plays at most a minor role in increasing IL-6 expression in PDR cells compared to P cells.
2. In Fig 3f, in the absence of exogenous IFN β treatment or IFNAR1 KD, p-STAT1 levels are decreased, but pSTAT3 levels are increased in PDR cells compared to P cells. This result indicates that elevated PTIS levels in PDR cells, in the absence of treatment with exogenous IFN β , are mainly driven by pSTAT3, not pSTAT1, suggesting that other cytokines that exclusively induce pSTAT3, not pSTAT1, (eg. IL-6) may be responsible for increased PTIS expression in PDR cells.
3. PD-L1 expression was decreased in PDR cells (Fig 3i), but T cell mediated cell death was decreased in PDR cells (Fig 4d). This result is inconsistent with the ability of PD-L1 to inhibit cancer cell killing by T cells.
4. Fig 5a shows that the growth of PDR cells was significantly decreased when IFNAR1 was knocked down, and the authors claim that inhibition of PTIS regulators (based on their assumption that IFN-I signaling is responsible for PTIS expression) inhibits PDR tumor growth. This paper does not present sufficient data showing IFN-I signaling is responsible for PTIS expression. Fig 3 data suggest that PTIS expression in PDR cells may be induced through other cytokines that activate STAT3, not STAT1. Therefore, the effect of IFNAR1 KD in Fig5a cannot be due to blockade of PTIS expression.
5. "PD-L1 treatment-induced secretome (PTIS)" in the abstract should read anti-PD-L1.
6. In EMT6-P cells, IFN β increased IL-6 protein levels (Fig 3c and e), but not IL-6 mRNA levels. Further discussion about this difference is needed.

Referee #3:

In this study the authors establish a PD-L1 EMT6 resistant tumor cell line that initially responds but progresses to anti-PDL1 treatment. The results are a comparison between PDR tumors and parental (untreated tumors). The authors establish that upregulation of IFN response genes or tumor IFN responsiveness after anti-PDL1 tumor immunotherapy results in acquired resistance. While resistance due to chronic IFN signaling is known, the IFN regulated program that separates response vs resistance in tumors was not established in this manuscript. The authors identify an interesting finding that IFNAR1 KD results in a decrease of PDR tumor growth however mechanisms of these results were not investigated.

Resistance mechanisms are drawn between responders and non-responders to immunotherapy. However, the authors perform a differential analysis between parental (untreated) tumors and tumors that initially respond but progress after anti-PD-L1 therapy. The authors state that reason of acquired resistance is upregulation of IFN response genes after anti PD-L1 treatment. But, upregulation of IFN response genes can be an outcome of an increase in IFN responsiveness of the tumor after anti-PD-L1 treatment which can also result in a delay in tumor growth compared to the parental tumor.

The authors state that treatment of PDR to IFN-A, B, Y results in an increase of PTIS (PD-L1 treatment-induced secretome) program in the tumors as indicative of acquired resistance. Here, CXCL9 and CXCL10 although within PTIS program are downregulated or not significant.

The authors noted a decrease in tumor cell death of PDR tumors vs control tumors however the mechanism behind the cell death decrease was not investigated. An association between upregulation of IFN response genes and decrease in tumor cell death needs to be established.

The authors observe a decrease in PDR tumor growth on IFNAR1 KD. Here, the resistance program within the IFN response genes, the knockdown of which results in a decrease of tumor growth should be defined.

** As a service to authors, EMBO Press provides authors with the ability to transfer a manuscript that one journal cannot offer to publish to another journal, without the author having to upload the manuscript data again. To transfer your manuscript to another EMBO Press journal using this service, please click on
Link Not Available

INTRODUCTION TO RESUBMISSION

This study was originally submitted to EMBO Reports and received reviews indicating support for the work; however, there were issues noted about the strength of the experimental data and narrative presented. In some cases, there were concerns raised that we feel stemmed from a lack of clarity in our manuscript message, something that we have now corrected with major revisions. Here we have attached our revised manuscript that includes extensive new datasets, a rearranged format, and significant improvements to the overall study conclusions. Below we have provided a point-by-point response to all comments and an introductory overview section that details the new/modified Figures. A comparison of the revised/original paper structure is shown in **Summary Fig 1 below** and in a **Reviewer Appendix** (attached at end). This manuscript now aligns with the EMBO formatting with the inclusion of Figures, Extended View (EV) Figures, and an Appendix section.

MANUSCRIPT CHANGES:

New Data added: The primary novelty of our study is to show that acquired resistance to anti-PDL1 (α PDL1) antibodies can drive tumor-intrinsic changes to the interferon (IFN) signaling machinery that, in turn, can control the production of secreted proteins that we link to immune-suppressing phenotypes. Our study includes the creation and use of mouse models of α PD-L1 Treatment-Resistance (PTR) that we use to identify secretory programs regulated by type I IFN-signaling. Using PTR models, we identified a α PD-L1 Treatment-Induced Secretome (PTIS) gene score that was unique to models of *acquired* resistance (i.e., that they derive from treatment-*sensitive* tumors). We then validated the PTIS in several preclinical/clinical datasets involving α PD-L1 therapy in treatment-sensitive settings (i.e, in responsive cancer patients or mouse tumors models). A key finding was that the PTIS can be *regulated* by type I IFN signaling and, when the intracellular IFN machinery is disrupted, we show that the immune-suppressing phenotypes of PTR cells are reduced, and tumor growth dramatically decreased when reimplanted into mice.

In our original manuscript we realize that several of these messages were not clear, and reviewers raised numerous questions about the novelty of these claims and the consistency of the data. We agreed with these sentiments and therefore undertook extensive studies to improve the narrative focus. Changes included the following:

- 1) **A modified α PDL1-treatment induced secretome (PTIS) gene score:** In our original manuscript, we described a 12-gene PTIS score; however, Reviewers 2 & 3 noted concerns that related directly/indirectly to our lack of detailed validation and/or sufficient explanation of instances where models had divergent/inconsistent results. To address this, we re-analyzed the original PTR RNAseq datasets and identified a condensed 5-gene score (a subset of the original 12) that provided a much more robust and consistent result in our validation studies. Using this revised/optimized PTIS gene score, we re-structured the paper to include expanded testing/validations of these 5 genes in multiple settings. For instance, in a new Fig 2A-L and new Fig EV1A-C, we now show extensive transcriptomic and proteomic studies confirming the PTIS signature (**Summary Fig 2 below shows a schematic detailing the process**). We also re-analyzed numerous published RNAseq (bulk and single-cell) datasets involving mouse and human (clinical) samples to show that the PTIS score consistently increases in α PD-L1 treatment-*sensitive* models. These are shown in a revised Fig 3A-E and Fig EV2A-E of our paper.

2) **Type I IFN-signaling regulates the PTIS:** Related to this, our revised manuscript now contains extensive new data showing that the 5-gene PTIS can be regulated by type I IFNs, something we demonstrated using various forms of manipulation of the IFN-signaling cellular machinery. This includes the use of IFNAR1-expression knock-down (IFNAR1^{KD}) cell variants (found to limit the PTIS) or exogenous stimulations with recombinant IFN α/β (found to enhance the PTIS). In **Summary Fig 3** we show some excerpts of this new data, with the rest of the revisions/additions described in more detail in reviewer responses below.

3) **Intrinsic PD-L1 signaling only partially controls PTIS:** As part of our original manuscript, we included investigations into whether PD-L1 itself might regulate the PTIS. This is because PD-L1, though well-known as a cell-surface protein that can mediate anti-tumor immunity by binding to PD-1 on T-cells, also has several less well-known tumor-intrinsic cellular functions linked to IFN signaling. In our revised studies, we included expanded *in vitro* experiments that test whether chronic/long-term α PD-L1 antibody exposure or PD-L1 knock-down (PDL1^{KD}) can mimic the type I IFN-regulated secretory changes that we observed in our *in vivo* models. In a new Fig 5A-J, we now show that disruption of PD-L1 can drive increases in some, but not all,

of the PTIS factors. This finding suggests that the secretory changes we observe in tumors derived *in vivo* depend, at least to some extent, on extrinsic host responses to treatment. However, our results also show that there is clearly a tumor-intrinsic component driving the PTIS that is dependent on PD-L1 protein signaling, even if it is not robust across all of the systems we tested. While our results do not definitively pinpoint whether the adaptive PTIS is a tumor- or host-derived mechanism, our findings suggest that there is a dynamic interplay between tumor:host that cause this effect. In our revised manuscript, we suggest that these results highlight the importance of using (and further developing) *in vivo*-based mouse models to evaluate/mimic the complex microenvironment pressures on tumor cells during cancer progression as it occurs in patients. Importantly, our study now includes expanded/new data that uses protein array assays, multiple tumor cell lines, and different PD-L1 antibodies to show that several additional secreted ISGs (i.e., non-PTIS) can also be increased in *in vivo* model systems. This data is partially shown in **Summary Fig 4 below**; in Figs 5A-J and Fig EV4A-G in the manuscript; and is explained further in comments to reviewers (see response to 1.2 below).

Revised Narrative focus: Finally, for all the new data introduced in our revised study, perhaps our most significant improvement was to clarify the underlying conclusions of the study. For instance, Reviewers 2 and 3 raised questions that related to topics we feel were the result of a misunderstanding of our original work which, in turn, was due to our confusing messaging. This included questions related to whether autocrine PTIS signaling may be fueling (and confounding) our signaling studies, whether type I IFNs are increased in our resistant models, and whether the PTIS is a cause or consequence of resistance overall. In our revised manuscript, we improved the study narrative with new data that addresses these questions and several others that were raised. These topics are discussed below.

***NOTE TO REVIEWERS: we originally used the term 'PDR' (α PD-L1 drug resistant) to describe selected cells but have changed it to 'PTR' (α PD-L1 treatment resistant) for simplicity.**

Referee #1:

Shi et al investigate potential mechanisms underlying acquired resistance to anti-PD-L1 therapy. The overarching thesis is that prolonged anti-PD-L1 therapy can rewire type I IFN signaling that then impacts efficacy of immune checkpoint inhibition. The authors focus on the therapy-induced secretome from tumors that results from anti-PD-L1. They first demonstrate that EMT6 tumors can acquire resistance to anti-PD-L1 *in vivo* and subsequently perform RNAseq from parental (control-treated) or 'resistant' (PTR, growing in the face of anti-PD-L1) tumors. This analysis point them to look at the secretome of the PTR tumors.

They focus on genes regulated by IFNs. They identify 12 factors/genes elevated by anti-PD-L1 (PTIS secretome) and then use this gene signature to probe other data sets. They find that tumors that initially show sensitivity to anti-PD-L1 show an elevation in the PTIS secretome whereas tumors that are intrinsically insensitive to anti-PD-L1 do not. They provide some validation of this using data sets from human patients treated with anti-PD-L1. They investigate the function of IFNAR1 expressed on tumor cells in the phenotype and provide data supporting that at some of the effects observed are due to IFNAR1 expression. In this regard the authors find that loss of IFNAR1 in PTR cells slows tumor growth *in vivo*, an effect that is stark contrast to the effect of loss of IFNAR1 in parental cells, where it increases tumor growth *in vivo*. They show similar albeit not as striking effects with the inhibition of IL6. Overall the study provides an overview of the complexity of tumor cell and immune cell response to anti-PD-L1 therapy. Additionally, the study provides a rationale for targeting type I IFN signaling in tumors that show acquired resistance to anti-PD-L1. Although it is not clear how wide spread this phenomena is, yet the authors do provide data in multiple syngeneic mouse models.

The introduction and discussion frame the work appropriately. The data are displayed clearly. The study is focused on complex biology and the authors do a good job at providing context.

Comments:

1.1 Do PTR cells fail to respond to anti-PD-L1 (or anti-PD-1) in subsequent *in vivo* studies? Meaning once the cells are collected from the original tumor/animal and then passed in culture do these cells maintain a resistant phenotype in subsequent *in vivo* studies? Is resistance a heritable change?

We thank the reviewer for these positive comments and for raising this question about whether treatment failure might derive from adaptations in the tumor, the host, or be a result of a concerted process related to both together. This is important because, if *in vivo*-selected PTR tumor cells are non-responsive to α PDL1 antibody upon reimplantation into treatment-naïve mice, then this would suggest resistance derives from a tumor-intrinsic mechanism. Conversely, if reimplanted PTR tumors are again α PDL1 treatment sensitive, then it would suggest host-adaptations are responsible for resistance. Though these are important and interesting studies to undertake, we did not include reimplantation studies here for a few reasons. **First**, the gene signature we developed is based on a single treatment cycle of response-relapse (acquired resistance) as it would occur in patients. Using the treatment-sensitive EMT6 cell line, we identified an IFN-regulated secretory signature and then validated using published clinical/preclinical datasets that were not retreated/retested for sensitivity after selection. In our limited study, such reimplantation studies would not immediately impact our study conclusions and would be best conducted in a follow-up study where the PTIS could be evaluated as a therapeutic target in more detail. On this topic, it is interesting to note that re-implantation studies involving acquired resistance to α PD(L)1 treatments are surprisingly rare. We are currently performing an extensive literature analysis for a review publication where we surveyed over 1000 preclinical studies involving acquired resistance modeling to ICIs and we have found that re-testing of tumor cells after selection was rarely done (i.e, less than 5% of cases). We hope that our review highlights this topic for others as there is a need to harmonize how acquired resistance is defined in mouse models, similar to the ongoing questions occurring in the clinical setting (described in (1, 2)). **Second**, and equally important, we have not re-implanted our selected tumor cells (or whole tissues) because it remains unclear whether the IFN/secretory changes are a cause or consequence of acquired resistance. In this regard, we understand and agree that our initial submission was not clear in its presentation of our data. If presented as a *cause*, then we agree reimplantation studies would provide context to the mechanistic role in

resistance. However, while we do investigate some potential causal reasons why PTR tumors may be resistant to α PD-L1 treatment if reimplanted (i.e., decreased ability for tumor cells to activate T-cells, shown in **Manuscript Figure 6**), we feel our study novelty is primarily related to the *consequence* of acquired resistance. What we mean is that the crux of our data shows that this resistance confers an unexpected permanent alteration in how tumors respond to IFN-activation in post-resistant settings. For instance, we found that the ISG-enriched secretory signature found in PTR cells is involved in T cell cytotoxicity *ex vivo* and tumor growth *in vivo*, and this is sensitive to IFN blockade and stimulation. As a consequence of treatment, these changes suggest that tumor cells are 'rewired' with this immune suppressive secretory program and enhanced by exogenous type I IFN during treatment-relapse that, at least theoretically, derives from the local microenvironment. While our data strongly suggests that the mechanism we uncovered could be a cause of resistance, more studies are needed to prove this more concretely. In the meantime, our work represents an important first step in identifying the secretory profiles that could be used as a biomarker and starting point for more expanded therapeutic study. For instance, we are currently developing treatment combinations that aim to block the PTIS directly (i.e, with multiple antibodies or even a vaccine) and that aim to target the IFN regulating machinery, such as the JAK pathway or the STING pathway. We have added some of these points to an expanded discussion (see below) but feel that these are beyond the scope of this current study. We hope this will suffice and thank you for raising this excellent and important topic.

[Modified Discussion Section]

'.....By examining tumor cells after long-term drug treatment, the results of our study identify PTIS as a consequence of α PD-L1 resistance. Yet, one question raised by our findings is whether PTIS can be a cause of resistance. Interestingly, the role of several PTIS proteins may largely be context-dependent, as several have been linked to tumor-promoting and tumor-inhibiting immune responses. An example includes CCL2 and IL6, which we found to be increased in PTR cells and, thus far, have been associated with immune suppressive effects including promoting T regulatory cells(3) and myeloid-derived suppressor cell populations(4) respectively. Furthermore, combination regimens of IL6 and CCL2 inhibition with ICIs have shown to delay development of resistance in preclinical studies (5, 6). But in other tumor models, these cytokines have been shown to have anti-tumor effects including proliferation, recruitment and trafficking of cytotoxic T cells(7-10). In this regard, it is also of particular interest that the PTIS was found to be enriched in non-tumor populations such as macrophages and T cells of Merkel cell carcinoma patients treated with avelumab (Fig 3E) - suggesting PD-L1 inhibition can likely induce 'off-target' host effects that can have negative effects on treatment efficacy(11). Using CIBERSORT tissue deconvolution of PTR tissue, we also show decreased activated T cell scores that suggest the presence of potential immune suppressive mechanisms in the tumor microenvironment. Furthermore, PTIS factors such as IL6 and IL1-RA have been shown to be associated with immune related adverse events (irAEs) known to be induced by various ICI treatments(12). PD-1 targeting agents have also been found to induce cytokine release syndrome, which is an adverse effect characterized by fever, myalgias, malaise, and high levels of cytokines including IL6 and IFNs which can be a barrier for continued treatment in patients(13, 14). Assessment of the PTIS as a potential cause of resistance deriving from the tumor, host, or both would require further investigation.....'

1.2. What is the make up of the immune landscape of parental EMT6 tumors vs PTR tumors?

We thank the reviewer for this question. While our study is focused on the tumor-intrinsic implications of resistance, it raises the question of whether tumor-extrinsic (host-mediated) processes in the tumor microenvironment are also impacted. In this regard, we do have some evidence to suggest changes in several tumor extrinsic/host processes.

First, in our transcriptomic comparisons of EMT6-PTR tumors (that contain a mix of tumor and host cell populations), we include CIBERSORT tissue deconvolution analysis which gives some estimation of immune infiltrates. In revised Figs 6A-B, we show that PTR tumors had higher total immune scores while displaying lower T cell and M2 macrophage scores (see **Summary Fig 5 below**). This result suggests a simultaneous suppressive and stimulatory immune response. This follows our data shown in a modified Fig 4I-J that PTR cells can express surface ISGs with opposing roles in immune regulation, such as MHC-I and PD-L1. To examine this further, Figs 6C-L of our manuscript show *ex vivo* co-culture studies combining splenocytes with tumor cells (i.e, direct cell-cell effects) or their conditioned media (i.e, indirect effects). Here our results confirm, at least with CD8+ T cells, there are type I IFN-mediated secretory changes in PTR cells that have immune suppressive effects. Yet, simultaneously, when cell-cell contact is allowed, this suppressive effect is limited/weakened (see **Summary Fig 5 and Modified Results text below**). While our study introduces this opposing effect in the PTR cells, it is likely that a more dedicated study will be needed that involves spatial transcriptomics to fully decipher the precise interplay between secreted/cell-surface factors in resistance. These studies are planned currently in our laboratory.

[Modified Results Section]

...’ Since secreted PTIS cytokines such as IL6, LCN2, CCL2 and non-secreted ISGs (i.e., MHC-I and PD-L1) can have opposing effects on immune activation and affect how tumor cells respond to ICI treatments (6, 15), we examined whether PTR cells might influence (or be influenced by) immune cell populations that are part of the anti-tumor response. **To test this, we first performed CIBERSORT/ImmuCC tissue deconvolution analysis to identify immune cell populations in PTR tumors using mouse-specific gene scores in RNAseq data (described in (16, 17)). EMT6-PTR tumors had higher total immune scores (Fig 6A), with activated cytotoxic CD8+ T lymphocyte (CTL) and M2 macrophage scores significantly decreased (Fig 6B), suggesting conflicting tumor immune responses may occur after acquired PD-L1 resistance. To examine this directly, we next tested EMT6-P/PTR cells for cell cytotoxicity in co-culture....**

Second, and related to this, if immune cell populations may be altered after resistance, it raised the question of whether the PTIS may be induced in non-tumor/host cells after α PD-L1 treatment. In Fig 3E of our study, we do include some evidence that a PTIS can be induced in host tissues/cells. Here we used published single-cell RNAseq data from Merkel cell carcinoma tumor biopsies taken from patients that responded to treatment with Avelumab (Paulson et al. 2018; see ref 39 in manuscript) and we found PTIS enrichment in macrophage and T cell compartments. This suggests that tumor-extrinsic secretory changes in ‘host’ cell populations can also be altered by treatment (See **Summary Fig 6 below; left box**). While we feel additional dedicated studies involving these host-responses to treatment go beyond our current study based on tumor-intrinsic changes, we have done a preliminary assessment that we include here for reviewer consideration. In **Summary Fig 6 below (right box)**, we examined bulk RNAseq datasets from our 2018 publication (Mastri et al 2018 (18)) where we tested lungs

taken from tumor-free mice treated with α PD-L1. Here we found an enrichment of PTIS genes as well as enrichment for IFN-gene signatures from various studies involving samples after treatment. This data increases the likelihood that PD-L1 inhibition can invoke cellular changes in tumor *and* host tissues to alter IFN-regulated secretory mechanisms, and strengthens our study conclusion that targeting PD-L1 can 'rewire' cellular type I IFN-controlled secretory programs. We are currently planning expanded studies involving single-cell spatial RNAseq in multiple tissues and cell types in tumor-free mice after prolonged drug exposure *in vivo* to identify a possible global PTIS. These studies would test whether the tumor, host, or both in concert, are responsible for a permanent resistant phenotype in mice. While this topic is mentioned in the Discussion section of our manuscript (see response to Comment 1.1. above) such experiments would be part of a separate/dedicated work. We hope the reviewers will agree.

1.3. Is there evidence of elevated Stat1/3 or JAK signaling in tumor cells in PTR tumors treated with anti-PD-L1 compared to parental tumors or anti-PD-L1 responsive tumors?

This is an important question. Our study shows that PTR cells have elevated STAT3 signaling which aligns with recent studies demonstrating that intrinsic PD-L1 signaling can control the STAT3/Caspase 7 activation (19). While we discuss the STAT1/STAT3 more in our response to Reviewer comment 2.4 below, our manuscript does include some discussion about the implications of our studies for STAT signaling (see below excerpted text from our Appendix Discussion). For JAK signaling, it is possible that some of the secretory programs observed after resistance are controlled by this program. To test this, our revised manuscript now includes additional analysis of PTR tumor tissues that shows a positive enrichment of multiple STAT and JAK gene sets (see Summary Fig 7 below; and new Appendix Fig 10 and Appendix Table S5 in our manuscript). These results complement our studies in Fig 4F-H and Fig EV3E-F demonstrating increased activation of STAT3 in PTR tumor cell variants selected from these tumor tissues. Our results raise the possibility that targeting the Jak pathway therapeutically or via genetic alterations may stem the PTIS and, potentially, be combined with PD-L1 antibody to limit resistance. Our discussion now raises this as a possibility for future study.

[Modified/added text in new Appendix Discussion Section]

‘.....These stimulating/inhibiting effects IFNs can, in turn, influence ICI treatment response. For inhibitory effects, IFNs (mostly IFN γ) have historically been characterized to be integral for anti-tumor immunity by driving antigen presentation(20) and chemokine secretion(21) that are typically part of the immune-editing process in normal physiological conditions. **IFNs can improve ICI responses as loss-of-function mutations in IFN signaling components (i.e. JAK1/2) have been identified in melanoma patients after pembrolizumab treatment relapse (22), and knockout of IFN pathway mediators such as Jak1, Stat1, Ifngr1 in tumors can weaken ICI treatment efficacy(23, 24).** For stimulating effects, ICIs can induce an enhanced expression of tumor cell ISGs transcribing additional T-cell co-inhibitory ligands (e.g. TNFRSF14, LGALS9) where blockade of type I and II IFN signaling could reverse this effect and improve ICI responses(25).....’

‘.....Therapeutically, sequential treatment of an ICI followed by a JAK inhibitor was found to sensitize IFN-driven resistant tumors to ICI treatment(25). Such treatment strategies tested in early phase trials have showed promising efficacy(26, 27). **Notably, we performed GSEA analysis and found an enrichment of JAK/STAT pathways in EMT6-PTR tumors, suggesting that targeting JAK therapeutically may undermine the PTIS when combined with PD-L1 inhibitors (Appendix F10 and Appendix Table 5). These studies are currently underway.....’**

‘.....Though often overlooked, PD-L1 has several tumor intrinsic functions that have recently been identified to regulate mTOR/AKT(28, 29), MAPK(30), STAT3/Caspase 7(19), integrin β 4(31), MerTK(32), and BIM/BIK(33) signaling, among others. **Recently, Gato-Canas and colleagues reported that conserved motifs of PD-L1 cytoplasmic domains (RMLDVEKC and DTSSK) block STAT3/Caspase 7 cleavage and, in turn, can control type I IFN-induced cytotoxicity(19).....’**

1.4. Results presented in Fig 4I-K should be clarified and/or discussed further. How do PTR cells increase CD8 T cell proliferation and activation (CD69, GZMB, IFN γ) via direct contact compared to parental clones?

*note: Fig 4I-K are now Fig 6I-K in the revised manuscript.

We thank the reviewer for this question and an opportunity to clarify these results. Since T cell proliferation and division in co-culture studies are known to be affected by factors involved in *direct* tumor:T cell contact (e.g. cell surface proteins) and indirect co-culture media (e.g. tumor derived secreted factors), we included experiments

in our study that assessed these separately. We found that conditioned media (CM) from PTR tumors reduced CD8 T cell division and activation (Fig 6G-H), inclusion of the tumor cells in co-culture effectively reversed this effect (with both increasing significantly). These results show the sharp contrast of how cell-surface and secreted factors can have opposing effects on surrounding immune cell activation. This result may be explained by our finding that the surface ISGs we identified to be altered in PTR tumor cells can have opposing effects on immunity. For example, in a modified Fig 4I-J we show that PTR cells have decreased PD-L1 expression, which should have an immune stimulating effect in co-culture models (34), but we also added new data to show that PTR cells have multiple MHC-I class ISGs simultaneously down-regulated (**See Summary Fig 8 below**). While *ex vivo* T cell experiments are designed to bypass antigen presentation via CD3/CD28 stimulatory antibodies, MHC-I is also known to have “reverse signaling” that can have both pro- or anti-proliferative effects on T cells depending on presence of co-stimulatory signals (35-37). Our study does not directly decipher which of these surface ISGs might be directly responsible for counteracting the immune-suppressive effects of the secretory changes in PTR cells primarily because this would take a dedicated study, but we have added clarification about these findings in a modified Discussion (excerpted below).

[Modified Discussion Text]

‘....This raises the question of whether disruption of the PTIS in PTR cells should involve targeting the secretory-controlling IFN-signaling machinery or targeting ISG secretory products directly, thereby potentially avoiding disruption of IFNs anti-tumor functions. In our study, both approaches were evaluated. In the first, IFNAR1 knockdown in PTR cells was found to effectively reverse IFNβ-mediated PTIS factor increases. **In splenocytes studies, we noted PTR secretory factors were capable of suppressing T cell activation while direct co-culture studies revealed PTR cells increased T cell activation but had decreased sensitivity to T cell cytotoxicity. This may be explained by changes in cell surface proteins in PTR cells that can alter activation when bound directly with T cells and have reverse signaling that affect tumor cell survival. For instance, PD-L1 levels are significantly decreased in PTR cells which can result in increased T cell activation (38). In addition, PD-L1 is also known to have reverse intrinsic signaling that can affect apoptosis and cell survival (19, 29). Our studies show that antibody blockade and genetic disruption of PD-L1 can have tumor intrinsic effects including impact on PTIS expression (see supplemental discussion). It is important to note that these *ex vivo* splenocyte studies are limited in recapitulating PTR-immune cell interactions for several reasons. First, splenocyte studies were conducted using CD3/CD28 agonistic antibodies to**

bypass initial signals for T cell activation such as antigen presentation. However, despite this, our findings suggest that PTR secretomes were capable of suppressing T cell activation independent of antigen presentation. Second, our studies were focused on assessing secretory changes in PTR cells and how these changes affect the resistance phenotype, additional studies would be needed to investigate changes in cell surface and signaling changes after resistance. Lastly, splenocyte studies were focused on assessing interactions between PTR cells and activated T cells, additional tumor-immune cell interactions known to affect tumor progression such as myeloid and NK cell populations would require further investigation. In this regard, we assessed tumor growth in vivo and found that PTR-IFNAR1^{KD} tumors grew slower than PTR tumors, suggesting that constant PD-L1 blockade confers a unique vulnerability to IFN blockade in tumor cells. However, our results also show that IFNAR1^{KD} tumors grew much faster than parental controls, emphasizing the sometimes contradictory role of IFNAR signaling in cancer controlling immunosurveillance(39) (see supplemental discussion) which might be exacerbated by anti-PD-L1 treatment and, as a consequence, may introduce potential challenges of intracellular IFN signaling inhibition strategies....'

1.5. Is the effect in Fig 5A, with loss of IFNAR in PTR cells slowing tumor growth dependent on T cells?

****note: Fig 5A is now Fig 7A in the revised manuscript.***

The reviewer raises an important question on whether slowed growth in PTR-IFNAR1^{KD} tumors is dependent on T cells. We initially addressed interactions between T cells and EMT6-PTR tumor cells utilizing *ex vivo* splenocyte experiments (Figure 6). Here we showed that IFNAR1^{KD} is capable of restoring T cell activation and tumor cell apoptosis in PTR tumor cells that correlates with *in vivo* tumor growth suppression seen in Figure 7. In some ways, these splenocyte studies represent a more direct assessment of tumor-T cell interactions as these systems isolate these two cell populations to assess relevant effects. But we agree that understanding the extent of T cell involvement in PTR tumor growth *in vivo* would be interesting and should be explored in subsequent studies along with a myriad of immune cell populations with suppressive/stimulatory effects. For example, given secretory changes such as IL6 and CCL2 found in PTR tumors, it is possible that tumor-myeloid cells (e.g. macrophages, MDSCs) may be impacted by PD-L1 inhibitor resistance. Such studies would ideally be accompanied by CD4 and CD8 depletion studies, CD4/CD8 knockout mouse models, and additional immune characterization of EMT6 P and PTR tumors – all of which we have plans to pursue in studies beyond this current manuscript. We have added these points to the Discussion text shown in Reviewer Comment 1.4 above. We hope this will suffice and thank the reviewer for this comment.

Referee #2:

The authors observe regulation of the expression of secretory proteins (α PD-L1 treatment-induced secretome or PTIS) in response to anti PD-L1 in cancer cells sensitive or resistant to anti PD-L1. The authors claim that PTIS expression is dependent on IFN-I signaling and that inhibition of PTIS by disrupting IFN-I signaling inhibits the growth of anti PD-L1 resistant tumors. The manuscript presents data inconsistent with this claim without sufficient explanation and therefore the conclusions are not well founded. No convincing evidence is presented that secretion of endogenous IFN is induced by treatment with anti PD-L1.

General Comments:

As mentioned in our Introduction above, we feel several concerns raised by reviewers stemmed from a lack of clarity in our original manuscript narrative and data presentation. An example of this includes the above statements by Reviewer 2 related to the role of endogenous/autocrine IFN α/β . The reviewer states that '*no convincing evidence is presented that secretion of endogenous IFN is induced by treatment with anti PD-L1*', to which we would agree because we did not intend to make this claim and did not directly test this in our original study. Yet it is an important concept to consider because our PTR cells clearly have elevated PTIS factors that are secreted ISGs, and these can be further elevated by exogenous type I IFNs, so it is possible that an autocrine mechanism may be contributing to the PTIS. If so, then this an added variable to our study conclusions. For this reason, in our revised manuscript we have included new data that shows the IFN α/β levels in EMT6-PTR cells. We found expression of both to be very low and likely functionally insignificant. IFN α is unchanged and while we did see a significant increase in transcripts for IFN β in PTR cells, we could not detect any IFN β protein by ELISA or western blotting, even in extremely high volumes of lysate (**see Summary Fig 9 below**). Our conclusion from this data is that these endogenous levels are below functional amounts. This is supported by our finding that, when exogenous IFN α/β is added to PTR cells, we observed a dramatic PTIS increase, which is then reversed when IFNAR1 expression is knocked down. We conclude that the 'rewiring' of the IFN signaling in PTR cells includes a secretome that, while elevated after resistance, is likely further enhanced *in vivo* via paracrine signaling. We have included changes to the results to reflect this new data and further discuss the IFN stimulation comment 2.5 below.

[New text in Modified Results]

'....Since IFN α /IFN β are known to bind to IFNAR(40) and drive ISG expression, we next examined whether activation of type I IFN signaling could further modulate PTIS expression. First we tested endogenous/intrinsic IFN α /IFN β levels in EMT6-P and -PTR variants. IFN α was not detectable at the transcript level and, while IFN β transcript levels did increase in

PTR cells, these values were low and highly variable and no protein could be detected in lysates (Fig 4A; protein shown in Appendix Fig S4). Together this suggests that PTIS and ISG expression in PTR cells is not driven by an autocrine IFNAR1 activation. Next, we added exogenous IFN α /IFN β and found that, after 48 hours stimulation, PTIS expression increased in EMT6-PTR cells....'

2.1. In Fig 3e, IL-6 protein level was not affected by IFNAR KD in the absence of treatment with exogenous IFN β , suggesting that endogenous IFN-I signaling plays at most a minor role in increasing IL-6 expression in PTR cells compared to P cells.

***note: our original Fig 3e is now shown as a modified/updated Fig 2H-J, Fig 4B-E, and Fig EV 3A-B in the revised manuscript.**

This is one of the questions that we feel originated from how we initially presented our results, which has now changed significantly in our revised manuscript. In this regard, we would note two important points. **First**, and related to IL-6 expression in IFNAR1^{KD} cells, our graphics originally displayed IFN signaling stimulation and inhibition together in the same graph in a way difficult (if not impossible) to accurately compare PTIS changes. Our new data, originally shown for only IL6, now shows all PTIS factors so more clear conclusions/comparisons can be made. These experiments have been expanded with larger N and replicated several times to show that IL6 (and other PTIS factors) all significantly increase at the protein and transcript level in EMT6-PTR cells, and then all decrease in IFNAR1^{KD} variants (**Summary Fig 10 below shows example**).

Second, in a modified Figure 4, we have significantly expanded our data showing the PTIS can be regulated by type I IFN stimulation. We show that PTIS is elevated by exogenous IFN α / β (Fig 4B) and then reduced following IFNAR1^{KD} (Fig 4C). This data includes IL6 for which we conducted more experiments and comparisons, including the relative amount that IFN α / β increased the already elevated PTIS (Fig 4D). We also included comparative examples of the magnitude of elevations by IFN α / β for IL6 (Fig 4E). This data is shown in **Summary Fig 11 below** and demonstrates the unique role type I IFN signaling in PTR cells can regulate the PTIS. Note that we have added extensive new analysis of this data to show the significance in Fig EV3A-B. We hope that this new data and explanation addresses this reviewer concern.

2.2. In Fig 3f, in the absence of exogenous IFN β treatment or IFNAR1 KD, p-STAT1 levels are decreased, but pSTAT3 levels are increased in PTR cells compared to P cells. This result indicates that elevated PTIS levels in PTR cells, in the absence of treatment with exogenous IFN β , are mainly driven by pSTAT3, not pSTAT1, suggesting that other cytokines that exclusively induce pSTAT3, not pSTAT1, (eg. IL-6) may be responsible for increased PTIS expression in PTR cells.

2.4. Fig 5a shows that the growth of PTR cells was significantly decreased when IFNAR1 was knocked down, and the authors claim that inhibition of PTIS regulators (based on their assumption that IFN-I signaling is responsible for PTIS expression) inhibits PTR tumor growth. This paper does not present sufficient data showing IFN-I signaling is responsible for PTIS expression. Fig 3 data suggest that PTIS expression in PTR cells may be induced through other cytokines that activate STAT3, not STAT1. Therefore, the effect of IFNAR1 KD in Fig5a cannot be due to blockade of PTIS expression.

***notes:**

- 1) Our original Fig 3F is now a modified Fig 4F-H, Fig EV3 E-F. We have also added a new Appendix Fig S7 to address concerns raised below.
- 2) Our original Fig 5A is now Fig 7A

We have chosen to address comments 2.2 and 2.4 together here because they both relate to the STAT 1/3 data presented and raise the possibility that the elevated STAT3 signaling in PTR cells may be driven by an autocrine secretome activation separate from the IFN-dependent mechanism we propose for the PTIS. To address this, we have significantly improved the data and analysis for this section and added new experiments with the following explanation. **First**, our paper does show evidence that the STAT3 activation is dependent, at least in part, on IFN signaling in the PTR cells but this may have been hidden in our original presentation of the data. In **Summary Figure 12 below (see red boxes)**, we show that the increases in activated pSTAT3 (relative to STAT3 expression) are decreased after IFNAR1^{KD}. While this decrease does not reach significance in our 3 replicates, it suggests the STAT3 elevation is causally connected to type I IFN signaling, and is an effect enhanced by IFN β stimulation.

Second, the reviewer raises the possibility that STAT3 activation may be driven by PTIS factors known to stimulate the STAT3 pathway, and therefore be an IFN-independent autocrine mechanism that would explain our results. The reviewer raises IL6 as an example because, while type I IFNs are known to activate STAT1 and STAT3 (41, 42), IL6 is known to primarily activate STAT3 (43). To test for a potential role for autocrine IL6 as a driver of the STAT3 activation we observe, we have included new data in an Appendix Fig S7 showing that treatment of EMT6 P/PTR cells with a mouse α IL6 antibody has no effect on the STAT3 activation observed (see **Summary Fig 13**). Finally, and related, we hope that our condensed 5-gene PTIS gene score and our expanded confirmation of these factors (detailed in Comment 1.1 above) provide a much stronger support to the claim that type I IFN signaling can control PTIS. We would agree that our original 13 gene PTIS did not support

the claims made. In this regard, we feel that this more focused narrative supports the claim made for Fig 7A and 7B which shows that, while IL6 inhibition can limit PTR tumor growth to an extent, it is when tumor-intrinsic type I IFN signaling is disrupted where this effect is most pronounced. That said, we fully agree that other IFNAR1 controlled mechanisms may be contributing to this effect, including surface ISG such as MHC-class proteins and other antigen expressing genes. The ideal experiment would include multi-antibody inhibition of the full 5 gene PTIS. While these experiments are underway, we feel they are beyond the scope of the current manuscript.

2.3. PD-L1 expression was decreased in PTR cells (Fig 3i), but T cell mediated cell death was decreased in PTR cells (Fig 4d). This result is inconsistent with the ability of PD-L1 to inhibit cancer cell killing by T cells.

****note: our original Fig 3i is now shown as a modified/updated Fig 4I-J, and Fig 4D is an updated Fig 6C-E in the revised manuscript.***

Thank you for this comment. In response to Reviewer comment 3.2 (below) we expand on this topic about the apoptosis markers; however, for the expression of PD-L1, the reviewer raises a key point about the dual (and conflicting) properties of the surface ISGs we observe in our study. First, multiple factors are likely impacting the results shown in our co-culture studies. T cell cytotoxicity can be influenced by secreted molecules and cell surface proteins, while tumor cell death can be influenced by secreted molecules as well as intracellular proteins/signaling. PD-L1 can influence both T cell and tumor cell cytotoxicity. This is because PD-L1 impacts T cell activation (44) (and subsequently T cell cytotoxicity) but also can have reverse/ intrinsic signaling that can impact tumor cell survival pathways such as STAT3, AKT, mTOR. Our results showing a decrease in PD-L1 expression in PTR cells suggest there would be an increase in cancer cell killing in co-culture, but this was not our observation. Instead, we saw a decrease and observed multiple MHC I genes were downregulated (See Reviewer comment 1.4 above and Summary Fig 8). This suggests that PD-L1 is only one factor likely impacting tumor cell survival in these co-culture assays, and we demonstrate there can be simultaneous opposing effects on immune cell functions. For example, at least for secretory factors, our revised manuscript shows an expansion of data with the PTIS factors that can directly impact cell survival. LCN2 can induce CD8 T cell apoptosis(45) and IL6 has been recently characterized to inhibit cytotoxic T cell functions (e.g. IFN γ and perforin) through BATF induction(46). Intracellularly, activation of STAT3 is known to be connected to the activation of tumor cell survival pathways(47). Together, these changes in PTR cells likely contribute to decreased sensitivity to T cell killing and may mask the overall effect of PD-L1 expression in this co-culture system. In our study, we have made clear that the results in Fig 6 represent an 'opposing' impact of PTR cells and we have framed these results as 'IFN-regulated' because we observe reversals of this phenomenon when IFNAR1 is knocked down. To highlight this point, our discussion has an expanded section detailing the opposing influences of IFN-signaling (and ISGs) on tumor growth (detailed in Reviewer Comment 1.2 and in the new Appendix Discussion section mentioned in Comment 1.3). We hope these added changes will suffice and thank you for the question.

2.5. "PD-L1 treatment-induced secretome (PTIS)" in the abstract should read anti-PD-L1.

Thank you, we have modified this here and in the rest of the document

2.6. In EMT6-P cells, IFN β increased IL-6 protein levels (Fig 3c and e), but not IL-6 mRNA levels. Further discussion about this difference is needed.

Thank you for raising this concern. As discussed in Comment 2.1 above, we have added more data to show that the protein and mRNA levels are both increased by IFN β (Summary Figs 10 and 11). This was not clear in our original manuscript.

Referee #3:

In this study the authors establish a PD-L1 EMT6 resistant tumor cell line that initially responds but progresses to anti-PDL1 treatment. The results are a comparison between PTR tumors and parental (untreated tumors). The authors establish that upregulation of IFN response genes or tumor IFN responsiveness after anti-PDL1 tumor immunotherapy results in acquired resistance. While resistance due to chronic IFN signaling is known, the IFN regulated program that separates response vs resistance in tumors was not established in this manuscript. The authors identify an interesting finding that IFNAR1 KD results in a decrease of PTR tumor growth however mechanisms of these results were not investigated.

Resistance mechanisms are drawn between responders and non-responders to immunotherapy. However, the authors perform a differential analysis between parental (untreated) tumors and tumors that initially respond but progress after anti-PD-L1 therapy. The authors state that reason of acquired resistance is upregulation of IFN response genes after anti PD-L1 treatment. But, upregulation of IFN response genes can be an outcome of an increase in IFN responsiveness of the tumor after anti-PD-L1 treatment which can also result in a delay in tumor growth compared to the parental tumor.

General comments:

We thank the reviewer for their comments. As we describe below, several points raised by reviewer 3 motivated us to make the significant changes now shown in our revised study and some comments seem based on our original lack of clarity in the conclusions presented. This includes the discussion about the implications of the PTIS that we discuss in our above introduction. We hope we have addressed the concern about whether it is a cause or consequence of resistance, and how that should be described in our text. Overall, our study is presenting these secretory changes as a consequence of a tumor-intrinsic 'rewiring' of the IFN-signaling machinery that, when blocked, can have significant inhibitory effects on tumor growth. This is a key point as the anti-tumor functions of PD-L1 targeted therapy are thought to be primarily involving the immune system. Our study demonstrates an example of the direct effects of resistance on the tumor cell population, which we show is partially controlled by direct action of the PD-L1 inhibitor and the tumor cells themselves (i.e, in our studies involving prolonged *in vitro* drug exposure).

3.1) The authors state that treatment of PTR to IFN-A, B, Y results in an increase of PTIS (PD-L1 treatment-induced secretome) program in the tumors as indicative of acquired resistance. Here, CXCL9 and CXCL10 although within PTIS program are downregulated or not significant.

We agree with this comment and, as described above, this prompted us to make significant changes to our study. This included the creation of a modified/optimized 5-gene PTIS signature and the addition of extensive data showing this signature is IFN-regulated. Importantly, we did not remove any of the original data shown in our original study and have included the many factors (i.e, CXCL9/10, etc) identified in our original PTIS. These are all known ISGs and represent important confirmations of the secretory changes in the PTR cells, even if they all do not show consistent changes across all models tested in our study. In our revised manuscript, we show several of these secreted ISGs using our *in vivo*-derived (Fig EV3C) and *in vitro*-derived (Figs EV4C-F) cell lines after PD-L1 signaling disruption (**see Summary Fig 14 below**). Related to this, one significant change made in our revised manuscript was to separate our stimulation studies with IFN γ and put into a dedicated Appendix Fig S5. The reason for this was to re-focus our manuscript on type I IFN effects using stimulation (using IFN α/β) and inhibition (via IFNAR1^{KD}). While our data does show several PTIS factors can be stimulated by IFN γ , we discuss this separately in the manuscript as further evidence that the IFN-signaling machinery in PTR cells is 'rewired'. In this case, our data shows that enhanced PTIS can occur via stimulation with multiple IFN types, something we are exploring via dedicated/separate studies.

3.2) The authors noted a decrease in tumor cell death of PTR tumors vs control tumors however the mechanism behind the cell death decrease was not investigated. An association between upregulation of IFN response genes and decrease in tumor cell death needs to be established.

We thank the reviewer for this point and agree that there is no mechanism for this cell death decrease presented in our study. We have not added more data exploring the relationship between PTR cells and T cell mediated cytotoxicity because we feel this would require a series of additional studies beyond the scope of this manuscript. For instance, to fully investigate this, several possible mechanistic reasons would need to be explored. This would include differential cell sensitivity to Granzyme/Perforin/T cell Fas-Fas ligand killing, expression of additional apoptosis regulators (e.g BCL2, Caspase proteins), and differential progression of cells through cell cycle. Furthermore, as noted in our response to Comment 2.3 above, there are several possible factors that could be acting in concert, or in conflict, with the induction of apoptosis. This could include PD-L1 expression, alternative cell surface immune checkpoint protein expression (e.g. Tim3, Lag3), and additional intracellular proteins that can impact tumor-T cell interactions. While we do not provide an answer for this in our current manuscript, we have modified the presentation of this data to explain that extra studies are needed to address this more fully. In addition, we have modified the graphical illustration for our results in our revised manuscript to illustrate (albeit broadly) that protection from T cell cytotoxicity as just one part of the general immune-suppressive system our work has uncovered (shown in a revised Fig 7G and in the Synopsis; see **Summary Fig 15**). We hope the reviewer will agree with this conclusion.

3.3) The authors observe a decrease in PTR tumor growth on IFNAR1 KD. Here, the resistance program within the IFN response genes, the knockdown of which results in a decrease of tumor growth should be defined.

We agree with the reviewer as stated above that the original manuscript did not include sufficient data to characterize PTIS expression in PTR or PTR-IFNAR1^{KD} models. The extensive changes in our revised study include significantly expanded studies of the revised PTIS in the IFNAR1^{KD} model at baseline, after resistance, and after IFN stimulation. In our *in vivo* studies we show that these PTR-IFNAR1^{KD} cells have significantly decreased tumor growth kinetics compared to the PTR cells which we link to PTIS decreases. However, there are two reasons why we do not further assess PTIS inhibition in this manuscript. **First**, in addition to tumor inhibitory effects seen in PTR tumors, therapeutic inhibition of IFN can have tumor promoting effects. Type I IFN promote antigen presentation (48), inhibit proliferation(49) and enhance anti-tumor T cell functions (39). Inhibition of these pathways can promote tumor growth. This effect is shown experimentally in our manuscript where

IFNAR1^{KD} in parental tumor cells results in decreased sensitivity to T cell cytotoxicity and growth acceleration upon implantation. Furthermore, IFN signaling in immune cells is associated with better prognosis for patient treated with ICIs (50). Taken together, this may indicate that a more practical target therapeutically would be downstream effects of IFN that can include PTIS changes identified in this manuscript. **Second**, in our manuscript, we tested targeting of one PTIS factor (IL6), which showed a modest inhibition of tumor growth that was only in PTR tumors. Therapeutically, it may be more beneficial to design strategies to inhibit multiple PTIS factors. This is an approach that we are currently investigating in the lab with the use of novel strategies that include multiple knockdown cell lines for each PTIS factor, multiple antibody combinations, and the use of vaccine-based approaches. But this will require extensive testing beyond the scope of this study. Again, we thank the reviewer for raising the above points as they are very critical to our future studies in examining therapeutic options after PD-L1 resistance. These points have been addressed in a revised discussion in our manuscript.

REVIEWER APPENDIX: SUMMARY OF NEW EXPERIMENTS

Fig 2: Revised α PDL1 treatment-induced secretome (PTIS) 5-gene score

A. 2C-I; EV1B-C; Appendix Figs 1-3: New proteomic confirmation of PTIS using:

- Protein Array: Analysis identifying IFN-regulated genes
 - New results showing increases in EMT6-PTR and decreases in EMT6-PTR-IFNAR1^{KD}
 - Total Heatmap/statistical summary
- Western blotting/densitometry: for PTIS molecules LCN2, NOS2
 - New results showing increases in EMT6-PTR and decreases in EMT6-PTR-IFNAR1^{KD}
 - New Appendix Figure of blotting replicates
- ELISA: for PTIS molecules LCN2, CCL2, IL6
 - New results showing increases in EMT6-PTR and decreases in EMT6-PTR-IFNAR1^{KD}

B. 2A-B; EV1D: Transcriptomic confirmation of PTIS

- qRT-PCR: for all PTIS molecules
 - New results showing increases in EMT6-PTR and decreases in EMT6-PTR-IFNAR1^{KD}
 - Total Heatmap/statistical summary

Fig 3: Revised PTIS validation in clinical/preclinical datasets

A. 3A-E: Revised analysis showing PTIS enrichment in α PDL1 treatment-*sensitive* tumors

- 3C: Heatmap summary showing significance

Fig 4 IFNs modulate PTIS, STAT signaling, and surface ISGs in PTR cells

A. 4A: new data showing EMT6-PTR cells do not increase type I IFN expression

B. 4B-D and EV3A-B: new transcriptomic data showing PTIS enhancement in PTR by exogenous type I IFN stimulation

C. 4E and EV3C: new protein data confirming select PTIS factors (IL-6) and other ISGs stimulated by enhancement in PTR by exogenous type I IFN stimulation

D. 4G-H and EV34-G: modified analysis of western densitometry showing STAT3 increase in PTR cells.

E. Appendix Fig S4: new data showing IFN β protein is not expressed in EMT6 P/PTR cells.

F. Appendix Figs S5A-I and Supplemental Results: new data showing PTIS can also be enhanced by type II IFN (IFN γ)

G. Appendix Figs S7A-D: new data showing autocrine IL-6 is not responsible for STAT3 activation in PTR cells (full uncropped blots are shown in new Appendix Fig 8)

H. 4I-J and EV3I: new data showing IFN-regulated surface ISGs (MHC-I class and PD-L1) downregulation in PTR cells

Fig 5: Intrinsic PD-L1 only partially regulates PTIS

A. 5A-C: New confirmation of PTIS using cytokine protein Array showing PTIS increases in EMT6-PTR^{VITRO} cells (new Appendix Fig 9 shows array exposures)

B. 5D-F: revised transcriptomic analysis of PTIS analysis in EMT6-PTR^{VITRO} cells alone or compared to *in vivo*-derived cells

C. 5G-H: Revised analysis of PTIS expression in additional tumor cells with PD-L1 signaling disrupted/targeted

D. EV4C-G: Revised analysis multiple ISG/PTIS genes in 3 ex vivo models with PD-L1 signaling disrupted/targeted

Appendix Discussion:

A. Appendix Fig 10 and Appendix Table 5: New data showing enrichment of JAK/STAT pathways in EMT6-PTR cells

REFERENCES

1. Schoenfeld AJ, Hellmann MD. Acquired Resistance to Immune Checkpoint Inhibitors. *Cancer cell*. 2020;37(4):443-55.
2. Kluger HM, Tawbi HA, Ascierto ML, Bowden M, Callahan MK, Cha E, et al. Defining tumor resistance to PD-1 pathway blockade: recommendations from the first meeting of the SITC Immunotherapy Resistance Taskforce. *J Immunother Cancer*. 2020;8(1).
3. Mondini M, Loyher PL, Hamon P, Gerbe de Thore M, Laviron M, Berthelot K, et al. CCR2-Dependent Recruitment of Tregs and Monocytes Following Radiotherapy Is Associated with TNF α -Mediated Resistance. *Cancer Immunol Res*. 2019;7(3):376-87.
4. Weber R, Riester Z, Huser L, Sticht C, Siebenmorgen A, Groth C, et al. IL-6 regulates CCR5 expression and immunosuppressive capacity of MDSC in murine melanoma. *J Immunother Cancer*. 2020;8(2).
5. Tsukamoto H, Fujieda K, Miyashita A, Fukushima S, Ikeda T, Kubo Y, et al. Combined Blockade of IL6 and PD-1/PD-L1 Signaling Abrogates Mutual Regulation of Their Immunosuppressive Effects in the Tumor Microenvironment. *Cancer Res*. 2018;78(17):5011-22.
6. Tu MM, Abdel-Hafiz HA, Jones RT, Jean A, Hoff KJ, Duex JE, et al. Inhibition of the CCL2 receptor, CCR2, enhances tumor response to immune checkpoint therapy. *Commun Biol*. 2020;3(1):720.
7. Harlin H, Meng Y, Peterson AC, Zha Y, Tretiakova M, Slingluff C, et al. Chemokine expression in melanoma metastases associated with CD8 $^+$ T-cell recruitment. *Cancer Res*. 2009;69(7):3077-85.
8. Hopewell EL, Zhao W, Fulp WJ, Bronk CC, Lopez AS, Massengill M, et al. Lung tumor NF-kappaB signaling promotes T cell-mediated immune surveillance. *J Clin Invest*. 2013;123(6):2509-22.
9. Fisher DT, Appenheimer MM, Evans SS. The two faces of IL-6 in the tumor microenvironment. *Semin Immunol*. 2014;26(1):38-47.
10. Fisher DT, Chen Q, Skitzki JJ, Muhitch JB, Zhou L, Appenheimer MM, et al. IL-6 trans-signaling licenses mouse and human tumor microvascular gateways for trafficking of cytotoxic T cells. *J Clin Invest*. 2011;121(10):3846-59.
11. Mastri M, Lee CR, Tracz A, Kerbel RS, Dolan M, Shi Y, et al. Tumor-Independent Host Secretomes Induced By Angiogenesis and Immune-Checkpoint Inhibitors. *Molecular cancer therapeutics*. 2018;17(7):1602-12.
12. Ramos-Casals M, Brahmer JR, Callahan MK, Flores-Chavez A, Keegan N, Khamashta MA, et al. Immune-related adverse events of checkpoint inhibitors. *Nat Rev Dis Primers*. 2020;6(1):38.
13. Rotz SJ, Leino D, Szabo S, Mangino JL, Turpin BK, Pressey JG. Severe cytokine release syndrome in a patient receiving PD-1-directed therapy. *Pediatr Blood Cancer*. 2017;64(12).
14. Ceschi A, Nosedà R, Palin K, Verhamme K. Immune Checkpoint Inhibitor-Related Cytokine Release Syndrome: Analysis of WHO Global Pharmacovigilance Database. *Front Pharmacol*. 2020;11:557.
15. Adler O, Zait Y, Cohen N, Blazquez R, Doron H, Monteran L, et al. Reciprocal interactions between innate immune cells and astrocytes facilitate neuroinflammation and brain metastasis via lipocalin-2. *Nat Cancer*. 2023;4(3):401-18.
16. Chen Z, Huang A, Sun J, Jiang T, Qin FX, Wu A. Inference of immune cell composition on the expression profiles of mouse tissue. *Sci Rep*. 2017;7:40508.
17. Chen Z, Quan L, Huang A, Zhao Q, Yuan Y, Yuan X, et al. seq-ImmuCC: Cell-Centric View of Tissue Transcriptome Measuring Cellular Compositions of Immune Microenvironment From Mouse RNA-Seq Data. *Front Immunol*. 2018;9:1286.
18. Mastri M, Lee CR, Tracz A, Kerbel RS, Dolan M, Shi Y, et al. Tumor-independent host secretomes induced by angiogenesis and immune-checkpoint inhibitors. *Molecular cancer therapeutics*. 2018;17(7):1602-12.
19. Gato-Canas M, Zuazo M, Arasanz H, Ibanez-Vea M, Lorenzo L, Fernandez-Hinojal G, et al. PDL1 Signals through Conserved Sequence Motifs to Overcome Interferon-Mediated Cytotoxicity. *Cell Rep*. 2017;20(8):1818-29.
20. Gettinger S, Choi J, Hastings K, Truini A, Datar I, Sowell R, et al. Impaired HLA Class I Antigen Processing and Presentation as a Mechanism of Acquired Resistance to Immune Checkpoint Inhibitors in Lung Cancer. *Cancer Discov*. 2017;7(12):1420-35.
21. Chheda ZS, Sharma RK, Jala VR, Luster AD, Haribabu B. Chemoattractant Receptors BLT1 and CXCR3 Regulate Antitumor Immunity by Facilitating CD8 $^+$ T Cell Migration into Tumors. *Journal of immunology*. 2016;197(5):2016-26.
22. Zaretsky JM, Garcia-Diaz A, Shin DS, Escuin-Ordinas H, Hugo W, Hu-Lieskovan S, et al. Mutations Associated with Acquired Resistance to PD-1 Blockade in Melanoma. *N Engl J Med*. 2016;375(9):819-29.

23. Manguso RT, Pope HW, Zimmer MD, Brown FD, Yates KB, Miller BC, et al. *In vivo* CRISPR screening identifies Ptpn2 as a cancer immunotherapy target. *Nature*. 2017;547(7664):413-8.
24. Torrejon DY, Abril-Rodriguez G, Champhekar AS, Tsoi J, Campbell KM, Kalbasi A, et al. Overcoming Genetically Based Resistance Mechanisms to PD-1 Blockade. *Cancer Discov*. 2020;10(8):1140-57.
25. Benci JL, Xu B, Qiu Y, Wu TJ, Dada H, Twyman-Saint Victor C, et al. Tumor Interferon Signaling Regulates a Multigenic Resistance Program to Immune Checkpoint Blockade. *Cell*. 2016;167(6):1540-54 e12.
26. Zak J, Pratumchai I, Marro BS, Marquardt KL, Zavareh RB, Lairson LL, et al. JAK inhibition enhances checkpoint blockade immunotherapy in patients with Hodgkin lymphoma. *Science*. 2024;384(6702):eade8520.
27. Mathew D, Marmarelis ME, Foley C, Bauml JM, Ye D, Ghinnagow R, et al. Combined JAK inhibition and PD-1 immunotherapy for non-small cell lung cancer patients. *Science*. 2024;384(6702):eadf1329.
28. Chang CH, Qiu J, O'Sullivan D, Buck MD, Noguchi T, Curtis JD, et al. Metabolic Competition in the Tumor Microenvironment Is a Driver of Cancer Progression. *Cell*. 2015;162(6):1229-41.
29. Clark CA, Gupta HB, Sareddy G, Pandeswara S, Lao S, Yuan B, et al. Tumor-Intrinsic PD-L1 Signals Regulate Cell Growth, Pathogenesis, and Autophagy in Ovarian Cancer and Melanoma. *Cancer Res*. 2016;76(23):6964-74.
30. Wu X, Li Y, Liu X, Chen C, Harrington SM, Cao S, et al. Targeting B7-H1 (PD-L1) sensitizes cancer cells to chemotherapy. *Heliyon*. 2018;4(12):e01039.
31. Wang S, Li J, Xie J, Liu F, Duan Y, Wu Y, et al. Programmed death ligand 1 promotes lymph node metastasis and glucose metabolism in cervical cancer by activating integrin beta4/SNAI1/SIRT3 signaling pathway. *Oncogene*. 2018;37(30):4164-80.
32. Du W, Zhu J, Zeng Y, Liu T, Zhang Y, Cai T, et al. KPNB1-mediated nuclear translocation of PD-L1 promotes non-small cell lung cancer cell proliferation via the Gas6/MerTK signaling pathway. *Cell Death Differ*. 2021;28(4):1284-300.
33. Feng D, Qin B, Pal K, Sun L, Dutta S, Dong H, et al. BRAF(V600E)-induced, tumor intrinsic PD-L1 can regulate chemotherapy-induced apoptosis in human colon cancer cells and in tumor xenografts. *Oncogene*. 2019;38(41):6752-66.
34. Azuma T, Yao S, Zhu G, Flies AS, Flies SJ, Chen L. B7-H1 is a ubiquitous antiapoptotic receptor on cancer cells. *Blood*. 2008;111(7):3635-43.
35. De Felice M, Turco MC, Corbo L, Carandente Giarrusso P, Lamberti A, Valerio G, et al. Lack of a role of monocytes in the inhibition by monoclonal antibodies to monomorphic and polymorphic determinants of HLA class I antigens of PHA-P-induced peripheral blood mononuclear cell proliferation. *Cellular immunology*. 1989;122(1):164-77.
36. Turco MC, de Felice M, Corbo L, Giarrusso PC, Yang SY, Ferrone S, et al. Enhancing effect of anti-HLA class I monoclonal antibodies on T cell proliferation induced via CD2 molecule. *Journal of immunology*. 1988;141(7):2275-81.
37. Muntjewerff EM, Meesters LD, van den Bogaart G, Revelo NH. Reverse Signaling by MHC-I Molecules in Immune and Non-Immune Cell Types. *Front Immunol*. 2020;11:605958.
38. Freeman GJ, Long AJ, Iwai Y, Bourque K, Chernova T, Nishimura H, et al. Engagement of the PD-1 immunoinhibitory receptor by a novel B7 family member leads to negative regulation of lymphocyte activation. *J Exp Med*. 2000;192(7):1027-34.
39. Boukhaled GM, Harding S, Brooks DG. Opposing Roles of Type I Interferons in Cancer Immunity. *Annual review of pathology*. 2021;16:167-98.
40. Schoggins JW. Interferon-Stimulated Genes: What Do They All Do? *Annu Rev Virol*. 2019;6(1):567-84.
41. Tsai MH, Pai LM, Lee CK. Fine-Tuning of Type I Interferon Response by STAT3. *Front Immunol*. 2019;10:1448.
42. Plataniias LC. Mechanisms of type-I- and type-II-interferon-mediated signalling. *Nature reviews Immunology*. 2005;5(5):375-86.
43. Johnson DE, O'Keefe RA, Grandis JR. Targeting the IL-6/JAK/STAT3 signalling axis in cancer. *Nat Rev Clin Oncol*. 2018;15(4):234-48.
44. Sharpe AH, Pauken KE. The diverse functions of the PD1 inhibitory pathway. *Nat Rev Immunol*. 2018;18(3):153-67.
45. Floderer M, Prchal-Murphy M, Vizzardelli C. Dendritic cell-secreted lipocalin2 induces CD8+ T-cell apoptosis, contributes to T-cell priming and leads to a TH1 phenotype. *PloS one*. 2014;9(7):e101881.
46. Huseni MA, Wang L, Klementowicz JE, Yuen K, Breart B, Orr C, et al. CD8(+) T cell-intrinsic IL-6 signaling promotes resistance to anti-PD-L1 immunotherapy. *Cell Rep Med*. 2023;4(1):100878.

47. Huynh J, Chand A, Gough D, Ernst M. Therapeutically exploiting STAT3 activity in cancer - using tissue repair as a road map. *Nat Rev Cancer*. 2019;19(2):82-96.
48. Stark GR, Kerr IM, Williams BR, Silverman RH, Schreiber RD. How cells respond to interferons. *Annu Rev Biochem*. 1998;67:227-64.
49. Evinger M, Rubinstein M, Pestka S. Antiproliferative and antiviral activities of human leukocyte interferons. *Arch Biochem Biophys*. 1981;210(1):319-29.
50. Benci JL, Johnson LR, Choa R, Xu Y, Qiu J, Zhou Z, et al. Opposing Functions of Interferon Coordinate Adaptive and Innate Immune Responses to Cancer Immune Checkpoint Blockade. *Cell*. 2019;178(4):933-48 e14.

Dear Dr. Ebos,

Thank you for the re-submission of your revised manuscript to our editorial offices. I have now received the reports from the three referees that I asked to re-evaluate your study, you will find below. As you will see, referees #1 and #2 now fully support the publication of the study. Referee #3 indicates remaining lack of mechanistic insight. As EMBO reports emphasizes novel functional over detailed mechanistic insight, I will nevertheless continue with the manuscript.

Before I can proceed with formal acceptance, I have these editorial requests I ask you to address in a final revised manuscript:

- We now use CRediT to specify the contributions of each author in the journal submission system. CRediT replaces the author contribution section. Please use the free text box to provide more detailed descriptions and do not provide your final manuscript text file with an author contributions section. See also our guide to authors:
<https://www.embopress.org/page/journal/14693178/authorguide#authorshipguidelines>

- Please order the manuscript sections like this, using these names:
Title page - Abstract - Keywords - Introduction - Results - Discussion - Methods - Data availability section - Acknowledgements (including funding information) - Disclosure and Competing Interests Statement - References - Figure legends - Expanded View Figure legends

- Please make sure that all the funding information is also entered into the online submission system and is complete and similar to the one in the manuscript text file (as part of the Acknowledgements).

- Please make sure that the number "n" for how many independent experiments were performed, their nature (biological versus technical replicates), the bars and error bars (e.g. SEM, SD) and the test used to calculate p-values is indicated in the respective figure legends (for main, EV and Appendix figures) of the final revised manuscript. Please also check that exact p-values are provided in the legend, and that these fit to those shown in the figure. Please provide statistical testing where applicable. Please avoid the phrase 'independent experiment', but clearly state if these were biological or technical replicates. Please also indicate (e.g. with n.s.) if testing was performed, but the differences are not significant. In case n=2, please show the data as separate datapoints without error bars and statistics. See also:
<http://www.embopress.org/page/journal/14693178/authorguide#statisticalanalysis>

If n<5, please show single datapoints for diagrams. It seems 'non significance' (n.s.) is not indicated in most diagrams.

- Please add to the legend of each figure a 'Data Information' section explaining the statistics used or providing information regarding replicates and scales. See:

- Please use our reference format:
<http://www.embopress.org/page/journal/14693178/authorguide#referencesformat>

- We now request the publication of original source data with the aim of making primary data more accessible and transparent to the reader. Our source data coordinator will contact you to discuss which figure panels we would need source data for and will also provide you with helpful tips on how to upload and organize the files.

- Our journal encourages inclusion of *data citations in the reference list* to directly cite datasets that were re-used and obtained from public databases. Data citations in the article text are distinct from normal bibliographical citations and should directly link to the database records from which the data can be accessed. In the main text, data citations are formatted as follows: "Data ref: Smith et al, 2001" or "Data ref: NCBI Sequence Read Archive PRJNA342805, 2017". In the Reference list, data citations must be labeled with "[DATASET]". A data reference must provide the database name, accession number/identifiers and a resolvable link to the landing page from which the data can be accessed at the end of the reference. Further instructions are available at:
<http://www.embopress.org/page/journal/14693178/authorguide#referencesformat>

- The Data Availability section (DAS) should be restricted to large primary datasets produced in this study (e.g. RNA-seq, ChIP-seq, structural and array data) that need to be deposited in an appropriate public database. Please remove any other information from the DAS, remove the referee tokens, add direct links to each dataset and make sure these are public latest upon publication of the study.

- Please upload a complete author checklist, which you can download from our author guidelines (<https://www.embopress.org/page/journal/14693178/authorguide>). Please insert page numbers in the checklist to indicate where the requested information can be found in the manuscript. The completed author checklist will also be part of the RPF.

- All Materials and Methods need to be described in the main text using our 'Structured Methods' format, which is required for all research articles. According to this format, the Methods section should include a Reagents and Tools Table (listing key reagents, experimental models, software, and relevant equipment and including their sources and relevant identifiers), uploaded as separate file (thus please remove this table from the manuscript text file), followed by a Methods section in which we encourage the authors to describe their methods using a step-by-step protocol format with bullet points, to facilitate the adoption of the methodologies across labs. More information on how to adhere to this format as well as downloadable templates (.doc) for the Reagents and Tools Table can be found in our author guidelines (section 'Structured Methods'):

- Please move all the results and discussion parts (and related references) from the Appendix to the main manuscript text file. The Appendix should contain only Appendix Figures or Tables with their legends. Do not upload Appendix items separately (thus, please include the Appendix tables with the Appendix). The Appendix should be supplied as a single pdf file labeled Appendix, have page numbers and needs to include a table of content (TOC) on the first page (with page numbers) and legends for all content (best directly below each Appendix item). Please follow the nomenclature Appendix Figure Sx, Appendix Table Sx etc. throughout the text, and also label the figures and tables according to this nomenclature. Please remove author information and affiliations from the title page of the Appendix. It is sufficient to have 'Appendix for:') as title, followed by the TOC.

In addition, I would need from you:

- a short, two-sentence summary of the manuscript (not more than 35 words).
- two to four short (!) bullet points highlighting the key findings of your study (two lines each).
- a schematic summary figure that provides a sketch of the major findings (not a data image) in jpeg or tiff format (with the exact width of 550 pixels and a height of not more than 400 pixels) that can be used as a visual synopsis on our website.

Best,

Referee #1:

The authors have addressed my prior concerns adequately. This is an interesting study that has potential impact for an area of high interest to investigators working in the area of immune oncology, especially in the realm of combination therapy.

Referee #2:

Thanks to the authors for having responded well to all my concerns.

Referee #3:

The mechanism behind a critical finding of the manuscript: decrease in PTR tumor growth upon IFNAR1 KD has not been investigated.

All editorial and formatting issues were resolved by the authors.

Dear Dr. Ebos,

Thank you for the submission of your further revised manuscript to our editorial offices. Before I can proceed with formal acceptance, I have these further editorial requests I ask you to address in a final revised manuscript:

- Please provide the abstract written in present tense.
- Please name the methods section just 'Methods'. Moreover, please remove the section 'Contact for reagent and resource sharing' from the Methods. Contact information for the corresponding author is already provided on the title page.
- The Data Availability section (DAS) is restricted to large datasets generated in this study and deposited. Please remove all information from this section and add direct links to the deposited datasets.
- Regarding data quantification and statistics, please check again that the number "n" for how many independent experiments were performed, their nature (biological versus technical replicates), the bars and error bars (e.g. SEM, SD) and the test used to calculate p-values are indicated in the respective figure legends (also for potential EV figures and all those in the final Appendix). Please also check that all the p-values are explained in the legend, and that these fit to those shown in the figure. Please provide statistical testing where applicable. Please also clearly state if these were biological or technical replicates. Please also indicate (e.g. with n.s.) if testing was performed, but the differences are not significant. In case n=2, please show the data as separate datapoints without error bars and statistics.

If n<5, please show single datapoints for diagrams. Moreover:

- Please note that the exact p values are not provided in the legends of figures EV 1a; EV 3a-i; EV 4c-g; EV 5b-d, f, h.
- Please indicate the statistical test used for data analysis in the legends of figures EV 2b-e; EV 3e-f, g-i; EV 4c-g; EV 5b-d, f, h.
- Please note that in figures 1i; 3e; 5i-j; 6d-e, g-h, j-l; 7a-d; EV 1a; EV 2e; there is a mismatch between the annotated p values in the figure legend and the annotated p values in the figure file that should be corrected.
- Please note that the box plots need to be defined in terms of minima, maxima, centre, bounds of box, and percentile in the legends of figures 4h, j; 6a; EV 2b-e; EV 3f-g.
- Please note that information related to n is missing in the legends of figures 3a-b, d-e; EV 5h.
- Please note that the error bars are not defined in the legends of figures 2f-i; 6d-e, g-h, j-k; EV 1a; EV 2e; EV 3a, c-e, g-h; EV 4c-f; EV 5b-d, f-h.

- Please name the section in italics below each legend 'Data Information'. See:
<https://www.embopress.org/page/journal/14693178/authorguide#figureformat>

- Please make sure that all figure panels (main, EV and Appendix figures) are called out separately and sequentially. Presently, a callout to panel 4K seems missing. Moreover, there is still a callout to 'Appendix results' left on page 9. Please check.
- Please acknowledge BioRender with a paragraph (named Graphics) in the Methods section, indicating which panels/objects were created using BioRender.com. Then, please remove all the mention of BioRender from the acknowledgements.
- Please make sure that all the funding information is also entered into the online submission system and that it is complete and similar to the one in the acknowledgement section of the manuscript text file. Presently, the Roswell Park Alliance Foundation (RPAF) seems missing from the submission system. Moreover, please remove the text in the Comments box in the online system and enter each funder separately.
- There is a nearly empty page (page 3) in the Appendix file. I think Appendix Fig. S1 can be moved there.
- In the Reagents and tools table no information is entered for 'Bacterial and virus strains' and 'Biological samples'. If these do not apply for this study, please remove these from the table.
- Please move the primer information from the Appendix (Appendix Table S4) to the Reagents and tools table. Then, please remove this table from the Appendix, and update the Appendix TOC and all affected callouts.

Best,

All editorial and formatting issues were resolved by the authors.

Dr. John Ebos
ROSWELL PARK CANCER INSTITUTE CORP
Cancer Genetics; Medicine; Experimental Therapeutics
Elm & Carlton Streets
buffalo, ny 14263
United States

Dear Dr. Ebos,

I am very pleased to accept your manuscript for publication in the next available issue of EMBO reports. Thank you for your contribution to our journal.

Yours sincerely,
